# DEPTH SEPARATION WITH MULTILAYER MEAN-FIELD NETWORKS

**Yunwei Ren**
Carnegie Mellon University
yunweir@andrew.cmu.edu

**Mo Zhou**
Duke University
mozhou@cs.duke.edu

**Rong Ge**
Duke University
rongge@cs.duke.edu

## ABSTRACT

Depth separation—why a deeper network is more powerful than a shallower one—has been a major problem in deep learning theory. Previous results often focus on representation power. For example, Safran et al. (2019) constructed a function that is easy to approximate using a 3-layer network but not approximable by any 2-layer network. In this paper, we show that this separation is in fact algorithmic: one can learn the function constructed by Safran et al. (2019) using an overparameterized network with polynomially many neurons efficiently. Our result relies on a new way of extending the mean-field limit to multilayer networks, and a decomposition of loss that factors out the error introduced by the discretization of infinite-width mean-field networks.

## 1 INTRODUCTION

One of the mysteries in deep learning theory is why we need deeper networks. In the early attempts, researchers showed that deeper networks can represent functions that are hard for shallow networks to approximate(Eldan & Shamir, 2016; Telgarsky, 2016; Poole et al., 2016; Daniely, 2017; Yarotsky, 2017; Liang & Srikant, 2017; Safran & Shamir, 2017; Poggio et al., 2017; Safran et al., 2019; Malach & Shalev-Shwartz, 2019; Vardi & Shamir, 2020; Venturi et al., 2022; Malach et al., 2021). In particular, seminal works of Eldan & Shamir (2016); Safran et al. (2019) constructed a simple function ($f_*(\boldsymbol{x}) = \mathrm{ReLU}(1 - \|\boldsymbol{x}\|)$) which can be computed by a 3-layer neural network but cannot be approximated by a 2-layer network.

However, these results are only about the *representation power* of neural networks and do not guarantee that *training* a deep neural network from reasonable initialization can indeed learn such functions. In this paper, we prove that one can train a neural network that approximates $f_*(\boldsymbol{x}) = \mathrm{ReLU}(1 - \|\boldsymbol{x}\|)$ to any desired accuracy – this gives an *algorithmic separation* between the power of 2-layer and 3-layer networks.

To analyze the training dynamics, we develop a new framework to generalize mean-field analysis of neural networks (Chizat & Bach, 2018; Mei et al., 2018) to multiple layers. As a result, all the layer weights can change significantly during the training process (unlike many previous works on neural tangent kernel or fixing lower-layer representations). Our analysis also gives a decomposition of loss that allows us to decouple the training of multiple layers.

In the remainder of the paper, we first introduce our new framework for multilayer mean-field analysis, then give our main result and techniques. We discuss several related works in the algorithmic aspect for depth separation in Section 1.3. Similar to standard mean-field analysis, we first consider the infinite-width dynamics in Section 3, then we discuss our new ideas in discretizing the result to a polynomial-size network (see Section 4).

### 1.1 MULTI-LAYER MEAN-FIELD FRAMEWORK

We propose a new way to extend the mean-field analysis to multiple layers. For simplicity, we state it for 3-layer networks here. See Appendix A for the general framework. In short, we break the middle layer into two linear layers and restrict the size of the layer in between. More precisely, we

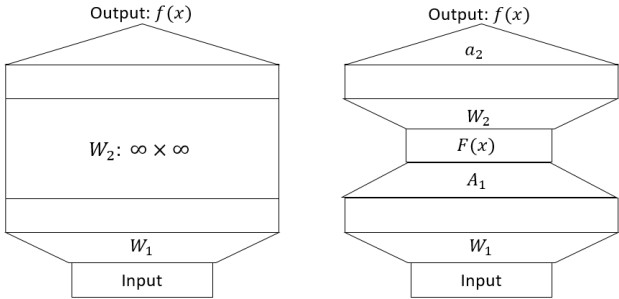

Figure 1: Difference between previous Nguyen & Pham (2020) (Left) and our framework (Right).

define

$$f(\boldsymbol{x}) = \frac{1}{m_2}\boldsymbol{a}_2^\top \sigma(\boldsymbol{W}_2\boldsymbol{F}(\boldsymbol{x})), \quad \boldsymbol{F}(\boldsymbol{x}) = \frac{1}{m_1}\boldsymbol{A}_1\sigma(\boldsymbol{W}_1\boldsymbol{x}),$$

where $\boldsymbol{W}_1 \in \mathbb{R}^{m_1 \times d}$, $\boldsymbol{A}_1 \in \mathbb{R}^{D \times m_1}$, $\boldsymbol{W}_2 \in \mathbb{R}^{m_2 \times D}$ $\boldsymbol{a}_2 \in \mathbb{R}^{m_2}$ are the parameters, and $\boldsymbol{F}(\boldsymbol{x}) \in \mathbb{R}^D$ represents the hidden feature. See Figure 1 for an illustration. Later we will refer to the step of $\boldsymbol{x} \mapsto \boldsymbol{F}(\boldsymbol{x})$ as the first layer and $\boldsymbol{F}(\boldsymbol{x}) \mapsto f(\boldsymbol{x})$ as the second layer, even though both of them actually are two-layer networks.

In the infinite-width limit, we will fix hidden feature dimension $D$ and let the number of neurons $m_1, m_2$ go to infinity. Then, we get the infinite-width network

$$f(\boldsymbol{x}) = \mathop{\mathbb{E}}_{(a_2,\boldsymbol{w}_2)\sim\mu_2} a_2\sigma(\boldsymbol{w}_2 \cdot \boldsymbol{F}(\boldsymbol{x})), \quad F_i(\boldsymbol{x}) = \mathop{\mathbb{E}}_{(a_1,\boldsymbol{w}_1)\sim\mu_{1,i}} a_1\sigma(\boldsymbol{w}_1 \cdot \boldsymbol{x}), \quad \forall i \in [D],$$

where $(\mu_{1,i})_{i\in[D]}$ are distributions over $\mathbb{R}^{1+d}$ with a shared marginal distribution over $\boldsymbol{w}_1$, and $\mu_2$ is a distribution over $\mathbb{R}^{1+D}$. Note that, unlike the formulation in Nguyen & Pham (2020), here the hidden layers are described using distributions of neurons, whence are automatically invariant under permutation of neurons, which is one of the most important properties of mean-field networks. One can choose $\mu_1, \mu_2$ to be empirical distributions over finitely many neurons to recover a finite-width network. In fact, we will do so in most parts of the paper so that our results apply to finite-width networks of polynomially many neurons. The network can be viewed as a 3-layer network with intermediate layer $\boldsymbol{W}_2\boldsymbol{A}$, which is low rank. This is reminiscent of the bottleneck structure used in ResNet (He et al. (2016)) and has also been used in previous theoretical analyses such as Allen-Zhu & Li (2020) for other purposes.

**Learner network** Now we are ready to introduce the specific network that we use to learn the target function. We set $D = 1$ and couple $a_1$ with $\boldsymbol{w}_1$.

$$\begin{cases} F(\boldsymbol{x}) = F(\boldsymbol{x}; \mu_1) := \mathop{\mathbb{E}}_{\boldsymbol{w}\sim\mu_1} \left\{ \|\boldsymbol{w}\| \, \sigma(\boldsymbol{w} \cdot \boldsymbol{x}) \right\}, \\ f(\boldsymbol{x}) = f(\boldsymbol{x}; \mu_2, \mu_1) := \mathop{\mathbb{E}}_{(w_2,b_2)\sim\mu_2} \sigma(w_2 F(\boldsymbol{x}; \mu_1) + b_2). \end{cases} \tag{1}$$

Here, $\sigma$ is the ReLU activation, and $\mu_1 \in \mathscr{P}(\mathbb{R}^d)$ and $\mu_2 \in \mathscr{P}(\mathbb{R}^2)$ are distributions encoding the weights of the first and second hidden layers, respectively. We multiply each first layer neuron by $\|\boldsymbol{w}\|$ to make $F$ more regular. This 2-homogeneous parameterization is also used in Li et al. (2020) and Wang et al. (2020). In most parts of the paper, $\mu_1$ and $\mu_2$ are empirical distributions over polynomially many neurons. We use $\mu_1, \mu_2$ to unify the notations in discussions on infinite- and finite-width networks.

Restricting the intermediate layer to have only one dimension ($D = 1$) is sufficient as one can learn $\boldsymbol{x} \mapsto \alpha \|\boldsymbol{x}\|$ for some $\alpha \in \mathbb{R}$ with the first layer $F(\boldsymbol{x})$ and $\alpha \|\boldsymbol{x}\| \mapsto \sigma(1 - \|\boldsymbol{x}\|)$ with the second layer. For the network that computes $F(\boldsymbol{x})$, we do not need a bias term as the intended function is homogeneous in $\boldsymbol{x}$. Though we restrict the first layer to be positive, it does not restrict the representation power of the network as the second layer can be either positive or negative. For the second layer, even though a single neuron is sufficient, we follow the framework and over-parameterize the network.

## 1.2 MAIN RESULT AND OUR TECHNIQUES

Our main result applies the framework in the previous section to the function constructed in Safran et al. (2019) (see details in Section 2). Informally, we prove:[1]

**Theorem 1.1** (Main result, Informal). *Given the learner network defined in (1) with input dimension $d$, for any $\epsilon > 0$, we can choose layer widths as $m_1 = \text{poly}(d, 1/\epsilon)$, $m_2 = \Theta(1)$ so that, with probability at least $1 - 1/\text{poly}(d, 1/\varepsilon)$ over random initialization, running a simple variant of gradient flow[2] reduces the loss $\mathcal{L} := \mathbb{E}_{\boldsymbol{x}} \left\{ (f_*(\boldsymbol{x}) - f(\boldsymbol{x}))^2 \right\} /2$ to $\varepsilon$ within $T = \text{poly}(d, 1/\epsilon)$ time.*

This result shows that one can train a multilayer neural network to learn the function $\text{ReLU}(1-\|x\|)$ that cannot be approximated by any 2-layer network. There are some technical details caused by the choice of a heavy-tail input distribution in Safran et al. (2019) which we discuss in Section 2.

To prove such a result, we first characterize the infinite-width dynamics (see Section 3). In particular, we show that in the infinite-width dynamics, the first layer will always compute a multiple of $\|\boldsymbol{x}\|$, while the second layer will behave like a single neuron.

However, it is often difficult to discretize such an infinite-width analysis to a polynomial-width network. The main difficulty is in the potential amplification of error in the network: if at the beginning, the first layer is $\delta$-close to computing a multiple of $\|x\|$, this $\delta$ value can potentially increase exponentially during the training process (Mei et al. (2018)). Given the large polynomial training time for our dynamics, this exponential increase would not be acceptable.

To fix this issue, we partition the analysis into two phases, and for the time-consuming second phase, we rely on a decomposition of the loss function:

$$\mathcal{L} := \frac{1}{2} \mathop{\mathbb{E}}_{\boldsymbol{x} \sim \mathcal{D}} \left\{ (f_*(\boldsymbol{x}) - f(\boldsymbol{x}))^2 \right\} \approx \frac{1}{2} \mathop{\mathbb{E}}_{\boldsymbol{x}} \left\{ (f_*(\boldsymbol{x}) - \tilde{f}(\boldsymbol{x}))^2 \right\} + \frac{\bar{w}_2^2}{2} \mathop{\mathbb{E}}_{\boldsymbol{x}} \left\{ (\tilde{F}(\boldsymbol{x}) - F(\boldsymbol{x}))^2 \right\}. \quad (2)$$

Here $\tilde{F}(\boldsymbol{x})$ is a multiple of $\|\boldsymbol{x}\|$ that is close to the actual first-layer output $F(\boldsymbol{x})$, $\tilde{f}(\boldsymbol{x})$ is the output of the network if the first layer is replaced by $\tilde{F}(x)$ – that is, if the first layer actually computes a multiple of $\|\boldsymbol{x}\|$ (see (5) for precise definition). The first term therefore characterizes the loss conditioned on a perfect first-layer; while the second term characterizes the difference between the first-layer output and a multiple of $\|\boldsymbol{x}\|$. We show that the gradients of these two terms do not affect each other, at least approximately. Therefore, we can view the training process as simultaneously doing two things: minimizing the loss given a good first-layer representation (reducing first term), and making first-layer output closer to a multiple of $\|x\|$ (reducing second term). We believe such a decomposition highlights how the lower-layer in the neural network receives useful gradient information to learn good representation for this particular objective.

## 1.3 RELATED WORKS

**Algorithmic aspect of depth separation** There have been other works that add algorithmic insights into depth separation. Allen-Zhu & Li (2020) showed that multi-layer quadratic networks can learn certain target functions in a hierarchical way, which cannot be learned by any kernel methods or shallow neural networks. Our work deals with more standard neural network architectures and target functions. A concurrent work Safran & Lee (2021) considers a similar problem as ours, where they show that GD with a certain three-layer network can learn the ball indicator which is not approximable by any two-layer network. Conceptually the main difference between our results lies in the training dynamics – the first layer of Safran & Lee (2021) is fixed while we train both layers. This leads to very different training dynamics and proof techniques.

**Overparametrized Neural Networks** One line of works studied the optimization of overparameterized neural network which couples the training dynamics to kernel regression with neural tangent kernel (NTK) (e.g., Jacot et al., 2018; Allen-Zhu et al., 2018b; Du et al., 2018). However, it is shown

---

[1]We say some quantity $a$ is $\text{poly}(d, 1/\varepsilon)$ if it is bounded by $C(d/\varepsilon)^C$ for some universal constant $C > 0$ that may change across lines.

[2]Though gradient flow, strictly speaking, is not a proper algorithm, it is common to use it as a surrogate for gradient descent in theoretical analysis. See Appendix E for discussions on how to convert the argument to a gradient descent one.

that neural network behaves like kernel methods in NTK regime, and several lower bounds have been developed (Yehudai & Shamir, 2019; Wei et al., 2019; Ghorbani et al., 2019; 2020). Our training dynamics is not in the NTK regime as all the weights change significantly. Another line of works studied the optimization of overparameterized neural network in the mean-field limit (Mei et al., 2018; Chizat & Bach, 2018; Nitanda & Suzuki, 2017; Wei et al., 2019; Rotskoff & Vanden-Eijnden, 2018; Sirignano & Spiliopoulos, 2020). Chizat et al. (2019) showed that the parameters can move away from its initialization in mean-field regime and learn useful features, which is different from NTK regime. However, most of the existing works require exponential/infinite number of neurons and do not provide a polynomial convergence rate. See more discussions in Appendix A.

**Multi-layer mean-field**   Although mean-field analysis has been successful for the optimization of two-layer overparameterized network, it is not easy to extend it to multiple-layer network since the width of intermediate layer goes to infinity. Many works have tried to address this issue to generalize mean-field analysis to deep networks. See e.g., Nguyen & Pham (2020); Pham & Nguyen (2021); Araújo et al. (2019); Sirignano & Spiliopoulos (2021); Fang et al. (2021); Lu et al. (2020); Ding et al. (2021) and references therein. Unlike most of the existing works, our multi-layer mean-field framework still has finite hidden feature dimension while the number of neurons can go to infinity to become a distribution of neurons. See Section 1.1 and Appendix A for more discussions.

**Mildly overparameterized neural networks**   Recently there are many works that consider the problem of learning certain target function with mildly overparameterized (polynomial size) network (Allen-Zhu et al., 2018a; Allen-Zhu & Li, 2019; Bai & Lee, 2019; Dyer & Gur-Ari, 2019; Woodworth et al., 2020; Bai et al., 2020; Huang & Yau, 2020; Chen et al., 2020; Li et al., 2020; Wang et al., 2020; Zhou et al., 2021). In particular, these works are different from the typical mean-field analysis where usually the infinite-width network are considered, or the typical NTK analysis where neural network behaves like kernel method. Our work is in a similar direction, but we need new insights to extend the discretization to our new multilayer framework.

## 2  PRELIMINARIES

In this section, we discuss the additional technical conditions for the input distributions in Safran et al. (2019), and how we deal with this in the training process.

**Notations**   For a vector $\boldsymbol{x}$, we let $\|\boldsymbol{x}\|$ denote its Euclidean norm. We use $a = b \pm c$ as a shorthand for the condition $a \in [b - |c|, b + |c|]$. For a distribution $\mu$, we write $\boldsymbol{v} \in \mu$ for the condition $\boldsymbol{v}$ is in the support of $\mu$. Other notations we use are mostly standard. We usually use $\boldsymbol{v}_1$ and $\boldsymbol{w}_1$ to denote a first layer neuron, and $(v_2, r_2)$ and $(w_2, b_2)$ to denote a second layer neuron. Keeping two sets of notations for neurons is intentional. When we are taking expectations over neurons, we use $\boldsymbol{w}_1$ and $(w_2, b_2)$. When considering a single neuron, we use $\boldsymbol{v}_1$ and $(v_2, r_2)$. For vectors, we write $\bar{\boldsymbol{v}} := \boldsymbol{v}/\|\boldsymbol{v}\|$. We will use $\mathbb{E}_{\boldsymbol{x}}$ as a shorthand for $\mathbb{E}_{\boldsymbol{x} \sim \mathcal{D}}$ when it is clear from the context. We also use $\boldsymbol{v} \in \mu$ as a shorthand for $\boldsymbol{v} \in \text{supp}(\mu)$.

**Target Function and Input Distribution**   The target function we consider is $f_*(\boldsymbol{x}) = \sigma(1 - \|\boldsymbol{x}\|)$, where $\sigma : \mathbb{R} \to \mathbb{R}$ is the ReLU activation. To describe the input distribution, first, we define $\varphi(\boldsymbol{x}) := \left(\frac{R_d}{\|\boldsymbol{x}\|}\right)^{d/2} J_{d/2}(2\pi R_d \|\boldsymbol{x}\|)$, where $R_d = \frac{1}{\sqrt{\pi}}(\Gamma(d/2+1))^{1/d}$ and $J_\nu$ is the Bessel function of the first kind of order $\nu$. Let $\alpha, \beta > 0$ be the universal constants from Safran et al. (2019) (cf. the proof of Theorem 5). We assume the inputs $\boldsymbol{x} \in \mathbb{R}^d$ are sampled from the distribution $\mathcal{D}$ whose density is given by $\boldsymbol{x} \mapsto (\sqrt{d}\beta\alpha)^d \varphi^2(\sqrt{d}\beta\alpha\boldsymbol{x})$. It has been verified in Eldan & Shamir (2016) and Safran et al. (2019) that this is indeed a valid probability distribution. Also, note that $\mathcal{D}$ is a spherically symmetric distribution. For more properties of $\mathcal{D}$, see Appendix B.2. By Theorem 5 of Safran et al. (2019), no two-layer networks of width $\text{poly}(d, 1/\varepsilon)$ can approximate $f_*$ to accuracy $\varepsilon$ in $L^2(\mathcal{D})$.[3] This distribution is heavy-tailed in the sense that $\mathbb{E}_{x \sim \mathcal{D}}[\|x\|^2]$ is undefined. The choice of such heavy-tailed distribution is mostly required for proving the lower bound. Our training result holds for most reasonable spherically symmetric distributions.

---

[3]Strictly speaking, the result in Safran et al. (2019) requires $\varepsilon = O(1/d^6)$. Even in that regime, our algorithm learns the function using $\text{poly}(d)$ neurons, which is not achievable by any two-layer network, therefore it is still a valid separation.

**Training Algorithm and Main Result** We use gradient flow with clipping over MSE loss to train a polynomial-size network. We write the loss as

$$\mathcal{L} = \mathcal{L}(\mu_1, \mu_2) = \frac{1}{2} \mathop{\mathbb{E}}_{\boldsymbol{x} \sim \mathcal{D}} \left\{ (f_*(\boldsymbol{x}) - f(\boldsymbol{x}))^2 \right\} =: \mathop{\mathbb{E}}_{\boldsymbol{x}} \mathcal{L}(\boldsymbol{x}), \tag{3}$$

Define $S(\boldsymbol{x}) = (f_*(\boldsymbol{x}) - f(\boldsymbol{x})) \mathbb{E}_{w_2, b_2} \{\sigma'(w_2 F(\boldsymbol{x}) + b_2) w_2\}$. One can verify that the dynamics of the neurons are given by

$$\begin{cases} \dot{\boldsymbol{v}}_1 = \mathop{\mathbb{E}}_{\boldsymbol{x} \sim \mathcal{D}} \left\{ \Pi_{R_{\boldsymbol{v}_1}} \left[ S(\boldsymbol{x}) \left( \bar{\boldsymbol{v}}_1 \sigma(\boldsymbol{v}_1 \cdot \boldsymbol{x}) + \|\boldsymbol{v}_1\| \sigma'(\boldsymbol{v}_1 \cdot \boldsymbol{x}) \boldsymbol{x} \right) \right] \right\}, \\ \dot{v}_2 = \mathop{\mathbb{E}}_{\boldsymbol{x} \sim \mathcal{D}} \left\{ \Pi_{R_{v_2}} \left[ (f_*(\boldsymbol{x}) - f(\boldsymbol{x})) \sigma'(v_2 F(\boldsymbol{x}) + r_2) F(\boldsymbol{x}) \right] \right\}, \\ \dot{r}_2 = \mathop{\mathbb{E}}_{\boldsymbol{x} \sim \mathcal{D}} \left\{ \Pi_{R_{r_2}} \left[ (f_*(\boldsymbol{x}) - f(\boldsymbol{x})) \sigma'(v_2 F(\boldsymbol{x}) + r_2) \right] \right\}, \end{cases} \tag{4}$$

where $\Pi_R$ stands for the projection to the ball of radius $R$, and $R_{\boldsymbol{v}_1} = \Theta(d)$, $R_{v_2} = \Theta(d^3)$, $R_{r_2} = \Theta(1)$ are the projection threshold. We add these additional gradient clipping because without them the gradients are not well-defined due to the heavy-tailed property of the distribution $\mathcal{D}$. Note that gradient clipping is indeed widely used in practice to avoid exploding gradients (Pascanu et al., 2013; Zhang et al., 2020). In fact, we believe our optimization result without using gradient clipping would still be true for a general spherically symmetric distribution $\mathcal{D}$ as long as it is more regular.

To initialize the learner network, we use $\text{Unif}(\sigma_1 \mathbb{S}^{d-1})$ to initialize the first layer weights $\boldsymbol{w}_1$, $\mathcal{N}(0, \sigma_2^2)$ for the second layer weights $w_2$, and choose all second layer bias $b_2$ to be $\sigma_r$, where $\sigma_1, \sigma_2, \sigma_r$ are some small positive real numbers. We initialize $\boldsymbol{w}_1$ on the sphere instead using a Gaussian only for technical convenience. We initialize the bias term to be a small positive value so that all second layer neurons are activated at initialization to avoid zero gradient.

Now we are ready to give our main result. It shows that gradient flow with a polynomial-sized learner network (1) defined in our mean-field framework can learn $f_*(\boldsymbol{x}) = \sigma(1 - \|\boldsymbol{x}\|)$ efficiently, which is not approximable by any two-layer network (Safran et al., 2019).

**Theorem 2.1** (Main result). *Given the learner network defined in (1) with initialization described above and suppose we run gradient flow, assuming it exists, on this finite-width network with clipping (4) on loss (3). Then, for any $\epsilon > 0$, we can choose $m_1 = \text{poly}_{m_1}(d, 1/\epsilon)$, $m_2 = \Theta(1)$, $\sigma_1 = 1/\text{poly}_{\sigma_1}(d, 1/\epsilon)$, $\sigma_2 = 1/\text{poly}_{\sigma_2}(d, 1/\epsilon)$, $\sigma_r = \Theta(1)$, $R_{\boldsymbol{v}_1} = \Theta(d)$, $R_{v_2} = \Theta(d^3)$ and $R_{r_2} = \Theta(1)$ so that with probability at least $1 - 1/\text{poly}(d, 1/\varepsilon)$ over the random initialization, we have loss $\mathcal{L} \leq \varepsilon$ within $T = \text{poly}(d, 1/\epsilon)$ time.*

## 3 THE INFINITE-WIDTH DYNAMICS

Our proof consists of analyzing the dynamics of the infinite-width mean-field network and controlling the discretization error. In this section, we characterized the infinite-width dynamics. For ease of presentation, we pretend there is no projection and the gradients are well-defined in this subsection and defer the discussion on handling the projections to Section 4.

First, note that both the input distribution $\mathcal{D}$ and the infinite-width network are spherically symmetric. That is, for any $\boldsymbol{x}, \boldsymbol{x}' \in \mathbb{R}^d$ with $\|\boldsymbol{x}\| = \|\boldsymbol{x}'\|$, the density/function value are the same. Any spherically symmetric $g : \mathbb{R}^d \to \mathbb{R}$ can be characterized by a function $h : [0, \infty) \to \mathbb{R}$ which satisfies $h(\|\boldsymbol{x}\|) = g(\boldsymbol{x})$. For convenience, we will abuse notation to also use $g : \mathbb{R} \to \mathbb{R}$ to denote this function $h$.

Assuming that the distribution $\mu_1$ of the first layer neurons is spherically symmetric, which is true at least at initialization, we can approximate the first layer with a simple function using the following lemma. The proof of it can be found in Appendix B.3.

**Lemma 3.1.** *Let $\mu$ be a spherically symmetric distribution. We have*

$$\mathop{\mathbb{E}}_{\boldsymbol{w} \sim \mu} \|\boldsymbol{w}\| \sigma(\boldsymbol{w} \cdot \boldsymbol{x}) = C_\Gamma \frac{\mathbb{E}_{\boldsymbol{w} \sim \mu} \|\boldsymbol{w}\|^2}{\sqrt{d}} \|\boldsymbol{x}\| \quad \text{where} \quad C_\Gamma := \frac{\Gamma(d/2)\sqrt{d}}{2\sqrt{\pi}\Gamma((d+1)/2)}.$$

*Note that, as $d \to \infty$, we have $C_\Gamma \to 1/\sqrt{2\pi}$, so $C_\Gamma$ is universally bounded for all $d$.*

This lemma implies that, in the infinite-width limit, we have $F(\boldsymbol{x}) = \alpha \lVert \boldsymbol{x} \rVert$ for some real $\alpha > 0$, at least at initialization. This suggests defining the infinite-width approximation as:

$$\alpha := \frac{C_\Gamma}{\sqrt{d}} \mathop{\mathbb{E}}_{\boldsymbol{w}_1 \sim \mu_1} \lVert \boldsymbol{w}_1 \rVert^2, \quad \tilde{F}(\boldsymbol{x}) := \alpha \lVert \boldsymbol{x} \rVert, \quad \tilde{f}(\boldsymbol{x}) := \mathop{\mathbb{E}}_{(w_2, b_2) \sim \mu_2} \sigma(w_2 \tilde{F}(\boldsymbol{x}) + r_2). \quad (5)$$

Note that (5) is well-defined no matter $\mu_1$ is infinite-width or not, though only in the infinite-width case will one have $F = \tilde{F}$. Later in Section 4 we will show that $F \approx \tilde{F}$ throughout the entire process in the discretization part of the proof.

For the infinite-width network, one can imagine that, thanks to the symmetry, as long as $\mu_1$ is spherically symmetric at time $t$, then no first layer neuron will change its direction and the change in norm is also uniform, i.e., it does not depend on the direction $\bar{\boldsymbol{v}}_1$. (See Appendix B.4 for the proof.) As a result, $\mu_1$ will remain spherically symmetric. Formally, one can show that, for any spherically symmetric $g : \mathbb{R}^d \to \mathbb{R}$, we have

$$\mathop{\mathbb{E}}_{\boldsymbol{x}} \{g(\boldsymbol{x}) \sigma(\boldsymbol{v} \cdot \boldsymbol{x})\} = \frac{C_\Gamma}{\sqrt{d}} \mathop{\mathbb{E}}_{\boldsymbol{x}} \{g(\boldsymbol{x}) \lVert \boldsymbol{x} \rVert\} \lVert \boldsymbol{v} \rVert \quad \text{and} \quad \mathop{\mathbb{E}}_{\boldsymbol{x}} \{g(\boldsymbol{x}) \sigma'(\boldsymbol{v} \cdot \boldsymbol{x}) \boldsymbol{x}\} = \frac{C_\Gamma}{\sqrt{d}} \mathop{\mathbb{E}}_{\boldsymbol{x}} \{g(\boldsymbol{x}) \lVert \boldsymbol{x} \rVert\} \bar{\boldsymbol{v}},$$

where $\bar{\boldsymbol{v}} = \boldsymbol{v} / \lVert \boldsymbol{v} \rVert$. Again, the proof of these two identities can be found in Appendix B.3. Apply these identities to $\dot{\boldsymbol{v}}_1$ with $g \equiv S$ and one can obtain

$$\dot{\boldsymbol{v}}_1 = \frac{2C_\Gamma}{\sqrt{d}} \mathop{\mathbb{E}}_{\boldsymbol{x}} \{S(\boldsymbol{x}) \lVert \boldsymbol{x} \rVert\} \boldsymbol{v}_1.$$

As a result, $\mu_1$ is always a uniform distribution over some sphere. Moreover, we have[4]

$$\dot{\alpha} = \mathop{\mathbb{E}}_{\boldsymbol{w}_1} \frac{\partial \alpha}{\partial \boldsymbol{w}_1} \frac{\mathrm{d}\boldsymbol{w}_1}{\mathrm{d}t} = \frac{4C_\Gamma^2}{d} \mathop{\mathbb{E}}_{\boldsymbol{x}} \{S(\boldsymbol{x}) \lVert \boldsymbol{x} \rVert\} \mathop{\mathbb{E}}_{\boldsymbol{w}_1} \lVert \boldsymbol{w}_1 \rVert^2 = \frac{4C_\Gamma}{\sqrt{d}} \mathop{\mathbb{E}}_{\boldsymbol{x}} \{S(\boldsymbol{x}) \lVert \boldsymbol{x} \rVert\} \alpha.$$

This implies that the dynamics of the first layer can also be characterized by $\alpha$ alone. This reduces the dynamics of the first layer to a single real number $\alpha$. That is, the outputs of the first layer depend only on $\alpha$ and $\boldsymbol{x}$, and the dynamics of $\alpha$ also depend only on $\alpha$ instead of every single neuron $\boldsymbol{w}_1$. In other words, we do not need to look at the actual dynamics of $\boldsymbol{w}_1$ in this infinite-width case. We will later show that the spread of the second layer is always small, hence the second layer can be approximated by $\alpha \lVert \boldsymbol{x} \rVert \mapsto \sigma(\bar{w}_2 \alpha \lVert \boldsymbol{x} \rVert + \bar{b}_2)$ where $(\bar{w}_2, \bar{b}_2) = \mathbb{E}(w_2, b_2)$. Combining these observations, one can characterize the dynamics of the entire network using three quantities: $\alpha$, $\bar{w}_2$ and $\bar{b}_2$.

We close this section with another interpretation of $\tilde{F}$, which is going to be handy in Section 4.2. Since we know that, in the idealized case, $F$ should be spherically symmetric. Hence, it makes sense to define the "idealized" $F$ to be the average over the sphere, that is, $\tilde{F}(\boldsymbol{x}) = \mathbb{E}_{\boldsymbol{x}' \in \lVert \boldsymbol{x} \rVert \mathbb{S}^{d-1}} F(\boldsymbol{x}')$. Note that in Lemma 3.1, the expectation is taken over the neurons while here it is over the inputs. However, similar to the proof of Lemma 3.1, one can still show that

$$\mathop{\mathbb{E}}_{\boldsymbol{x}' \in \lVert \boldsymbol{x} \rVert \mathbb{S}^{d-1}} F(\boldsymbol{x}') = \mathop{\mathbb{E}}_{\boldsymbol{w} \sim \mu_1} \mathop{\mathbb{E}}_{\boldsymbol{x}' \in \lVert \boldsymbol{x} \rVert \mathbb{S}^{d-1}} \lVert \boldsymbol{w} \rVert^2 \sigma(\bar{\boldsymbol{w}} \cdot \boldsymbol{x}) = \frac{C_\Gamma \mathbb{E}_{\boldsymbol{w} \sim \mu_1} \lVert \boldsymbol{w} \rVert^2}{\sqrt{d}} \lVert \boldsymbol{x} \rVert = \alpha \lVert \boldsymbol{x} \rVert.$$

In other words, these two derivations are equivalent. In some sense, this means that the infinite-width network can be interpreted as a symmetrization of the actual finite-width network.

## 4 Discretizing the Dynamics with Polynomial-size Network

In this section, we show how to discretize the infinite-width dynamics to get our main results. See Fig. 2 for simulation results. As we can see, even though the network has a finite width, at any time step, the function $f(x)$ is close to a function of the form $\boldsymbol{x} \mapsto \sigma(\bar{b}_2 - \bar{w}_2 \alpha \lVert \boldsymbol{x} \rVert)$, and throughout the training the second layer weights are well-concentrated.

Let $\delta_2 := \max_{(v_2, r_2), (v_2', r_2')} \lVert (v_2, r_2) - (v_2', r_2') \rVert$ be the spread of the second layer, we will split the training procedure into two stages. Recall that $(\bar{w}_2, \bar{b}_2) := \mathbb{E}_{(w_2, b_2) \sim \mu_2}(w_2, b_2)$. In Stage 1, $\bar{w}_2$ will decrease to $-\operatorname{poly}(d)\delta_2$. We show that after this condition is true, the projection operators in (4) can be ignored (that is, the corresponding terms never exceed the thresholds, see Lemma 4.1). In Stage 2, we show that the network can fit the target function in polynomial time.

---

[4]As in the standard mean-field arguments, we rescale the gradients by $m$ so that it does not go to 0 as $m \to \infty$. In most cases regarding gradient calculation, this is equivalent to using the formal rule $\partial_{\boldsymbol{v}} \mathbb{E}_{\boldsymbol{w}} g(\boldsymbol{w}) = \partial_{\boldsymbol{v}} g(\boldsymbol{v})$.

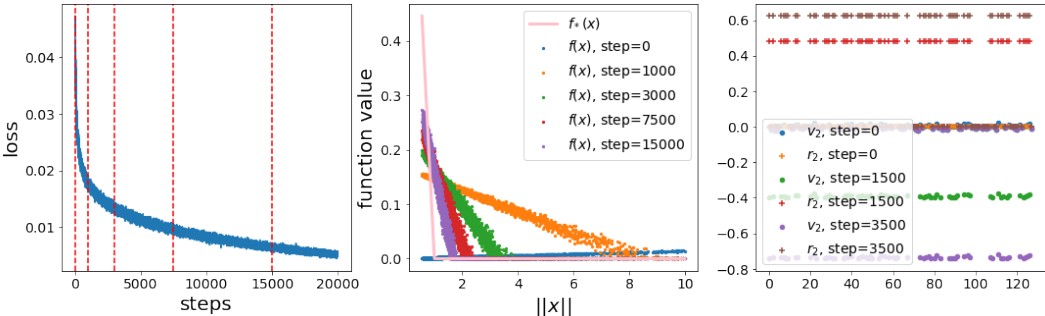

Figure 2: Simulation results. The left figure shows the loss during training. Each vertical dashed line corresponds to a time point plotted in the other two figures. The center figure depicts the shape of $f$ at certain steps. The right figure shows the values of the second-layer neurons at certain steps. One can observe that $f \approx \tilde{f}$ indeed holds, and the second layer neurons are concentrated around $(\bar{w}_2, \bar{b}_2)$, which matches our theoretical analysis. Simulation is performed on a finite-width network with widths $m_1 = 512$, $m_2 = 128$ and input dimension $d = 100$.

## 4.1 STAGE 1: REMOVING THE PROJECTIONS

Our first step shows that after a short amount of time in training, it is OK to ignore the projection operators in (4). To see why the projections can be ignored in certain circumstances, first note that if $f \approx \tilde{f}$, second layer neurons concentrate around their mean, $\bar{b}_2 = \Theta(1)$ and $\bar{w}_2 < 0$, then $f \approx \sigma(\bar{w}_2 \alpha \|x\| + \bar{b}_2)$ vanishes outside $\{\|x\| \le \Theta(1/|\bar{w}_2 \alpha|)\}$, whence the gradients also vanish for those large $x$. Meanwhile, by upper bounding the norm of the gradients, one can show that in order for the projections to be triggered, it is necessary for $\|x\|$ to be large. As a result, when $f$ decreases sufficiently fast, $f(x)$ will reach 0 before $\|x\|$ becomes too large. Formally, we have the following lemma, whose proof can be found in Appendix C.

**Lemma 4.1.** *Choose the projection threshold $R_{v_1} = \Theta(d)$, $R_{v_2} = \Theta(d^3)$ and $R_{r_2} = \Theta(1)$ in (4). Suppose that $\alpha = \Theta(1/\sqrt{d})$. Then, the projection operators in $\dot{r}_2$, $\dot{v}_1$ and $\dot{v}_2$ will no longer be activated if all second layer weights are nonpositive, $-\bar{w}_2 > \Theta(1)\delta_2$ for some large constant, and $-\bar{w}_2 \ge \Theta(1)/R_{v_2}$ for some large constant, respectively.*

Based on this lemma, we further split Stage 1 into three substages. We define $T_{1.1}$ to be the first time all second layer weights become negative, and $T_{1.2}$ and $T_{1.3}$ the first time $|\bar{w}_2|$ becomes $\Theta(d)\delta_2$ and $\Theta(1/R_{v_2})$, respectively. They represent the end time of Stage 1.1, 1.2, and 1.3, respectively. We require $|\bar{w}_2|$ to be $\Theta(d)\delta_2$ instead of $\Theta(1)\delta_2$ at the end of Stage 1.2 so that the starting state of Stage 1.3 is more regular. By definition and Lemma 4.1, after each substage, one more projection can be ignored, and all of them can be ignored after Stage 1.

The main lemma of Stage 1 is as follows. Recall that $R_{v_1}, R_{v_2}, R_{r_2}$ are the clipping thresholds.

**Lemma 4.2** (Stage 1, informal). *Define the end time of Stage 1 as $T_1 := \inf\{t \ge 0 : -\bar{w}_2(t) = C_1/R_{v_2}\}$ for some large constant $C_1$. Under the assumptions of Theorem 2.1, we have $T_1 \le \text{poly}(d, 1/\varepsilon)$ and the following conditions hold throughout Stage 1.*

(a) **Approximation error of the first layer.** *For each $v_1 \in \mu_1$, both the tangent movement and the radial spread can be controlled as $\|\bar{v}_1(t) - \bar{v}_1(0)\| \le \delta_{1,T}^{(1)}(t)$ and $\|v_1\|^2 = (1 \pm \delta_{1,R}^{(1)}(t)) \mathbb{E} \|w_1\|^2$, where $\delta_{1,T}^{(1)}$ and $\delta_{1,R}^{(1)}$ are two processes which are always small.*

(b) **Spread of the second layer.** *For any $(v_2, r_2), (v_2', r_2') \in \mu_2$, $\|(v_2, r_2) - (v_2', r_2')\|$ is small.*

(c) **Regularity conditions.** *$r_2 = \Theta(1)$ for all $(v_2, r_2) \in \mu_2$, $|\bar{w}_2| = O(1/R_{v_2}) = O(1/d^3)$ and $\alpha = \Theta(\sqrt{d}/R_{v_1}) = \Theta(1/d^{1.5})$.*

The first two conditions mean the approximation $f(x) \approx \sigma(\bar{w}_2 \alpha \|x\| + \bar{b}_2)$ is valid throughout Stage 1 and the third condition describes the shape of $f$ in Stage 1. To maintain these conditions, we use the so-called continuity argument, which can be viewed as a continuous version of mathematical induction. See Appendix B.1 for explanations of this technique.

With the approximation $F(\boldsymbol{x}) \approx \alpha \|\boldsymbol{x}\|$ and the fact $f(\boldsymbol{x})\sigma'(v_2 F(\boldsymbol{x}) + r_2) = f(\boldsymbol{x})$ for most $\boldsymbol{x}$, we can rewrite the dynamics of $v_2$ as

$$\dot{v}_2 \approx \mathop{\mathbb{E}}_{\boldsymbol{x}} \left\{ \Pi_{R_{v_1}} \left[ (f_*(\boldsymbol{x}) - f(\boldsymbol{x}))\alpha \|\boldsymbol{x}\| \right] \right\}.$$

Since $f$ is much flatter than $f_*$, $f$ is still $\Omega(1)$ when $f_*$ vanishes because of $\|\boldsymbol{x}\| \geq 1$. As a result, the RHS is always negative. In fact, we show that it is $-\Theta(\alpha \log d)$. Recall that $T_{1.2}$ is the time $|\bar{w}_2|$ reaches $\Theta(d\delta_2)$. If $\delta_2$ roughly remains constant, the time needed for Stage 1.1 and Stage 1.2 is proportional to the initial $\delta_2$. Then, we can make the initial $\delta_2$ small by selecting a small enough $\sigma_2$. This also helps control the movement of $\boldsymbol{v}_1$ and $r_2$ in Stage 1.1 and Stage 1.2 as their dynamics depend on $|w_2|$.

One also needs to show that $\delta_2$ cannot increase too much during Stages 1.1 and 1.2 to maintain the approximation $f(\boldsymbol{x}) \approx \sigma(\bar{w}_2 F(\boldsymbol{x}) + \bar{b}_2)$. Intuitively, this is because for inputs with small $\|\boldsymbol{x}\|$, the gradient $\nabla_{v_2}\mathcal{L}(\boldsymbol{x})$ does not depend on $(v_2, r_2)$ itself; for the inputs with a large norm, they cannot contribute too much to the gradient due to gradient clipping. As a result, the dynamics of $v_2$ are approximately uniform in Stage 1.1 and Stage 1.2, whence the distance between different $(v_2, r_2)$, $(v_2', r_2')$ stays small.

The same method does not work in Stage 1.3 as now the target value of $\bar{w}_2$ no longer depends on $\delta_2$, and we need a finer analysis for the first layer. Recall that, after Stage 1.2, the projection in $\dot{\boldsymbol{v}}_1$ can be ignored. Therefore, we can decompose $\dot{\boldsymbol{v}}_1$ along the radial and tangent direction as

$$\begin{aligned}
\dot{\boldsymbol{v}}_1 = \mathrm{Rad}(\dot{\boldsymbol{v}}_1) + \mathrm{Tan}(\dot{\boldsymbol{v}}_1) &= \langle \dot{\boldsymbol{v}}_1, \bar{\boldsymbol{v}}_1 \rangle \bar{\boldsymbol{v}}_1 + (\boldsymbol{I} - \bar{\boldsymbol{v}}_1 \bar{\boldsymbol{v}}_1^\top)\dot{\boldsymbol{v}}_1 \\
&= 2\mathop{\mathbb{E}}_{\boldsymbol{x}} \left\{ S(\boldsymbol{x})\sigma(\boldsymbol{v}_1 \cdot \boldsymbol{x}) \right\} + \|\boldsymbol{v}_1\| \mathop{\mathbb{E}}_{\boldsymbol{x}} \left\{ S(\boldsymbol{x})\sigma'(\boldsymbol{v}_1 \cdot \boldsymbol{x})(\boldsymbol{I} - \bar{\boldsymbol{v}}_1 \bar{\boldsymbol{v}}_1^\top)\boldsymbol{x} \right\}.
\end{aligned}$$

Then, we write $S(\boldsymbol{x}) \approx (f_*(\boldsymbol{x}) - f(\boldsymbol{x}))\bar{w}_2 = (f_*(\boldsymbol{x}) - \tilde{f}(\boldsymbol{x}))\bar{w}_2 + (\tilde{f}(\boldsymbol{x}) - f(\boldsymbol{x}))\bar{w}_2$. The terms related to $f_* - \tilde{f}$ is essentially what one should expect to have in the infinite-width dynamics. For those terms, the radial movement is uniform and tangent movement is $0$. Then, we bound terms related to $\tilde{f} - f$ using the radial spread and tangent movement of the first layer to obtain $\frac{\mathrm{d}}{\mathrm{d}t}\left(\delta_{1,R}^{(1)} + \delta_{1,T}^{(1)}\right) \lesssim \frac{O(1)}{d^{2.5}}\left(\delta_{1,R}^{(1)} + \delta_{1,T}^{(1)}\right)$ (cf. Lemma C.16). Though, with this bound, the error can grow exponentially fast $(\exp(t/d^{2.5}))$, this is sufficient since Stage 1.3 only takes $O(d^{1.5})$ time.

## 4.2 Stage 2: Fitting the Target Function

The goal of Stage 2 is for the gradient flow to converge to a point with loss at most $\varepsilon$ in polynomial time. The main difficulty in this stage is that we need to bound the approximation error of the first layer more carefully, as Stage 2 is potentially long and the brute-force estimations used in Stage 1 is too loose towards the end of training. We write $\bar{F} := F/\alpha$ and measure the approximation error using $\left\|\bar{F}|_{\mathbb{S}^{d-1}} - 1\right\|$ and $\left\|\bar{F} - \|\cdot\|_2\right\|_{L^2}$. Strictly speaking, for the $L^2$ error, we only consider those $\boldsymbol{x}$ with $\|\boldsymbol{x}\| \leq \Theta(1/|\bar{w}\alpha|) = \mathrm{poly}(d)$ since otherwise it can be ill-defined. This is valid because, as we have discussed earlier, $f$ vanishes for large $\boldsymbol{x}$. In Stage 2, $\mathbb{E}_{\boldsymbol{x}}$ always means $\mathbb{E}_{\|\boldsymbol{x}\| \leq \Theta(1/|\bar{w}_2\alpha|)}$ and, for the simplicity of presentation, we usually do not explicitly state this. The main result of Stage 2 is as follows.

**Lemma 4.3** (Stage 2, informal). *Define the end time of Stage 2 as $T_2 := \inf\{t \geq T_1 : \mathcal{L} = \varepsilon\}$. Under the assumptions of Theorem 2.1, we have $T_2 - T_1 \leq \mathrm{poly}(d, 1/\varepsilon)$ and the following conditions hold throughout Stage 2:*

(a) **Approximation error of the first layer.** *Both $\left\|\bar{F} - \|\cdot\|\right\|_{L^2}$ and $\left\|\bar{F}|_{\mathbb{S}^{d-1}} - 1\right\|_{L^\infty}$ are small.*

(b) **Spread of the second layer.** $\max_{(v_2, r_2),(v_2', r_2')} \|(v_2, r_2) - (v_2', r_2')\|$ *does not grow.*

(c) **Regularity conditions.** *The shape of $f$ is similar to the one shown in Figure 2.*

As we mentioned, the main technical challenge is to bound the approximation error of the first layer. The overall strategy is to first show that, in Stage 2, the $L^2$ error barely grows and then show that, as long as the $L^2$ error is small, the $L^\infty$ error can also be controlled. Unlike Stage 1, $|\bar{w}_2\alpha|$ is fairly large in Stage 2 and, as a result, the first layer can receive some signal from the loss function.

Intuitively, this signal should push the first layer to become closer to a multiple of $\|\boldsymbol{x}\|$ as that is what the global optimal solution would do. Formally, we first show the following approximation:

$$\mathcal{L} \approx \frac{1}{2} \mathbb{E}_{\boldsymbol{x}} \left\{ (f_*(\boldsymbol{x}) - \tilde{f}(\boldsymbol{x}))^2 \right\} + \frac{\bar{w}_2^2}{2} \mathbb{E}_{\boldsymbol{x}} \left\{ (\tilde{F}(\boldsymbol{x}) - F(\boldsymbol{x}))^2 \right\}, \tag{6}$$

in the sense that the gradients $\nabla_{\boldsymbol{v}_1}$ of both sides are approximately the same, where $\tilde{f}(x)$ is defined as $\mathbb{E}_{(w_2, b_2) \sim \mu_2} \sigma(w_2 \tilde{F}(x) + b_2)$. The first term of (6) measures the distance between the target function and the infinite-width network and the second term measures the approximation error of the first layer. In some sense, one can view this formula as a bias-variance decomposition for discretizing mean-field networks.

With this approximation in hand, we then show that, thanks to the 2-homogeneity of $F$, the first term, after certain normalization, does not affect the approximation error of the first layer. Meanwhile, since we are following the gradient flow, the second term can only decrease the approximation error.

To establish (6), we first decompose the loss function as

$$\mathcal{L} = \frac{1}{2} \mathbb{E}_{\boldsymbol{x}} \left\{ (f_*(\boldsymbol{x}) - \tilde{f}(\boldsymbol{x}))^2 \right\} + \frac{1}{2} \mathbb{E}_{\boldsymbol{x}} \left\{ (\tilde{f}(\boldsymbol{x}) - f(\boldsymbol{x}))^2 \right\} + \mathbb{E}_{\boldsymbol{x}} \left\{ (f_*(\boldsymbol{x}) - \tilde{f}(\boldsymbol{x}))(\tilde{f}(\boldsymbol{x}) - f(\boldsymbol{x})) \right\}$$
$$=: \mathcal{L}_1 + \mathcal{L}_2 + \mathcal{L}_3.$$

We claim that $\mathcal{L}_2$ is approximately the second term of (6) and the third term is approximately $0$[5]. Let $X_1$ be the largest spherically symmetric set on which $v_2 F(\boldsymbol{x}) + r_2 > 0$ for all $(v_2, r_2) \in \mu_2$. We show that those $\boldsymbol{x}$ outside $X_1$ contribute a little. Therefore, we can rewrite $\mathcal{L}_2$ as

$$\mathcal{L}_2 \approx \frac{1}{2} \mathbb{E}_{X_1} \left\{ \left( \mathbb{E}_{w_2, b_2} (w_2 \tilde{F}(\boldsymbol{x}) + b_2) - \mathbb{E}_{w_2, b_2} (w_2 F(\boldsymbol{x}) + b_2) \right)^2 \right\}$$
$$= \frac{\bar{w}_2^2}{2} \mathbb{E}_{X_1} \left\{ (\tilde{F}(\boldsymbol{x}) - F(\boldsymbol{x}))^2 \right\} \approx \frac{\bar{w}_2^2}{2} \mathbb{E}_{\boldsymbol{x}} \left\{ (\tilde{F}(\boldsymbol{x}) - F(\boldsymbol{x}))^2 \right\}.$$

Similarly, we can rewrite $\mathcal{L}_3$ as $\mathcal{L}_3 \approx \bar{w}_2 \mathbb{E}_{\boldsymbol{x}} \left\{ (f_*(\boldsymbol{x}) - \tilde{f}(\boldsymbol{x}))(\tilde{F}(\boldsymbol{x}) - F(\boldsymbol{x})) \right\}$. Recall from Section 3 that $\tilde{F}(\boldsymbol{x}) = \mathbb{E}_{\boldsymbol{x}' \in \|\boldsymbol{x}\| \mathbb{S}^{d-1}} F(\boldsymbol{x})$. With this in mind, one can easily verify that, for any spherically symmetric function $g : \mathbb{R}^d \to \mathbb{R}$, $\mathbb{E}_{\boldsymbol{x}} \{ g(\boldsymbol{x}) F(\boldsymbol{x}) \} = \mathbb{E}_{\boldsymbol{x}} \left\{ g(\boldsymbol{x}) \tilde{F}(\boldsymbol{x}) \right\}$. Setting $g = f_*(x) - \tilde{f}(x)$ gives $\mathcal{L}_3 \approx 0$. Combine these two estimations together and we obtain (6).

Provided that the $L^2$ error is always small, we show that, up to some higher order terms,

$$\left| \frac{\mathrm{d}}{\mathrm{d}t} \bar{F}(\bar{\boldsymbol{x}}) \right| \lesssim O(d^3) \left\| \bar{F} - \|\cdot\|_2 \right\|_{L^2}, \quad \forall \bar{\boldsymbol{x}} \in \mathbb{S}^{d-1}.$$

In words, the change of $\frac{\mathrm{d}}{\mathrm{d}t} \bar{F}(\boldsymbol{x})$ can be bounded by the $L^2$ error. Hence, $\left\| \bar{F}|_{\mathbb{S}^{d-1}} - 1 \right\|_{L^\infty}$ is always small as long as we choose a sufficiently large $m_1$ so that $\bar{F}(x)|_{x \in \mathbb{S}^{d-1}}$ is close to 1 at initialization. This should not be a surprise since, after all, in the infinite-width dynamics $\bar{F}(x)|_{x \in \mathbb{S}^{d-1}} = 1$. The formal proof of the above argument can be found in Section D.2.

Given that the approximation error can be controlled, one can then derive a convergence rate using the infinite-width dynamics. See Section D.3 for details.

## 5 CONCLUSION

In this paper we give a new framework for extending mean-field limit to multilayer networks, and use this framework to show that three-layer networks can learn a function that is not approximable by two-layer networks. There are still many open problems: for the current objective the loss is spherically symmetric so the first-layer neurons don't move much tangentially, what if the function is instead $\sigma(1 - \|P_S \boldsymbol{x}\|)$ where $P_S$ is projection to some unknown subspace? How about functions that require an intermediate layer of size more than 1? Can one generalize the saddle point analysis to deeper networks? We hope this work will be a starting point for understanding how deep neural networks can learn useful features.

---

[5]For the ease of presentation, here we are talking about the function values instead of the gradients. Strictly speaking, this is incorrect as the function value being small does not necessarily imply the gradient is small. The ideas, however, are essentially the same. See Section D.2 for the actual proof.

## ACKNOWLEDGEMENT

This work is supported by NSF Award DMS-2031849, CCF-1845171 (CAREER), CCF-1934964 (Tripods) and a Sloan Research Fellowship.

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

# A MULTI-LAYER MEAN-FIELD NETWORKS

In this section, we first briefly review existing theories of two-layer mean-field networks, and then introduce our framework for multi-layer mean-field networks.

## A.1 TWO-LAYER NETWORKS AND PERMUTATION INVARIANCE

A two-layer network $f$ of width $m$ can usually be represented by[6]

$$f(\boldsymbol{x}; \boldsymbol{W}, \boldsymbol{a}) = \frac{1}{m} \boldsymbol{a}^\top \sigma(\boldsymbol{W}\boldsymbol{x}) = \frac{1}{m} \sum_{i=1}^{m} a_i \sigma(\boldsymbol{w}_i \cdot \boldsymbol{x}). \tag{7}$$

where $\boldsymbol{W} \in \mathbb{R}^{m \times d}$ is the weight matrix of the hidden layer and $\boldsymbol{a} \in \mathbb{R}^m$ the output weights. Let $\mu$ be the empirical distribution of $\{(a_i, \boldsymbol{w}_i)\}_{i=1}^{m} \subset \mathbb{R}^{d+1}$. Then, we can write

$$f(\boldsymbol{x}; \mu) = \mathop{\mathbb{E}}_{(a,\boldsymbol{w}) \sim \mu} \{ a\sigma(\boldsymbol{w} \cdot \boldsymbol{x}) \}. \tag{8}$$

By allowing $\mu$ to be an arbitrary sufficiently regular distribution over $\mathbb{R}^d$, we obtain a neural network, represented by (8), that can contain infinitely many neurons.

To describe the gradient flow of this infinite-width network, it suffices to assign a vector field to $\mathbb{R}^{d+1}$ that describes how each neuron $(a, \boldsymbol{w}) \in \mathbb{R}^{d+1}$ should move at time $t$. One simple heuristic way to do so is to first compute the gradient in the finite-width case and then replace all summations with expectations as in (8) and treat the gradient as a vector field. We now illustrate the idea under realizable setting and with the MSE loss

$$\mathcal{L} = \frac{1}{2} \mathop{\mathbb{E}}_{\boldsymbol{x}} \left\{ (f_*(\boldsymbol{x}) - f(\boldsymbol{x}))^2 \right\}.$$

The theory can be generalized to much more general settings and can be formally justified using the theory of Wasserstein gradient flow. Readers can refer to, for example, Chizat & Bach (2018) and Mei et al. (2018) for details. For a finite-width network (7), the gradient of $\mathcal{L}$ w.r.t. a neuron $(a_k, \boldsymbol{w}_k)$ is

$$-m\nabla_{a_k}\mathcal{L} = \mathop{\mathbb{E}}_{\boldsymbol{x}} \left\{ (f_*(\boldsymbol{x}) - f(\boldsymbol{x}; \boldsymbol{W}, \boldsymbol{a}))\sigma(\boldsymbol{w}_k \cdot \boldsymbol{x}) \right\},$$

$$-m\nabla_{\boldsymbol{w}_k}\mathcal{L} = \mathop{\mathbb{E}}_{\boldsymbol{x}} \left\{ (f_*(\boldsymbol{x}) - f(\boldsymbol{x}; \boldsymbol{W}, \boldsymbol{a}))a_k\sigma'(\boldsymbol{w}_k \cdot \boldsymbol{x})\boldsymbol{x} \right\}.$$

Replace $f(\boldsymbol{x}; \boldsymbol{W}, \boldsymbol{a})$ with $f(\boldsymbol{x}; \mu)$, treat $(a_k, \boldsymbol{w}_k)$ as a generic neuron, and we obtain a vector field $\tilde{\nabla} : \mathbb{R}^{d+1} \to \mathbb{R}^{d+1}$

$$-\tilde{\nabla}(a, \boldsymbol{w}) := \mathop{\mathbb{E}}_{\boldsymbol{x}} \left\{ (f_*(\boldsymbol{x}) - f(\boldsymbol{x}; \mu)) \begin{bmatrix} \sigma(\boldsymbol{w} \cdot \boldsymbol{x}) \\ a\sigma'(\boldsymbol{w} \cdot \boldsymbol{x})\boldsymbol{x} \end{bmatrix} \right\}.$$

At each time $t$, we update the neurons in $\mu$ according to $-\tilde{\nabla}$.

One of the most important properties of this mean-field formulation is that **it factors out the permutation invariance of neurons**. That is, we can permute $(a_1, \boldsymbol{w}_1), \ldots, (a_m, \boldsymbol{w}_m)$ without changing the output of the network. However, when we treat training as an optimization problem over the space of $(\boldsymbol{a}, \boldsymbol{W})$, i.e., $\mathbb{R}^m \times \mathbb{R}^{m \times d}$, permuting $(a_i, \boldsymbol{w}_i)$ entirely changes $(\boldsymbol{a}, \boldsymbol{W})$. On the other hand, if we describe the network using a distribution $\mu$ over $\mathbb{R}^{d+1}$, then it is automatically permutation invariant. Note that this is not restricted to infinite-width networks. When we choose $\mu$ to be an empirical distribution over finitely many neurons, we recover a finite-width network without breaking the permutation invariance.

---

[6]Here, $\boldsymbol{w}_i \in \mathbb{R}^d$ means the $i$-th row of $\boldsymbol{W}$. Later we will notations $\boldsymbol{v}_i$, $\boldsymbol{a}_i$ to denote $i$-th row or column of the corresponding matrix. Whether it is a row or column can be easily inferred from the dimension. The general rule is that if $\boldsymbol{V} \in \mathbb{R}^{D \times m}$ where $m$ represents the number of neurons, then $\boldsymbol{v}_i \in \mathbb{R}^D$ is $i$-th column, and if $\boldsymbol{W} \in \mathbb{R}^{m \times D}$, then $\boldsymbol{w}_i \in \mathbb{R}^D$ is the $i$-th row.

## A.2 MULTI-LAYER MEAN-FIELD NETWORKS

Unfortunately, the above strategy cannot be directly generalized to multi-layer networks. Consider the three-layer network

$$f(\boldsymbol{x}; \boldsymbol{a}, \boldsymbol{W}_2, \boldsymbol{W}_1) = \frac{1}{m_2} \boldsymbol{a}^\top \sigma\left(\boldsymbol{W}_2 \boldsymbol{h}(\boldsymbol{x}; \boldsymbol{W}_1)\right), \quad \boldsymbol{h}(\boldsymbol{x}; \boldsymbol{W}_1) = \frac{1}{m_1} \sigma(\boldsymbol{W}_1 \boldsymbol{x}),$$

where $\boldsymbol{a} \in \mathbb{R}^{m_2}$, $\boldsymbol{W}_2 \in \mathbb{R}^{m_2 \times m_1}$, $\boldsymbol{W}_1 \in \mathbb{R}^{m_1 \times d}$. One can still write

$$f(\boldsymbol{x}; \boldsymbol{a}, \boldsymbol{W}_2, \boldsymbol{W}_1) = \frac{1}{m_2} \sum_{i=1}^{m_2} a_i \sigma(\boldsymbol{w}_{2,i} \cdot \boldsymbol{h}(\boldsymbol{x}; \boldsymbol{W}_1)) = \underset{(a_i, \boldsymbol{w}_2) \sim \mu_2}{\mathbb{E}} \left\{ a\sigma(\boldsymbol{w}_2 \cdot \boldsymbol{h}(\boldsymbol{x}; \boldsymbol{W}_1)) \right\}.$$

However, now $\mu_2$ is a distribution over $\mathbb{R}^{m_1}$, and if $m_1 \to \infty$, it will become a distribution over $\mathbb{R}^\infty$, which is not readily defined. One way to resolve this issue is to view $\boldsymbol{W}_2$ as a function from $[m_2] \times [m_1]$ to $\mathbb{R}$ and then generalize it to handle the infinite-width case by replacing the index sets $[m_2]$, $[m_1]$ by two general index sets $I_2$, $I_1$ that can potentially be uncountable. For example, we can choose $I_1 = I_2 = \mathbb{R}$. This is the strategy employed by Nguyen & Pham (2020). (See Pham & Nguyen (2021) for a more accessible version of this paper.) The drawback of this formulation is that, with the introduction of index sets, the permutation invariance is no longer factored out. Though with this formulation, it is still possible to obtain global convergence results for infinite-width networks, it become less useful when we want to analyze a finite-width network as it becomes essentially the same as the usual matrix formulation.

We now present a formulation that does factor out the permutation invariance of neurons, and it is built upon composing a sequence of vector-valued two-layer networks. As a first step, we consider a two-layer network with $D$-dimensional outputs:

$$\boldsymbol{f}(\boldsymbol{x}; \boldsymbol{A}, \boldsymbol{W}) = \frac{1}{m} \boldsymbol{A} \sigma(\boldsymbol{W} \boldsymbol{x}), \tag{9}$$

where $\boldsymbol{A} \in \mathbb{R}^{D \times m}$ and $\boldsymbol{W} \in \mathbb{R}^{m \times d}$. For each index $i \in [D]$, we still have

$$f_i(\boldsymbol{x}; \boldsymbol{A}, \boldsymbol{W}) = \frac{1}{m} \sum_{j=1}^{m} a_{i,j} \sigma(\boldsymbol{w}_j \cdot \boldsymbol{x}) = \underset{(a, \boldsymbol{w}) \sim \mu_i}{\mathbb{E}} \left\{ a\sigma(\boldsymbol{w} \cdot \boldsymbol{x}) \right\},$$

where $\mu_i$ is the empirical distribution of $\{(a_{i,j}, \boldsymbol{w}_j)\}_{j \in [m]} \subset \mathbb{R}^{d+1}$. Range over $i$ and we obtain the output vector of this network. For two-layer networks with scalar outputs, in order to obtain its mean-field counterpart, it suffices to allow $\mu$ to take a general distribution over $\mathbb{R} \times \mathbb{R}^d$. This, however, is not the case for networks with vector outputs as the $\boldsymbol{W}$ parts of $\mu_i$ are coupled. Hence, we need to additionally impose the constraint that all $(\mu_i)_{i \in [D]}$ share the same second margin, that is, $\pi_2 \# \mu_i = \mu_{\boldsymbol{W}}$ for some distribution $\mu_{\boldsymbol{W}}$ over $\mathbb{R}^d$ and all $\in [D]$, where $\pi_2 : \mathbb{R} \times \mathbb{R}^d \to \mathbb{R}^d$ is the projection that takes $(a, \boldsymbol{w})$ to $\boldsymbol{w}$. Intuitively, this condition says that they share the same first layer weights $\boldsymbol{W}$. We formalize this idea in the following definition.

**Definition A.1.** *Let $(\mu_i)_{i \in [D]}$ be $D$ sufficiently regular[7] distributions over $\mathbb{R} \times \mathbb{R}^d$. We call $(\mu_i)_{i=1}^{D}$ an **admissible configuration of dimension** $(D, d)$ if there exists a measure $\mu_{\boldsymbol{W}}$ over $\mathbb{R}^d$ such that $\pi_2 \# \mu_i = \mu_{\boldsymbol{W}}$ holds for all $i \in [D]$.*

**Remark**. Note that here, by a neuron, we mean a $(D + d)$-dimensional vector $(a_1, \ldots, a_D, \boldsymbol{w})$. In the finite-width network (9), this corresponds to a row in $\boldsymbol{W}$ and the corresponding column in $\boldsymbol{A}$. This point of view is important when deriving the infinite-width gradient flow since, as in the two-layer case, the vector field at the position of a certain neuron can only depend on the other neurons as a whole. ♣

To complement the discussion, here we consider the problem that, given an admissible infinite-width configuration $(\mu_i)_{i \in [D]}$, how to obtain a finite-width network with $m$ neurons. For a scalar-valued

---

[7]Our focus is on factoring out the permutation invariance and, in this paper, essentially all distributions are empirical distributions over finitely many neurons, with respect to which the integral is just summation and is always well-defined. We leave the work of figuring out specific regularity conditions to future works.

mean-field network characterized by $\mu$, it suffices to generate $m$ samples from $\mu$. For a vector-valued network, the procedure is slightly different. We first sample a weight vector $\boldsymbol{w}$ from the shared margin $\mu_{\boldsymbol{W}}$. Then, for each $i \in [D]$, we generate a real number $a_i$ conditioning on $\boldsymbol{w}$. This gives us a neuron $(a_1, \ldots, a_D, \boldsymbol{w}) \in \mathbb{R}^D \times \mathbb{R}^d$. Repeat this procedure $m$ times and we obtain a finite-width network with $m$ neurons.

We formally define two-layer vector-valued mean-field networks as follows.

**Definition A.2.** *Given an admissible $(\mu_i)_{i \in [D]}$, the two-layer vector-valued network it defines is*

$$\boldsymbol{F}(\boldsymbol{x}; \mu_1, \ldots, \mu_D) = (F_1(\boldsymbol{x}; \mu_1), \ldots, F_D(\boldsymbol{x}; \mu_D)), \tag{10}$$

*where*

$$F_i(\boldsymbol{x}; \mu_i) = \mathop{\mathbb{E}}_{(a, \boldsymbol{w}) \sim \mu_i} \{a \sigma(\boldsymbol{w} \cdot \boldsymbol{x})\}, \quad \forall i \in [D].$$

Now, we are ready to define a multi-layer mean-field network. Basically, a multi-layer mean-field network is a composition of a sequence of two-layer vector-valued networks (10).

**Definition A.3.** *Let $L \geq 1$ be an integer. Let $D^{(1)}, \ldots, D^{(L)}$ be a sequence of positive integers and put $D^{(0)} = d$. For each $l \in [L]$, let $(\mu_i^{(l)})_{i \in [D_l]}$ be an admissible configuration of dimension $(D^{(l)}, D^{(l-1)})$. The $L$-layer mean-field network $\boldsymbol{f}$ defined by the configuration $\Theta := ((\mu_i^{(l)})_{i \in [D_l]})_{l \in [L]}$ is defined recursively as*

$$\begin{aligned} \boldsymbol{f}(\boldsymbol{x}; \Theta) &= \boldsymbol{F}^{(L)}(\boldsymbol{x}; \Theta), \\ \boldsymbol{F}^{(l)}(\boldsymbol{x}; \Theta) &:= \boldsymbol{F}\left(\boldsymbol{F}^{(l-1)}(\boldsymbol{x}; \Theta); \mu_1^{(l)}, \ldots, \mu_{D_l}^{(l)}\right), \quad \forall l \geq 1, \\ \boldsymbol{F}^{(0)}(\boldsymbol{x}; \Theta) &:= \boldsymbol{x}, \end{aligned} \tag{11}$$

*where $\boldsymbol{F}$ is the two-layer mean-field network given by (10).*

**Example** As an example, we consider the case $L = 3$ here. In this case, the finite-width network corresponding to (11) is

$$\boldsymbol{f}(\boldsymbol{x}; \boldsymbol{A}_2, \boldsymbol{W}_2, \boldsymbol{A}_1, \boldsymbol{W}_1) = \frac{1}{m_2} \boldsymbol{A}_2 \sigma\left(\boldsymbol{W}_2 \frac{1}{m_1} \boldsymbol{A}_1 \sigma(\boldsymbol{W}_1 \boldsymbol{x})\right),$$

which is exactly the usual multi-layer network used in practice except the normalizing terms $1/m_2$, $1/m_1$ and an additional matrix $\boldsymbol{A}_1 \in \mathbb{R}^{D_1 \times m_1}$. This matrix compresses an $m_1$ dimensional feature vector to a $D_1$ dimensional one, where $D_1$ is an integer that does not go to $\infty$. It is a reminiscent of the bottleneck structure used in ResNet (He et al. (2016)).

**Remark.** Note that this formulation is indeed invariant under permutation of each layer's neurons. However, it does not factor out all permutation invariance of a deep network. For example, one can permute the columns of $\boldsymbol{W}_1$ and adjusting $\boldsymbol{A}_1, \boldsymbol{W}_2, \boldsymbol{A}_2$ accordingly without changing the output of the network. In some sense, this corresponds to permuting the entires of the hidden feature $\boldsymbol{F}^{(1)}$. We believe it is not necessary or useful to factor out this symmetry since, after all, even in the two-layer case, we do not permute the entries of the inputs $\boldsymbol{x}$. ♣

Finally, we consider the problem of formulating mean-field gradient flow so that it matches the usual gradient flow. The idea is simple: We compute the gradient in the finite-width setting and then replace summations with integrals. For the ease of presentation, we consider a three-layer network and the MSE loss. Again, this framework can be easily generalized to deeper networks and other loss functions. We write

$$\begin{aligned} f(\boldsymbol{x}) &= f(\boldsymbol{x}; \boldsymbol{a}, \boldsymbol{W}_2, \boldsymbol{V}_1, \boldsymbol{W}_1) = \frac{1}{m_2} \boldsymbol{a}^\top \sigma\left(\boldsymbol{W}_2 \boldsymbol{F}(\boldsymbol{x}; \boldsymbol{V}_1, \boldsymbol{W}_1)\right), \\ \boldsymbol{F}(\boldsymbol{x}) &= \boldsymbol{F}(\boldsymbol{x}; \boldsymbol{V}_1, \boldsymbol{W}_1) = \frac{1}{m_1} \boldsymbol{V}_1 \sigma(\boldsymbol{W}_1 \boldsymbol{x}), \\ \mathcal{L} &= \mathcal{L}(\boldsymbol{a}, \boldsymbol{W}_2, \boldsymbol{V}, \boldsymbol{W}_1) = \frac{1}{2} \mathop{\mathbb{E}}_{\boldsymbol{x}} \left\{(f_*(\boldsymbol{x}) - f(\boldsymbol{x}; \boldsymbol{a}, \boldsymbol{W}_2, \boldsymbol{V}, \boldsymbol{W}_1))^2\right\}, \end{aligned}$$

where $\boldsymbol{a} \in \mathbb{R}^{m_2}$, $\boldsymbol{W}_2 \in \mathbb{R}^{m_2 \times D}$, $\boldsymbol{V}_1 \in \mathbb{R}^{D \times m_1}$, $\boldsymbol{W}_1 \in \mathbb{R}^{m_1 \times d}$. We have

$$-m_2 \nabla_{a_i} \mathcal{L} = \mathbb{E}_{\boldsymbol{x}} \left\{ (f_*(\boldsymbol{x}) - f(\boldsymbol{x}))\sigma(\boldsymbol{w}_{2,i} \cdot \boldsymbol{F}(\boldsymbol{x})) \right\}, \qquad \forall i \in [m_2],$$

$$-m_2 \nabla_{\boldsymbol{w}_{2,i}} \mathcal{L} = \mathbb{E}_{\boldsymbol{x}} \left\{ (f_*(\boldsymbol{x}) - f(\boldsymbol{x}))a_i\sigma'(\boldsymbol{w}_{2,i} \cdot \boldsymbol{F}(\boldsymbol{x}))\boldsymbol{F}(\boldsymbol{x}) \right\}, \qquad \forall i \in [m_2],$$

$$-m_1 \nabla_{\boldsymbol{v}_{1,i}} \mathcal{L} = \mathbb{E}_{\boldsymbol{x}} \left\{ (f_*(\boldsymbol{x}) - f(\boldsymbol{x}))\frac{1}{m_2}\sum_{j=1}^{m_2} a_j \sigma'(\boldsymbol{w}_{2,j} \cdot \boldsymbol{F}(\boldsymbol{x}))\boldsymbol{w}_{2,j}\sigma(\boldsymbol{w}_{1,i} \cdot \boldsymbol{x}) \right\}, \qquad \forall i \in [m_1],$$

$$-m_1 \nabla_{\boldsymbol{w}_{1,i}} \mathcal{L} = \mathbb{E}_{\boldsymbol{x}} \left\{ (f_*(\boldsymbol{x}) - f(\boldsymbol{x}))\frac{1}{m_2}\sum_{j=1}^{m_2} a_j \sigma'(\boldsymbol{w}_{2,j} \cdot \boldsymbol{F}(\boldsymbol{x})) \langle \boldsymbol{w}_{2,j}, \boldsymbol{v}_{1,i} \rangle \sigma'(\boldsymbol{w}_{1,i} \cdot \boldsymbol{x})\boldsymbol{x} \right\}, \quad \forall i \in [m_1].$$

Replace summations with integrals and we obtain

$$-\tilde{\nabla}_{(a,\boldsymbol{w}_2)} = \mathbb{E}_{\boldsymbol{x}} \left\{ (f_*(\boldsymbol{x}) - f(\boldsymbol{x})) \begin{bmatrix} \sigma(\boldsymbol{w}_2 \cdot \boldsymbol{F}(\boldsymbol{x})) \\ a\sigma'(\boldsymbol{w}_2 \cdot \boldsymbol{F}(\boldsymbol{x}))\boldsymbol{F}(\boldsymbol{x}) \end{bmatrix} \right\},$$

$$-\tilde{\nabla}_{(\boldsymbol{v}_1,\boldsymbol{w}_1)} = \mathbb{E}_{\boldsymbol{x}} \left\{ (f_*(\boldsymbol{x}) - f(\boldsymbol{x})) \mathbb{E}_{(a,\boldsymbol{w}_2) \sim \mu_2} \left\{ a\sigma'(\boldsymbol{w}_2 \cdot \boldsymbol{F}(\boldsymbol{x})) \begin{bmatrix} \sigma(\boldsymbol{w}_1 \cdot \boldsymbol{x})\boldsymbol{w}_2 \\ \langle \boldsymbol{w}_2, \boldsymbol{v}_1 \rangle \sigma'(\boldsymbol{w}_1 \cdot \boldsymbol{x})\boldsymbol{x} \end{bmatrix} \right\} \right\}. \quad (12)$$

Namely, at each step $t$, we update the second layer neurons $(a, \boldsymbol{w}_2)$ with $-\tilde{\nabla}_{(a,\boldsymbol{w}_2)}$, and first layer neurons $(\boldsymbol{v}_1, \boldsymbol{w}_1)$ with $-\tilde{\nabla}_{(\boldsymbol{v}_1,\boldsymbol{w}_1)}$. Note that, unlike many other multi-layer mean-field frameworks, we do not introduce any notion of paths. The dynamics of each first layer neuron depends on the second layer as a whole as we take expectation over $\mu_2$ in (12). The same is also true for second layer neurons. In some sense, the additional matrix $V_1$ decouples the dynamics of the first and second layer neurons.

## B PRELIMINARIES

### B.1 INDUCTION HYPOTHESIS AND CONTINUITY ARGUMENT

We extensively use the continuos-time version of mathematical induction in our proof, which is also called the continuity argument. We briefly discuss this technique in this subsection and explain some conventions we employ in the writing of the proof. One may refer to, for example, Chapter 1.3 of Tao (2006) for details.

Similar to the discrete-time induction argument, the goal is to maintain a collection of conditions, which we call the Induction Hypothesis, throughout a period of time (cf. Induction Hypothesis C.2 and Induction Hypothesis D.1). There are mainly two types of conditions.

The first type has the form "certain process $A_t$ is bounded by another process $B_t$". In the proof, $A_t$ is usually the error we want to control and $B_t$ an non-decreasing process representing the corresponding upper bound. To maintain this type of condition, it suffices to show that $A_t \leq B_t$ at initialization and $\dot{A}_t \leq \dot{B}_t$ as long as the Induction Hypothesis is true.

For this type of condition, usually we also have an upper bound for $B_t$, say, $B_t \leq B_\infty$. The most rigorous way to maintain these bounds is to argue by contradiction. Let $T$ be the minimum between the time $T_1$ the process ends and the time $T_2$ this bound first get violated. By definition, the Induction Hypothesis holds for any $t \leq T$. Using the Induction Hypothesis, one can then derive an upper bound $T'$ on $T_1$, which then leads to an upper bound on $T$. Then, all we need to show is that $B_{T'}$ is smaller than $B_\infty$ so that $T$ is attained by $T_1$ instead of $T_2$. For the ease of presentation, for this type of conditions, instead of arguing by contradiction explicitly, we will simply show that, provided that the Induction Hypothesis is true over $[0, T_1]$, then $B_{T_1} \leq B_\infty$ holds.

The second type has the form "certain process $C_t$ is bounded some value $D$". Here, $C_t$ is usually some quantity related to the shape of the learner function such as $\bar{w}_2$ and $\alpha$. In order to maintain, say, $C_t \leq D$, we show that when $C_t \in [D - \varepsilon, D]$, we have $\dot{C}_t < 0$. This implies that, as long as $C_t$ is continuous, this implies $C_t$ can never reach $D$.

### B.2 Properties of the Input Distribution

In this subsection, we derive some basic properties of the input distribution that will be useful in later analysis.

The following lemma gives the distribution of $\|\boldsymbol{x}\|$ and its tail bound.

**Lemma B.1.** *Let $\boldsymbol{x} \sim \mathcal{D}$ and let $\|\mathcal{D}\|$ denote the distribution of $\|\boldsymbol{x}\|$. We have*

$$\|\mathcal{D}\| (r) = \frac{d}{r} J_{d/2}^2 (2\pi R_d \beta \alpha \sqrt{d} r) = O\left(\frac{1}{r^2}\right), \quad \forall r > 0.$$

*As a result, we have the tail bound: for all $R > 0$, $\mathbb{P}[\|\boldsymbol{x}\| \geq R] \leq O(1/R)$.*

We now give some regularity conditions on the input distribution that will be used in our proof. Roughly speaking, it shows that the distribution is heavy-tailed and still has large enough mass for $\|\boldsymbol{x}\| \in [0, 1]$

**Lemma B.2** (Regularity conditions on input distribution)**.** *For the input distribution $\mathcal{D}$, we have*

(a) $\mathbb{E}_{\|\boldsymbol{x}\| \leq 0.99} \|\boldsymbol{x}\| = \Theta(1)$.

(b) $\mathbb{E}_{\boldsymbol{x} \sim \mathcal{D}} f_*(\boldsymbol{x}) = \Omega(1)$.

(c) $\mathbb{E}_{\|\boldsymbol{x}\| \leq \Omega(d)} \|\boldsymbol{x}\| \geq \Theta(\log d)$ and $\mathbb{E}_{\|\boldsymbol{x}\| \leq \mathrm{poly}(d)} \|\boldsymbol{x}\| \leq \Theta(\log(d))$.

*Proof of Lemma B.1.* Recall that the input distribution of $\boldsymbol{x}$ is

$$\left(\beta \alpha \sqrt{d}\right)^d \varphi^2(\beta \alpha \sqrt{d} \boldsymbol{x}),$$

where $\alpha, \beta > 0$ are the universal constants from Safran et al. (2019) (cf. the proof of Theorem 5),

$$\varphi(\boldsymbol{x}) = \left(\frac{R_d}{\|\boldsymbol{x}\|}\right)^{d/2} J_{d/2}(2\pi R_d \|\boldsymbol{x}\|), \quad \boldsymbol{x} \in \mathbb{R}^d,$$

$R_d = \frac{1}{\sqrt{\pi}} (\Gamma(d/2 + 1))^{1/d} = \Theta(\sqrt{d})$ (Lemma 5 in Eldan & Shamir (2016)) and $J_\nu$ is the Bessel function of the first kind of order. Note that since $\varphi$ only depends on $\|\boldsymbol{x}\|$, we can abuse the notation to use $\varphi(r)$ to denote $\varphi(\boldsymbol{x})$ with $\|\boldsymbol{x}\| = r$.

For any test function $g : \mathbb{R} \mapsto \mathbb{R}$, we have

$$\mathbb{E}_{x \sim \mathcal{D}}[g(\|\boldsymbol{x}\|)] = \int_{\mathbb{R}^d} g(\|\boldsymbol{x}\|) \left(\beta \alpha \sqrt{d}\right)^d \varphi^2(\beta \alpha \sqrt{d} \boldsymbol{x}) \mathrm{d}\boldsymbol{x}$$

$$= \left(\beta \alpha \sqrt{d}\right)^d S_{d-1} \int_0^\infty g(r) \varphi^2(\beta \alpha \sqrt{d} r) r^{d-1} \mathrm{d}r,$$

where $S_{d-1} = 2\pi^{d/2}/\Gamma(d/2)$ is the surface of unit ball $\mathbb{S}^{d-1}$. Therefore, we have the density of $\|\boldsymbol{x}\|$ with $\|\boldsymbol{x}\| = r$ is

$$\left(\beta \alpha \sqrt{d}\right)^d S_{d-1} \varphi^2(\beta \alpha \sqrt{d} r) r^{d-1} = \frac{2\pi^{d/2} \left(\beta \alpha \sqrt{d}\right)^d}{\Gamma(d/2)} \frac{R_d^d}{\left(\beta \alpha \sqrt{d} r\right)^d} J_{d/2}^2(2\pi R_d \beta \alpha \sqrt{d} r) r^{d-1}$$

$$= \frac{d}{r} J_{d/2}^2(2\pi R_d \beta \alpha \sqrt{d} r)$$

$$= O\left(\frac{1}{r^2}\right),$$

where we use the fact that $J_\nu(z) = O(1/\sqrt{z})$ (Krasikov (2006)). Then, it is easy to see that $\mathbb{P}(\|\boldsymbol{x}\| \geq R) = O(1/R)$.

$\square$

*Proof of Lemma B.2.*

(a) It is easy to see the upper bound

$$\mathbb{E}_{\|\boldsymbol{x}\|\leq 0.99} \|\boldsymbol{x}\| \leq 0.99.$$

For lower bound, note that $\mathbb{E}_{\|\boldsymbol{x}\|\leq 0.99} \|\boldsymbol{x}\| \geq 0.1\,\mathbb{P}(0.1 \leq \|\boldsymbol{x}\| \leq 0.99)$. Hence, it suffices to lower bound $\mathbb{P}(0.1 \leq \|\boldsymbol{x}\| \leq 0.99)$. We have

$$
\begin{aligned}
\mathbb{P}(0.1 \leq \|\boldsymbol{x}\| \leq 0.99) &= \int_{0.1}^{0.99} \frac{d}{r} J_{d/2}^2(2\pi R_d \beta \alpha \sqrt{d} r)\mathrm{d}r \\
&\geq \Omega(1) \int_{0.2\pi R_d \beta \alpha \sqrt{d}}^{1.98\pi R_d \beta \alpha \sqrt{d}} J_{d/2}^2(r)\mathrm{d}r \\
&= \Omega(1),
\end{aligned}
$$

where in the last line we use Lemma 23 in Eldan & Shamir (2016). This implies that $\mathbb{E}_{\|\boldsymbol{x}\|\leq 0.99} \|\boldsymbol{x}\| = \Omega(1)$. Together with the upper bound, we have $\mathbb{E}_{\|\boldsymbol{x}\|\leq 0.99} \|\boldsymbol{x}\| = \Theta(1)$.

(b) We have

$$
\begin{aligned}
\mathbb{E}_{\boldsymbol{x}\sim\mathcal{D}} f_*(\boldsymbol{x}) &= \mathbb{E}_{\|\boldsymbol{x}\|\leq 1}[1 - \|\boldsymbol{x}\|] \geq \mathbb{E}_{\|\boldsymbol{x}\|\leq 0.99}[1 - \|\boldsymbol{x}\|] \\
&\geq 0.01\,\mathbb{P}(\|\boldsymbol{x}\| \leq 0.99) \geq 0.01\,\mathbb{P}(0.1 \leq \|\boldsymbol{x}\| \leq 0.99) = \Omega(1),
\end{aligned}
$$

where the last inequality we use the calculation in (a).

(c) The upper bound follows directly from the tail bound $\|\mathcal{D}\|\,(r) \leq O(1/r^2)$. For the lower bound, recall the density of $\|\boldsymbol{x}\|$ when $\|\boldsymbol{x}\| = r$ is $\frac{d}{r} J_{d/2}^2(2\pi R_d \beta \alpha \sqrt{d} r)$. For notational simplicity, put $R_\mathcal{D} = \Theta(d)$. We have

$$
\begin{aligned}
\mathbb{E}_{\|\boldsymbol{x}\|\leq R_\mathcal{D}} \|\boldsymbol{x}\| &= \int_0^{R_\mathcal{D}} d J_{d/2}^2(2\pi R_d \beta \alpha \sqrt{d} r)\mathrm{d}r \\
&= \frac{d}{2\pi R_d \beta \alpha \sqrt{d}} \int_0^{2\pi R_d R_\mathcal{D} \beta \alpha \sqrt{d}} J_{d/2}^2(r)\mathrm{d}r \\
&\geq \Omega(1) \int_{cd}^{cd^2} J_{d/2}^2(r)\mathrm{d}r,
\end{aligned}
$$

where $c$ is a large enough constant.

To lower bound $\mathbb{E}\|\boldsymbol{x}\|$, it suffices to lower bound $\int_{cd}^{cd^2} J_{d/2}^2(r)\mathrm{d}r$. In the following, we will lower bound it by following a similar calculation in Lemma 23 in Eldan & Shamir (2016). From the proof of Lemma 23 in Eldan & Shamir (2016), we have for $x \geq d \geq 2$

$$J_{d/2}^2(x) \geq \frac{2}{\pi x} \cos^2\left(-\frac{(d+1)\pi}{4} + f_{d,x}x\right) - 3x^{-2},$$

where $f_{d,x}$ is a quantity that depends on $d$ and $x$, and satisfies $1.3 \geq f_{d,x} \geq 0.85$.

Then, we have

$$
\begin{aligned}
\int_{cd}^{cd^2} J_{d/2}^2(x)\mathrm{d}x &\geq \int_{cd}^{cd^2} \frac{2}{\pi x} \cos^2\left(-\frac{(d+1)\pi}{4} + f_{d,x}x\right)\mathrm{d}x - \int_{cd}^{cd^2} 3x^{-2}\mathrm{d}x \\
&= \frac{2}{\pi} \int_{cd}^{cd^2} \frac{1}{x} \cos^2\left(-\frac{(d+1)\pi}{4} + f_{d,x}x\right)\mathrm{d}x - \frac{3(d-1)}{cd^2}
\end{aligned}
$$

Note that in the proof of Lemma 23 in Eldan & Shamir (2016), it is shown that

$$\frac{\partial}{\partial x}(f_{d,x}x) = \sqrt{1 - \frac{d^2-1}{4x^2}} \leq 1.$$

Then, since $1.3 \geq f_{d,x} \geq 0.85$ we have

$$
\frac{2}{\pi} \int_{cd}^{cd^2} \frac{1}{x} \cos^2\left(-\frac{(d+1)\pi}{4} + f_{d,x}x\right) \mathrm{d}x
$$

$$
\geq \frac{2}{\pi} \int_{cd}^{cd^2} \frac{0.85}{f_{d,x}x} \cos^2\left(-\frac{(d+1)\pi}{4} + f_{d,x}x\right) \frac{\partial}{\partial x}(f_{d,x}x)\mathrm{d}x
$$

$$
= \frac{2}{\pi} \int_{f_{d,cd}cd}^{f_{d,cd^2}cd^2} \frac{0.85}{z} \cos^2\left(-\frac{(d+1)\pi}{4} + z\right) \mathrm{d}z
$$

$$
\geq \frac{1.7}{\pi} \int_{1.3cd}^{0.85cd^2} \frac{1}{z} \cos^2\left(-\frac{(d+1)\pi}{4} + z\right) \mathrm{d}z.
$$

Then, using integration by parts and the fact that $\cos^2(z-(d+1)\pi/4) = \frac{\partial}{\partial z}(z/2+\sin(2z-(d+1)\pi/2)/4)$, we have

$$
\int_{1.3cd}^{0.85cd^2} \frac{1}{z} \cos^2\left(-\frac{(d+1)\pi}{4} + z\right) \mathrm{d}z
$$

$$
= \frac{(\frac{z}{2} + \frac{1}{4}\sin(2z - \frac{(d+1)\pi}{2}))}{z}\Bigg|_{1.3cd}^{0.85cd^2} + \int_{1.3cd}^{0.85cd^2} \frac{(\frac{z}{2} + \frac{1}{4}\sin(2z - \frac{(d+1)\pi}{2}))}{z^2}\mathrm{d}z
$$

$$
\geq -\frac{1}{4}\left(\frac{1}{0.85cd^2} + \frac{1}{1.3cd}\right) + \int_{1.3cd}^{0.85cd^2} \frac{1}{4z}\mathrm{d}z
$$

$$
= -\frac{1}{4}\left(\frac{1}{0.85cd^2} + \frac{1}{1.3cd}\right) + \frac{1}{4}\ln\frac{0.85cd^2}{1.3cd} = \Omega(\log d).
$$

Therefore, we have

$$
\int_{cd}^{cd^2} J_{d/2}^2(x)\mathrm{d}x = \Omega(\log d),
$$

which implies $\mathbb{E}_{\|\boldsymbol{x}\|\leq\Theta(d)}\|\boldsymbol{x}\| = \Omega(\log d)$.

$\square$

## B.3 Properties of Spherically Symmetric Functions and Distributions

In this subsection, we give some useful proprieties of spherically symmetric functions and distributions. These will be useful tools in our later analysis. Basically, these lemmas allow us to disentangle input $\boldsymbol{x}$ and neuron $\boldsymbol{v}$ when considering integration against spherically symmetric function.

**Lemma 3.1.** *Let $\mu$ be a spherically symmetric distribution. We have*

$$
\mathbb{E}_{\boldsymbol{w}\sim\mu}\|\boldsymbol{w}\|\sigma(\boldsymbol{w}\cdot\boldsymbol{x}) = C_\Gamma\frac{\mathbb{E}_{\boldsymbol{w}\sim\mu}\|\boldsymbol{w}\|^2}{\sqrt{d}}\|\boldsymbol{x}\| \quad \text{where} \quad C_\Gamma := \frac{\Gamma(d/2)\sqrt{d}}{2\sqrt{\pi}\Gamma((d+1)/2)}.
$$

*Note that, as $d \to \infty$, we have $C_\Gamma \to 1/\sqrt{2\pi}$, so $C_\Gamma$ is universally bounded for all $d$.*

**Lemma B.3.** *For any spherically symmetric $g : \mathbb{R}^d \to \mathbb{R}$ and $\boldsymbol{v} \in \mathbb{R}^d$, we have*

$$
\mathbb{E}_{\boldsymbol{x}}\{g(\boldsymbol{x})\sigma(\boldsymbol{v}\cdot\boldsymbol{x})\} = \frac{C_\Gamma}{\sqrt{d}}\mathbb{E}_{\boldsymbol{x}}\{g(\boldsymbol{x})\|\boldsymbol{x}\|\}\|\boldsymbol{v}\|.
$$

**Corollary B.4.** *Let $g : \mathbb{R}^d \to \mathbb{R}$ be a spherically symmetric function. We have*

$$
\mathbb{E}_{\boldsymbol{x}}\{g(\boldsymbol{x})F(\boldsymbol{x})\} = \alpha\mathbb{E}_{\boldsymbol{x}}\{g(\boldsymbol{x})\|\boldsymbol{x}\|\}.
$$

**Lemma B.5.** *Let $g : \mathbb{R}^d \to \mathbb{R}$ be a spherically symmetric function. Then, for any $\boldsymbol{v} \in \mathbb{R}^d$, we have*

$$
\mathbb{E}_{\boldsymbol{x}\sim\mathcal{D}}\{g(\boldsymbol{x})\sigma'(\boldsymbol{v}\cdot\boldsymbol{x})\boldsymbol{x}\} = \mathbb{E}_{\boldsymbol{x}\sim\mathcal{D}}\{g(\boldsymbol{x})\|\boldsymbol{x}\|\}\frac{C_\Gamma}{\sqrt{d}}\bar{\boldsymbol{v}}.
$$

*Proof of Lemma 3.1.* For simplicity, put $g(\boldsymbol{x}) = \mathbb{E}_{\boldsymbol{w} \sim \mu} \|\boldsymbol{w}\| \, \sigma(\boldsymbol{w} \cdot \boldsymbol{x})$. Since $\sigma$ is 1-homogenous and $\mu$ is spherically symmetric, we have

$$
\begin{aligned}
g(\boldsymbol{x}) &= \int_{\mathbb{R}^d} \|\boldsymbol{w}\|^2 \, \sigma(\bar{\boldsymbol{w}} \cdot \boldsymbol{x}) \mu(\boldsymbol{w}) \, \mathrm{d}\boldsymbol{w} \\
&= \int_0^\infty \int_{\mathbb{S}^{d-1}} r^2 \sigma(\bar{\boldsymbol{w}} \cdot \boldsymbol{x}) \mu(r\bar{\boldsymbol{w}}) r^{d-1} \, \mathrm{d}\sigma^{d-1}(\bar{\boldsymbol{w}}) \mathrm{d}r \\
&= \int_0^\infty r^{d+1} \mu(r) \, \mathrm{d}r \int_{\mathbb{S}^{d-1}} \sigma(\bar{\boldsymbol{w}} \cdot \boldsymbol{x}) \, \mathrm{d}\sigma^{d-1}(\bar{\boldsymbol{w}}).
\end{aligned}
$$

For the first term, note that[8]

$$
\int_{\mathbb{R}^d} \|\boldsymbol{w}\|^2 \, \mu(\boldsymbol{w}) \, \mathrm{d}\boldsymbol{w} = \int_0^\infty \int_{\mathbb{S}^{d-1}} r^2 \mu(r\bar{\boldsymbol{w}}) \, \mathrm{d}\sigma^{d-1}(\bar{\boldsymbol{w}}) \mathrm{d}r = \frac{2\pi^{d/2}}{\Gamma(d/2)} \int_0^\infty r^{d+1} \mu(r) \, \mathrm{d}r.
$$

Hence,

$$
\int_0^\infty r^{d+1} \mu(r) \, \mathrm{d}r = \frac{\Gamma(d/2)}{2\pi^{d/2}} \int_{\mathbb{R}^d} \|\boldsymbol{w}\|^2 \, \mu(\boldsymbol{w}) \, \mathrm{d}\boldsymbol{w} = \frac{\Gamma(d/2)}{2\pi^{d/2}} \mathbb{E}_{\boldsymbol{w} \sim \mu} \|\boldsymbol{w}\|^2 .
$$

Then we compute the second term as follows. Since it is also spherically symmetric, we have

$$
\int_{\mathbb{S}^{d-1}} \sigma(\bar{\boldsymbol{w}} \cdot \boldsymbol{x}) \, \mathrm{d}\sigma^{d-1}(\bar{\boldsymbol{w}}) = \|\boldsymbol{x}\| \int_{\mathbb{S}^{d-1}} \sigma(\bar{w}_1) \, \mathrm{d}\sigma^{d-1}(\bar{\boldsymbol{w}}) = \frac{\|\boldsymbol{x}\|}{2} \int_{\mathbb{S}^{d-1}} |\bar{w}_1| \, \mathrm{d}\sigma^{d-1}(\bar{\boldsymbol{w}}).
$$

Define $I = \int_{\mathbb{R}^d} |w_1| e^{-\|\boldsymbol{w}\|^2} \, \mathrm{d}\boldsymbol{w}$. We have

$$
I = \int_{\mathbb{R}^d} |w_1| \prod_{i=1}^d e^{-w_i^2} \, \mathrm{d}\boldsymbol{w} = \left( \int_{-\infty}^\infty |w_1| e^{-w_1^2} \, \mathrm{d}w_1 \right) \prod_{i=2}^d \int_{-\infty}^\infty e^{-w_i^2} \, \mathrm{d}w_i = \pi^{(d-1)/2}.
$$

We also have

$$
\begin{aligned}
I = \int_{\mathbb{S}^{d-1}} \int_0^\infty r|\bar{w}_1| e^{-r^2} r^{d-1} \, \mathrm{d}r \mathrm{d}\sigma^{d-1}(\bar{\boldsymbol{w}}) &= \int_0^\infty e^{-r^2} r^d \, \mathrm{d}r \int_{\mathbb{S}^{d-1}} |\bar{w}_1| \, \mathrm{d}\sigma^{d-1}(\bar{\boldsymbol{w}}) \\
&= \frac{\Gamma((d+1)/2)}{2} \int_{\mathbb{S}^{d-1}} |\bar{w}_1| \, \mathrm{d}\sigma^{d-1}(\bar{\boldsymbol{w}}).
\end{aligned}
$$

Therefore,

$$
\int_{\mathbb{S}^{d-1}} \sigma(\bar{\boldsymbol{w}} \cdot \boldsymbol{x}) \, \mathrm{d}\sigma^{d-1}(\bar{\boldsymbol{w}}) = \frac{\|\boldsymbol{x}\|}{2} \int_{\mathbb{S}^{d-1}} |\bar{w}_1| \, \mathrm{d}\sigma^{d-1}(\bar{\boldsymbol{w}}) = \frac{\pi^{(d-1)/2}}{\Gamma((d+1)/2)} \|\boldsymbol{x}\| . \tag{13}
$$

Thus,

$$
g(\boldsymbol{x}) = \frac{\Gamma(d/2)}{2\pi^{d/2}} \mathbb{E}_{\boldsymbol{w} \sim \mu} \|\boldsymbol{w}\|^2 \frac{\pi^{(d-1)/2}}{\Gamma((d+1)/2)} \|\boldsymbol{x}\| = C_\Gamma \frac{\mathbb{E}_{\boldsymbol{w} \sim \mu} \|\boldsymbol{w}\|^2}{\sqrt{d}} \|\boldsymbol{x}\| .
$$

$\square$

*Proof of Lemma B.3.* We compute

$$
\begin{aligned}
\mathbb{E}_{\boldsymbol{x} \sim \mathcal{D}} \{ g(\boldsymbol{x}) \sigma(\boldsymbol{v} \cdot \boldsymbol{x}) \} &= \int_{\mathbb{R}^d} g(\boldsymbol{x}) \sigma(\boldsymbol{v} \cdot \boldsymbol{x}) \mathcal{D}(\boldsymbol{x}) \, \mathrm{d}\boldsymbol{x} \\
&= \int_0^\infty \int_{\mathbb{S}^{d-1}} g(r\bar{\boldsymbol{x}}) \sigma(\boldsymbol{v} \cdot (r\bar{\boldsymbol{x}})) \mathcal{D}(r\bar{\boldsymbol{x}}) r^{d-1} \, \mathrm{d}\sigma^{d-1}(\bar{\boldsymbol{x}}) \mathrm{d}r \\
&= \int_0^\infty \int_{\mathbb{S}^{d-1}} g(r) \sigma(\boldsymbol{v} \cdot \bar{\boldsymbol{x}}) \mathcal{D}(r) r^d \, \mathrm{d}\sigma^{d-1}(\bar{\boldsymbol{x}}) \mathrm{d}r \\
&= \int_0^\infty g(r) \mathcal{D}(r) r^d \, \mathrm{d}r \int_{\mathbb{S}^{d-1}} \sigma(\boldsymbol{v} \cdot \bar{\boldsymbol{x}}) \, \mathrm{d}\sigma^{d-1}(\bar{\boldsymbol{x}}) \\
&= \int_0^\infty g(r) \mathcal{D}(r) r^d \, \mathrm{d}r \frac{\pi^{(d-1)/2}}{\Gamma((d+1)/2)} \|\boldsymbol{v}\| ,
\end{aligned}
$$

---

[8]Recall the surface area of the $d$-dimensional unit sphere is $\int \mathrm{d}\sigma^{d-1} = \frac{2\pi^{d/2}}{\Gamma(d/2)}$.

where the last line comes from (13). (Note the integral is taken w.r.t. $\bar{\boldsymbol{x}}$ instead of $\bar{\boldsymbol{w}}$ here.) For the first term, note that

$$
\begin{aligned}
\mathop{\mathbb{E}}_{\boldsymbol{x}\sim\mathcal{D}}\left\{g(\boldsymbol{x})\left\|\boldsymbol{x}\right\|\right\} &= \int_{\mathbb{R}^d} g(\boldsymbol{x})\left\|\boldsymbol{x}\right\|\mathcal{D}(\boldsymbol{x})\,\mathrm{d}\boldsymbol{x} \\
&= \int_0^\infty \int_{\mathbb{S}^{d-1}} g(r)\mathcal{D}(\boldsymbol{x})r^d\,\mathrm{d}\sigma^{d-1}(\bar{\boldsymbol{x}})\mathrm{d}r \\
&= \int_0^\infty \int_{\mathbb{S}^{d-1}} g(r)\mathcal{D}(\boldsymbol{x})r^d\,\mathrm{d}\sigma^{d-1}(\bar{\boldsymbol{x}})\mathrm{d}r \\
&= \frac{2\pi^{d/2}}{\Gamma(d/2)}\int_0^\infty g(r)\mathcal{D}(\boldsymbol{x})r^d\,\mathrm{d}r.
\end{aligned}
$$

Thus,

$$
\begin{aligned}
\mathop{\mathbb{E}}_{\boldsymbol{x}\sim\mathcal{D}}\left\{g(\boldsymbol{x})\sigma(\boldsymbol{v}\cdot\boldsymbol{x})\right\} &= \mathop{\mathbb{E}}_{\boldsymbol{x}\sim\mathcal{D}}\left\{g(\boldsymbol{x})\left\|\boldsymbol{x}\right\|\right\}\left(\frac{2\pi^{d/2}}{\Gamma(d/2)}\right)^{-1}\frac{\pi^{(d-1)/2}}{\Gamma((d+1)/2)}\left\|\boldsymbol{w}\right\| \\
&= \mathop{\mathbb{E}}_{\boldsymbol{x}\sim\mathcal{D}}\left\{g(\boldsymbol{x})\left\|\boldsymbol{x}\right\|\right\}\frac{C_\Gamma}{\sqrt{d}}\left\|\boldsymbol{v}\right\|.
\end{aligned}
$$

$\square$

*Proof of Corollary B.4.* By the previous Lemma, we have

$$
\begin{aligned}
\mathop{\mathbb{E}}_{\boldsymbol{x}}\left\{g(\boldsymbol{x})F(\boldsymbol{x})\right\} &= \mathop{\mathbb{E}}_{\boldsymbol{x}}\left\{g(\boldsymbol{x})\mathop{\mathbb{E}}_{\boldsymbol{w}\sim\mu_1}\left\{\left\|\boldsymbol{w}\right\|\sigma(\boldsymbol{w}\cdot\boldsymbol{x})\right\}\right\} \\
&= \mathop{\mathbb{E}}_{\boldsymbol{w}\sim\mu_1}\left\{\left\|\boldsymbol{w}\right\|\mathop{\mathbb{E}}_{\boldsymbol{x}}\left\{g(\boldsymbol{x})\sigma(\boldsymbol{w}\cdot\boldsymbol{x})\right\}\right\} \\
&= \mathop{\mathbb{E}}_{\boldsymbol{w}\sim\mu_1}\left\{\left\|\boldsymbol{w}\right\|^2\frac{C_\Gamma}{\sqrt{d}}\mathop{\mathbb{E}}_{\boldsymbol{x}}\left\{g(\boldsymbol{x})\left\|\boldsymbol{x}\right\|\right\}\right\} = \alpha\mathop{\mathbb{E}}_{\boldsymbol{x}}\left\{g(\boldsymbol{x})\left\|\boldsymbol{x}\right\|\right\}.
\end{aligned}
$$

$\square$

*Proof of Lemma B.5.* Define $\boldsymbol{R} = \bar{\boldsymbol{v}}\bar{\boldsymbol{v}}^\top - (\boldsymbol{I}_d - \bar{\boldsymbol{v}}\bar{\boldsymbol{v}}^\top) = 2\bar{\boldsymbol{v}}\bar{\boldsymbol{v}}^\top - \boldsymbol{I}_d$. That is, $\boldsymbol{R}$ is the reflection matrix associated with $\bar{\boldsymbol{v}}$. Since $\mathcal{D}$ is spherically symmetric, we have $\boldsymbol{R}\#\mathcal{D} = \mathcal{D}$. For the same reason, $g\circ\boldsymbol{R} = g$. Moreover, by construction, $\boldsymbol{R}\boldsymbol{v} = \boldsymbol{v}$. Hence,

$$
\begin{aligned}
\mathop{\mathbb{E}}_{\boldsymbol{x}\sim\mathcal{D}}\left\{g(\boldsymbol{x})\sigma'(\boldsymbol{v}\cdot\boldsymbol{x})\boldsymbol{x}\right\} &= \frac{1}{2}\left(\mathop{\mathbb{E}}_{\boldsymbol{x}\sim\mathcal{D}}\left\{g(\boldsymbol{x})\sigma'(\boldsymbol{v}\cdot\boldsymbol{x})\boldsymbol{x}\right\} + \mathop{\mathbb{E}}_{\boldsymbol{x}\sim\boldsymbol{R}\#\mathcal{D}}\left\{g(\boldsymbol{x})\sigma'(\boldsymbol{v}\cdot\boldsymbol{x})\boldsymbol{x}\right\}\right) \\
&= \frac{1}{2}\left(\mathop{\mathbb{E}}_{\boldsymbol{x}\sim\mathcal{D}}\left\{g(\boldsymbol{x})\sigma'(\boldsymbol{v}\cdot\boldsymbol{x})\boldsymbol{x} + g(\boldsymbol{R}\boldsymbol{x})\sigma'(\boldsymbol{v}\cdot\boldsymbol{R}\boldsymbol{x})\boldsymbol{R}\boldsymbol{x}\right\}\right) \\
&= \frac{1}{2}\left(\mathop{\mathbb{E}}_{\boldsymbol{x}\sim\mathcal{D}}\left\{g(\boldsymbol{x})\sigma'(\boldsymbol{v}\cdot\boldsymbol{x})\boldsymbol{x} + g(\boldsymbol{R}\boldsymbol{x})\sigma'(\boldsymbol{R}\boldsymbol{v}\cdot\boldsymbol{x})\boldsymbol{R}\boldsymbol{x}\right\}\right) \\
&= \frac{1}{2}\left(\mathop{\mathbb{E}}_{\boldsymbol{x}\sim\mathcal{D}}\left\{g(\boldsymbol{x})\sigma'(\boldsymbol{v}\cdot\boldsymbol{x})\left(\boldsymbol{x} + \boldsymbol{R}\boldsymbol{x}\right)\right\}\right).
\end{aligned}
$$

Note that $\boldsymbol{x} + \boldsymbol{R}\boldsymbol{x} = 2\bar{\boldsymbol{v}}\bar{\boldsymbol{v}}^\top\boldsymbol{x} = 2\left\langle\bar{\boldsymbol{v}},\boldsymbol{x}\right\rangle\bar{\boldsymbol{v}}$. Hence,

$$
\mathop{\mathbb{E}}_{\boldsymbol{x}\sim\mathcal{D}}\left\{g(\boldsymbol{x})\sigma'(\boldsymbol{v}\cdot\boldsymbol{x})\boldsymbol{x}\right\} = \mathop{\mathbb{E}}_{\boldsymbol{x}\sim\mathcal{D}}\left\{g(\boldsymbol{x})\sigma(\bar{\boldsymbol{v}}\cdot\boldsymbol{x})\right\}\bar{\boldsymbol{v}} = \mathop{\mathbb{E}}_{\boldsymbol{x}\sim\mathcal{D}}\left\{g(\boldsymbol{x})\left\|\boldsymbol{x}\right\|\right\}\frac{C_\Gamma}{\sqrt{d}}\bar{\boldsymbol{v}},
$$

where the second identity comes from Lemma B.3. $\square$

### B.4 THE INFINITE-WIDTH NETWORK REMAINS SPHERICALLY SYMMETRIC

In this subsection, we show that the infinite-width network remains spherically symmetric throughout the whole process. Clear that $\mu_1$ is spherically symmetric at initialization. Now, assume that it is spherically symmetric at time $t$. We claim that $\boldsymbol{v}_1$ does not move tangentially, and its radial speed does not depend on its direction $\bar{\boldsymbol{v}}_1$. That is, $\dot{\boldsymbol{v}}_1 = h(\left\|\boldsymbol{v}_1\right\|)\bar{\boldsymbol{v}}_1$ for some function $h$.

By our induction hypothesis, $S$ is also spherically symmetric at time $t$. Let $\boldsymbol{T} := 2\bar{\boldsymbol{v}}_1\bar{\boldsymbol{v}}_1^\top - \boldsymbol{I}_d$ be the reflection w.r.t. $\boldsymbol{v}_1$. Clear that $\boldsymbol{T}\boldsymbol{v}_1 = \boldsymbol{v}_1$. Moreover, it does not change the norm and, as a result, $S(\boldsymbol{T}\boldsymbol{x}) = S(\boldsymbol{x})$, $\boldsymbol{T}\#\mathcal{D} = \mathcal{D}$ and $\Pi \circ \boldsymbol{T} = \boldsymbol{T} \circ \Pi$. Hence, we have

$$
\begin{aligned}
\dot{\boldsymbol{v}}_1 &= \mathop{\mathbb{E}}_{\boldsymbol{x}\sim\mathcal{D}} \left\{ \Pi_{R_{\boldsymbol{v}_1}} \left[ S(\boldsymbol{x}) \left( \bar{\boldsymbol{v}}_1 \sigma(\boldsymbol{v}_1 \cdot \boldsymbol{x}) + \|\boldsymbol{v}_1\| \sigma'(\boldsymbol{v}_1 \cdot \boldsymbol{x})\boldsymbol{x} \right) \right] \right\} \\
&= \frac{1}{2} \mathop{\mathbb{E}}_{\boldsymbol{x}\sim\mathcal{D}} \left\{ \Pi_{R_{\boldsymbol{v}_1}} \left[ S(\boldsymbol{x}) \left( \bar{\boldsymbol{v}}_1 \sigma(\boldsymbol{v}_1 \cdot \boldsymbol{x}) + \|\boldsymbol{v}_1\| \sigma'(\boldsymbol{v}_1 \cdot \boldsymbol{x})\boldsymbol{x} \right) \right] \right\} \\
&\quad + \frac{1}{2} \mathop{\mathbb{E}}_{\boldsymbol{x}\sim\boldsymbol{T}\#\mathcal{D}} \left\{ \Pi_{R_{\boldsymbol{v}_1}} \left[ S(\boldsymbol{x}) \left( \bar{\boldsymbol{v}}_1 \sigma(\boldsymbol{v}_1 \cdot \boldsymbol{x}) + \|\boldsymbol{v}_1\| \sigma'(\boldsymbol{v}_1 \cdot \boldsymbol{x})\boldsymbol{x} \right) \right] \right\} .
\end{aligned}
$$

For the second term, we have

$$
\begin{aligned}
&\mathop{\mathbb{E}}_{\boldsymbol{x}\sim\boldsymbol{T}\#\mathcal{D}} \left\{ \Pi_{R_{\boldsymbol{v}_1}} \left[ S(\boldsymbol{x}) \left( \bar{\boldsymbol{v}}_1 \sigma(\boldsymbol{v}_1 \cdot \boldsymbol{x}) + \|\boldsymbol{v}_1\| \sigma'(\boldsymbol{v}_1 \cdot \boldsymbol{x})\boldsymbol{x} \right) \right] \right\} \\
&= \mathop{\mathbb{E}}_{\boldsymbol{x}\sim\mathcal{D}} \left\{ \Pi_{R_{\boldsymbol{v}_1}} \left[ S(\boldsymbol{x}) \left( \bar{\boldsymbol{v}}_1 \sigma(\boldsymbol{v}_1 \cdot \boldsymbol{T}\boldsymbol{x}) + \|\boldsymbol{v}_1\| \sigma'(\boldsymbol{v}_1 \cdot \boldsymbol{T}\boldsymbol{x})\boldsymbol{T}\boldsymbol{x} \right) \right] \right\} \\
&= \mathop{\mathbb{E}}_{\boldsymbol{x}\sim\mathcal{D}} \left\{ \Pi_{R_{\boldsymbol{v}_1}} \left[ S(\boldsymbol{x}) \left( \bar{\boldsymbol{v}}_1 \sigma(\boldsymbol{v}_1 \cdot \boldsymbol{x}) + \|\boldsymbol{v}_1\| \sigma'(\boldsymbol{v}_1 \cdot \boldsymbol{x})\boldsymbol{T}\boldsymbol{x} \right) \right] \right\} \\
&= \mathop{\mathbb{E}}_{\boldsymbol{x}\sim\mathcal{D}} \left\{ \Pi_{R_{\boldsymbol{v}_1}} \left[ S(\boldsymbol{x})\boldsymbol{T} \left( \sigma(\boldsymbol{v}_1 \cdot \boldsymbol{x})\bar{\boldsymbol{v}}_1 + \|\boldsymbol{v}_1\| \sigma'(\boldsymbol{v}_1 \cdot \boldsymbol{x})\boldsymbol{x} \right) \right] \right\} \\
&= \mathop{\mathbb{E}}_{\boldsymbol{x}\sim\mathcal{D}} \left\{ \boldsymbol{T}\,\Pi_{R_{\boldsymbol{v}_1}} \left[ S(\boldsymbol{x}) \left( \sigma(\boldsymbol{v}_1 \cdot \boldsymbol{x})\bar{\boldsymbol{v}}_1 + \|\boldsymbol{v}_1\| \sigma'(\boldsymbol{v}_1 \cdot \boldsymbol{x})\boldsymbol{x} \right) \right] \right\} .
\end{aligned}
$$

Thus,

$$
\begin{aligned}
\dot{\boldsymbol{v}}_1 &= \frac{1}{2} \left( \boldsymbol{I} + \boldsymbol{T} \right) \mathop{\mathbb{E}}_{\boldsymbol{x}\sim\mathcal{D}} \left\{ \Pi_{R_{\boldsymbol{v}_1}} \left[ S(\boldsymbol{x}) \left( \bar{\boldsymbol{v}}_1 \sigma(\boldsymbol{v}_1 \cdot \boldsymbol{x}) + \|\boldsymbol{v}_1\| \sigma'(\boldsymbol{v}_1 \cdot \boldsymbol{x})\boldsymbol{x} \right) \right] \right\} \\
&= 2 \left\langle \bar{\boldsymbol{v}}_1, \mathop{\mathbb{E}}_{\boldsymbol{x}\sim\mathcal{D}} \left\{ \Pi_{R_{\boldsymbol{v}_1}} \left[ S(\boldsymbol{x}) \left( \bar{\boldsymbol{v}}_1 \sigma(\boldsymbol{v}_1 \cdot \boldsymbol{x}) + \|\boldsymbol{v}_1\| \sigma'(\boldsymbol{v}_1 \cdot \boldsymbol{x})\boldsymbol{x} \right) \right] \right\} \right\rangle \bar{\boldsymbol{v}}_1.
\end{aligned}
$$

Namely, $\dot{\boldsymbol{v}}_1 = h(\boldsymbol{v}_1)\bar{\boldsymbol{v}}_1$ where

$$
h(\boldsymbol{v}_1) = 2 \left\langle \bar{\boldsymbol{v}}_1, \mathop{\mathbb{E}}_{\boldsymbol{x}\sim\mathcal{D}} \left\{ \Pi_{R_{\boldsymbol{v}_1}} \left[ S(\boldsymbol{x}) \left( \bar{\boldsymbol{v}}_1 \sigma(\boldsymbol{v}_1 \cdot \boldsymbol{x}) + \|\boldsymbol{v}_1\| \sigma'(\boldsymbol{v}_1 \cdot \boldsymbol{x})\boldsymbol{x} \right) \right] \right\} \right\rangle .
$$

Now, we show that $h$ is spherically symmetric to complete the proof. Let $\boldsymbol{R}$ be an arbitrary rotation matrix. We have

$$
\begin{aligned}
h(\boldsymbol{R}\boldsymbol{v}_1) &= 2 \left\langle \boldsymbol{R}\bar{\boldsymbol{v}}_1, \mathop{\mathbb{E}}_{\boldsymbol{x}\sim\mathcal{D}} \left\{ \Pi_{R_{\boldsymbol{v}_1}} \left[ S(\boldsymbol{x}) \left( \boldsymbol{R}\bar{\boldsymbol{v}}_1 \sigma(\boldsymbol{R}\boldsymbol{v}_1 \cdot \boldsymbol{x}) + \|\boldsymbol{v}_1\| \sigma'(\boldsymbol{R}\boldsymbol{v}_1 \cdot \boldsymbol{x})\boldsymbol{x} \right) \right] \right\} \right\rangle \\
&= 2 \left\langle \boldsymbol{R}\bar{\boldsymbol{v}}_1, \mathop{\mathbb{E}}_{\boldsymbol{x}\sim\mathcal{D}} \left\{ \Pi_{R_{\boldsymbol{v}_1}} \left[ S(\boldsymbol{x}) \left( \boldsymbol{R}\bar{\boldsymbol{v}}_1 \sigma(\boldsymbol{v}_1 \cdot \boldsymbol{R}^\top \boldsymbol{x}) + \|\boldsymbol{v}_1\| \sigma'(\boldsymbol{v}_1 \cdot \boldsymbol{R}^\top \boldsymbol{x})\boldsymbol{R}\boldsymbol{R}^\top \boldsymbol{x} \right) \right] \right\} \right\rangle \\
&= 2 \left\langle \boldsymbol{R}\bar{\boldsymbol{v}}_1, \mathop{\mathbb{E}}_{\boldsymbol{x}\sim\boldsymbol{R}^\top\#\mathcal{D}} \left\{ \Pi_{R_{\boldsymbol{v}_1}} \left[ S(\boldsymbol{x}) \left( \boldsymbol{R}\bar{\boldsymbol{v}}_1 \sigma(\boldsymbol{v}_1 \cdot \boldsymbol{x}) + \|\boldsymbol{v}_1\| \sigma'(\boldsymbol{v}_1 \cdot \boldsymbol{x})\boldsymbol{R}\boldsymbol{x} \right) \right] \right\} \right\rangle \\
&= 2 \left\langle \boldsymbol{R}\bar{\boldsymbol{v}}_1, \boldsymbol{R} \mathop{\mathbb{E}}_{\boldsymbol{x}\sim\mathcal{D}} \left\{ \Pi_{R_{\boldsymbol{v}_1}} \left[ S(\boldsymbol{x}) \left( \bar{\boldsymbol{v}}_1 \sigma(\boldsymbol{v}_1 \cdot \boldsymbol{x}) + \|\boldsymbol{v}_1\| \sigma'(\boldsymbol{v}_1 \cdot \boldsymbol{x})\boldsymbol{x} \right) \right] \right\} \right\rangle \\
&= 2 \left\langle \bar{\boldsymbol{v}}_1, \mathop{\mathbb{E}}_{\boldsymbol{x}\sim\mathcal{D}} \left\{ \Pi_{R_{\boldsymbol{v}_1}} \left[ S(\boldsymbol{x}) \left( \bar{\boldsymbol{v}}_1 \sigma(\boldsymbol{v}_1 \cdot \boldsymbol{x}) + \|\boldsymbol{v}_1\| \sigma'(\boldsymbol{v}_1 \cdot \boldsymbol{x})\boldsymbol{x} \right) \right] \right\} \right\rangle \\
&= h(\boldsymbol{v}_1).
\end{aligned}
$$

Thus, $h$ is spherically symmetric.

## C  STAGE 1

The goal of Stage 1 is for all $v_2$ to decrease to $-\Theta(1/R_{\boldsymbol{v}_2})$ so that we can ignore all projection operators in $\dot{r}_2$, $\dot{\boldsymbol{v}}_1$ and $\dot{v}_2$. We split Stage 1 into three substages, in which $v_2$ decreases to $0$, $-\operatorname{poly}(d)\delta_2$ and $-\Theta(1/R_{\boldsymbol{v}_2})$, respectively. By Lemma C.3, at the end of each substage, one more projection operator can be ignored. We also show that, in Stage 1, the approximation error of the first layer and the spread of second layer cannot grow too much.

First, for the initialization, by some standard concentration argument, we have the following lemma.

**Lemma C.1** (Initialization). *We choose $m_1 = \operatorname{poly}(d, 1/\varepsilon)$, $m_2 = \Theta(1)$, $\sigma_1 = 1/\sqrt{d}$, $\sigma_2 = 1/\operatorname{poly}(d, 1/\varepsilon)$, and $\sigma_r$ to be a small constant. We initialize $\boldsymbol{w}_1 \sim \operatorname{Unif}(\sigma_1 \mathbb{S}^{d-1})$ for $\mu_1$, and $w_2 \sim \mathcal{N}(0, \sigma_2^2)$ and $b_2 = \sigma_r$ for $\mu_2$.*

*Given $\delta_{1,I} = 1/\operatorname{poly}_1(d, 1/\varepsilon)$, we choose a sufficiently large $m_1$ so that, at initialization, with probability at least $1 - 1/\operatorname{poly}(d)$, $\left\|\bar{F}|_{\mathbb{S}^{d-1}} - 1\right\|_{L^\infty} \leq \delta_{1,I}$. We also choose $\sigma_2 = \delta_{1,I}/d^7$. With probability at least $1 - 1/\operatorname{poly}(d)$, we have $\max_{w_2} |w_2| \leq O(\log d)\sigma_2$.*

Then, we formally state the Induction Hypothesis we are going to maintain for Stage 1.

**Induction Hypothesis C.2** (Stage 1). *We define $T_1 := \inf\{t \geq 0 : -\bar{w}_2(t) = \Theta(1)/R_{v_2}\}$ for some large constant. Define $\delta_{1,T}^{(1)}, \delta_{1,R}^{(1)}, \delta_2^{(1)}$ as[9]*

$$
\begin{cases}
\delta_{1,T}^{(1)} = \max\left\{ \delta_{1,T}^{(1)}(0). \max_{\boldsymbol{v}_1 \in \mu_1} \|\bar{\boldsymbol{v}}_1(t) - \bar{\boldsymbol{v}}_1(0)\| \right\}, \\[2ex]
\delta_{1,R}^{(1)} = \max\left\{ \delta_{1,R}^{(1)}(0). \max_{\boldsymbol{v}_1 \in \mu_1} \left| \dfrac{\|\boldsymbol{v}_1\|^2 - \mathbb{E}_{\boldsymbol{w}_1}\|\boldsymbol{w}_1\|^2}{\mathbb{E}_{\boldsymbol{w}_1}\|\boldsymbol{w}_1\|^2} \right| \right\}, & \text{in Stage 1.1 and Stage 1.2,} \\[2ex]
\delta_2^{(1)} = \max\left\{ \delta_2^{(1)}(0), \max_{(v_2,r_2),(v_2',r_2')} \|(v_2, r_2) - (v_2', r_2')\| \right\},
\end{cases}
$$

*and*

$$
\begin{cases}
\dfrac{\mathrm{d}}{\mathrm{d}t}\delta_{1,T}^{(1)} = \operatorname{ReLU}\left( \dfrac{\mathrm{d}}{\mathrm{d}t} \max_{\boldsymbol{v}_1 \in \mu_1} \|\bar{\boldsymbol{v}}_1(t) - \bar{\boldsymbol{v}}_1(0)\| \right), \\[2ex]
\dfrac{\mathrm{d}}{\mathrm{d}t}\delta_{1,R}^{(1)} = \operatorname{ReLU}\left( \dfrac{\mathrm{d}}{\mathrm{d}t} \max_{\boldsymbol{v}_1 \in \mu_1} \left| \dfrac{\|\boldsymbol{v}_1\|^2 - \mathbb{E}_{\boldsymbol{w}_1}\|\boldsymbol{w}_1\|^2}{\mathbb{E}_{\boldsymbol{w}_1}\|\boldsymbol{w}_1\|^2} \right| \right), & \text{in Stage 1.3,} \\[2ex]
\dfrac{\mathrm{d}}{\mathrm{d}t}\delta_2^{(1)} = \operatorname{ReLU}\left( \dfrac{\mathrm{d}}{\mathrm{d}t} \max_{(v_2,r_2),(v_2',r_2')} \|(v_2, r_2) - (v_2', r_2')\| \right),
\end{cases}
$$

*with initial value $\delta_{1,T}^{(1)}(0) = \delta_{1,R}^{(1)}(0) = 0$ and $\delta_2^{(1)}(0) = \Theta(\sigma_2 \log d)$.*

*We say that this Induction Hypothesis is true at time $t \in [0, T_1]$ if the following hold.[10]*

(a) **Approximation error of the first layer.** *For each $\boldsymbol{v}_1 \in \mu_1$, $\|\bar{\boldsymbol{v}}_1(t) - \bar{\boldsymbol{v}}_1(0)\| \leq \delta_{1,T}^{(1)}$ and $\|\boldsymbol{v}_1\|^2 = \left(1 \pm \delta_{1,R}^{(2)}\right) \mathbb{E}_{\boldsymbol{w}_1 \sim \mu_1}\|\boldsymbol{w}_1\|^2$.*

(b) **Spread of the second layer.** *For any $(v_2, r_2), (v_2', r_2') \in \mu_2$, $\|(v_2, r_2) - (v_2', r_2')\| \leq \delta_2^{(1)}$.*

(c) **The bias term.** *For any $(v_2, r_2) \in \mu_2$, $r_2 = \Theta(1)$.*

(d) **Size of $f$.** *$|\bar{w}_2| = O(1/R_{v_2}) = O(1/d^3)$ and $\alpha = \Theta(\sqrt{d}/R_{\boldsymbol{v}_1}) = \Theta(1/d^{1.5})$.*

(e) **Bounds for the errors.** *$\delta_2^{(1)} \leq O(d^{1.5}(\log d)\sigma_2)$ and $\delta_{1,R}^{(1)} + \delta_{1,T}^{(1)} \leq O(d^7(\log d)\sigma_2 + \delta_{1,I})$*

The next lemma describes when the projection operators can be ignored. Roughly speaking, we first bound the gradients to show that in order for a projection operator to be triggered, $\|\boldsymbol{x}\|$ must be larger than a certain quantity. Meanwhile, note that $f$, whence the gradients, vanishes for those $\boldsymbol{x}$ with $\|\boldsymbol{x}\| \geq \Theta(1/|\bar{w}_2\alpha|)$. Hence, as long as $\Theta(1/|\bar{w}_2\alpha|)$ is smaller than that quantity, we can ignore the projection.

---

[9]Note that we define these $\delta$'s to be upper bounds of the corresponding values instead the values themselves. The only reason we define these $\delta$'s in such a twisted way is to make the proof easier to write rigorously. See the footnote in Induction Hypothesis D.1, where this type of definitions plays more technically important role, for further discussions.

[10]The first two conditions actually follow directly from the definition of the $\delta$'s. We put repeat them here only for easier reference. The actual result we need to prove for these $\delta$'s is condition (e), which says that these $\delta$'s are always small.

**Lemma C.3.** *Suppose that Induction Hypothesis C.2 is true. The projection operators in $\dot{r}_2$, $\dot{v}_1$ and $\dot{v}_2$ will no longer be activated if all second layer weights are nonpositive, $-\bar{w}_2 > \Theta(1)\delta_2^{(1)}$ for some large constant, and $-\bar{w}_2 \geq \Theta(1)/R_{v_2}$ for some large constant, respectively.*

**Remark.** Though we only need $-\bar{w}_2$ to be $\Theta(1)\delta_2^{(1)}$ to ignore the projection operator in $\dot{v}_1$, we will actually define the end of Stage 1.2 to be the time $-\bar{w}_2$ becomes $\mathrm{poly}(d)\delta_2^{(1)}$ to get a more regular start for Stage 1.3. ♣

Now, we present the main lemma of Stage 1. One can see that, by properly choosing the parameters, the errors can be made arbitrarily small without affecting the final value of $\alpha$ and $\bar{w}_2$. To prove the main lemma, it suffices to combine Lemma C.6, Lemma C.9 and Lemma C.10 together.

**Lemma C.4** (Main lemma of Stage 1)**.** *Induction Hypothesis C.2 is true throughout Stage 1. Stage 1 takes at most $O(d^4\sigma_2 + 1/d^{1.5})$ amount of time. At the end of Stage 1, we have $\alpha = \Theta(1/d^{1.5})$ and $-\bar{w}_2 = \Theta(1/d^3)$. For the errors, we have $\delta_2^{(2)} \leq O(d^{1.5}\log d\sigma_2)$ and $\delta_{1,R}^{(1)} + \delta_{1,T}^{(1)} \leq O(\delta_{1,I})$.*

*Proof of Lemma C.3.* First, note that when all $v_2$ are nonpositive, we have $f = O(1)$. Since we choose $R_{r_2}$ to be a large constant, this implies the projection operator in $\dot{r}_2$ will not be activated. When $-\bar{w}_2 > \Theta(1)\delta_2^{(1)}$, we have $f(\boldsymbol{x}) \leq \sigma(c\bar{w}_2\alpha\|\boldsymbol{x}\| + O(1))$ for some small constant $c > 0$. As a result, $f$ vanishes on $\{\|\boldsymbol{x}\| \geq (-c\bar{w}_2\alpha)^{-1}\}$. Then, for those $\boldsymbol{x}$ with $\|\boldsymbol{x}\| \leq (-c\bar{w}_2\alpha)^{-1}$, the gradient w.r.t. $\boldsymbol{v}_1$ can be bounded as

$$\|\nabla_{\boldsymbol{v}_1}\mathcal{L}(\boldsymbol{x})\| \leq O(1)|\bar{w}_2|\|\boldsymbol{x}\|\|\boldsymbol{v}_1\| \leq O(1)|\bar{w}_2|\|\boldsymbol{v}_1\|\frac{1}{|\bar{w}_2|\alpha} \leq O(d).$$

Since we choose $R_{\boldsymbol{v}_1} = \Theta(d)$ with a large constant, this implies the projection operator in $\dot{v}_1$ will not be triggered. Finally, for $\dot{v}_2$, for those $\boldsymbol{x}$ with $\|\boldsymbol{x}\| \leq (-c\bar{w}_2\alpha)^{-1}$, we have

$$|\nabla_{v_2}\mathcal{L}(\boldsymbol{x})| \leq O(1)\alpha\|\boldsymbol{x}\| \leq \frac{O(1)}{|\bar{w}_2|}.$$

By assumption, $|\bar{w}_2| = \Theta(1)/R_{v_2}$ for some large constant. Hence, this inequality implies the projection operator in $\dot{v}_2$ will not be triggered. □

## C.1 STAGE 1.1

The goal of Stage 1.1 is to make sure that all second layer weights $v_2$ become non-positive, that is,

$$T_{1.1} := \inf\{t \geq 0 : \forall(v_2, r_2) \in \mu_2, v_2 \leq 0\}.$$

As a result, at the end of Stage 1.1, $f$ is $O(1)$ and, by Lemma C.3, the projection operator in $\dot{r}_2$ can be ignored. Since this stage only takes a very small amount of time, we shall control the first layer error by directly bounding the movement of $\boldsymbol{v}_1$. For the second layer, we bound the movement of the bias term in the same brute-force way. For second layer weights, we show that those positive $v_2$'s decrease faster than the negative $v_2$'s, so the spread will not increase.

**Lemma C.5.** *Suppose that Induction Hypothesis C.2 is true at time $t$ and $t \leq T_{1.1}$. Then the following hold.*

(a) *$\|\dot{\boldsymbol{v}}_1\| \leq R_{\boldsymbol{v}_1}$ and $|\dot{r}_2| \leq R_{r_2}$.*

(b) *$\max_{w_2} w_2 - \min_{w_2} w_2$ is non-increasing.*

(c) *For any positive second layer weight $v_2$, we have $\dot{v}_2 \leq -\Theta(\log d/d^{1.5})$.*

**Remark.** In fact, (c) holds whenever $\alpha = \Omega(1/d^{1.5})$ and $v_2F(\boldsymbol{x})+r_2 \geq \Theta(1)$ for any $(v_2, r_2) \in \mu_2$ and $\boldsymbol{x} \in \{\|\boldsymbol{x}\| \leq d^{1.5}\}$, which is always true throughout Stage 1. This estimation will also be used in Stage 1.2 and Stage 1.3. ♣

**Lemma C.6** (Main lemma of Stage 1.1). *Stage 1.1 takes at most $O(d^{1.5}\delta_2^{(1)}(0))$ amount of time. At the end of Stage 1.1, all second layer weights $v_2$ are non-positive. Hence, $f = O(1)$ and, by Lemma C.3, the projection operator in $\dot{r}_2$ can no longer be activated.*

*For the errors, we have $\delta_2^{(1)}(T_{1.1}) \le O(d^{1.5}\delta_2^{(1)}(0))$, and both $\delta_{1,R}^{(1)}(T_{1.1})$ and $\delta_{1,T}^{(1)}(T_{1.1})$ can be bounded by $O(d^3\delta_2^{(1)}(0))$.*

*Proof of Lemma C.5.*

(a) This is obvious.

(b) First, we decompose $v_2$ as

$$\dot{v}_2 = \mathbb{E}_{\|\boldsymbol{x}\|\le 1}\left\{(f_*(\boldsymbol{x}) - f(\boldsymbol{x}))F(\boldsymbol{x})\right\} - \mathbb{E}_{\|\boldsymbol{x}\|\ge 1}\left\{\Pi_{R_{v_2}}\left[f(\boldsymbol{x})\sigma'(v_2F(\boldsymbol{x}) + r_2)F(\boldsymbol{x})\right]\right\}.$$

Note that the first term does not depend on $v_2$, and, for the second term, $\sigma'(v_2F(\boldsymbol{x})+r_2) = 1$ whenever $v_2 \ge 0$. As a result, the speed of positive $v_2$ is uniform and more negative than those $v_2 < 0$. Thus, $\max_{w_2} w_2 - \min_{w_2} w_2$ is non-increasing.

(c) Clear that $\mathbb{E}_{\|\boldsymbol{x}\|\le 1}\left\{(f_*(\boldsymbol{x}) - f(\boldsymbol{x}))F(\boldsymbol{x})\right\} = O(\alpha)$. For the second term, first note that for any $\boldsymbol{x}$ with $\|\boldsymbol{x}\| \le d^{1.5}$, we have

$$f(\boldsymbol{x})F(\boldsymbol{x}) \le O\left(1 + \max_{w_2} w_2\alpha\|\boldsymbol{x}\|\right)\alpha\|\boldsymbol{x}\| \le R_{v_2} \quad \text{and} \quad f(\boldsymbol{x}) \ge \Theta(1) - \max_{w_2}|w_2|\alpha\|\boldsymbol{x}\| = \Theta(1).$$

As a result,

$$\mathbb{E}_{\|\boldsymbol{x}\|\ge 1}\left\{\Pi_{R_{v_2}}\left[f(\boldsymbol{x})\sigma'(v_2F(\boldsymbol{x}) + r_2)F(\boldsymbol{x})\right]\right\} \ge \Theta(\alpha)\mathbb{E}_{1\le\|\boldsymbol{x}\|\le d^{1.5}}\|\boldsymbol{x}\| = \Theta\left((\log d)\alpha\right).$$

Thus, $\dot{v}_2 \le -\Theta(\log d/d^{1.5})$.

$\square$

*Proof of Lemma C.6.* By Lemma C.5, it takes at most $O(d^{1.5}\delta_2^{(1)}(0))$ amount of time for all $v_2$ to become nonpositive. Within this amount of time, $r_2$ at most changes $O(d^{1.5}\delta_2^{(1)}(0))$. Since the spread of $w_2$ does not increase, this implies $\delta_2^{(1)}(T_{1.1}) \le O(d^{1.5}\delta_2^{(1)}(0))$. Finally, the change of $\boldsymbol{v}_1$ can be bounded by $O(d^{2.5}\delta_2^{(1)}(0))$. As a result, both $\delta_{1,R}^{(1)}(T_{1.1})$ and $\delta_{1,T}^{(1)}(T_{1.1})$ can be bounded by $O(d^3\delta_2^{(1)}(0))$. $\square$

## C.2  STAGE 1.2

The goal of Stage 1.2 is to make sure $-\bar{w}_2 \ge d\delta_2^{(1)}(T_{1.1})$. Namely,

$$T_{1.2} := \inf\left\{t \ge T_{1.1} : -\bar{w}_2 = d\delta_2^{(1)}(T_{1.1})\right\}.$$

We will also show that $\delta_2^{(1)}(T_{1.2}) = O(\delta_2^{(1)}(T_{1.1}))$ so $\delta_2^{(1)}(T_{1.2})/|\bar{w}_2| = O(1/d)$ at the end of Stage 1.2. Moreover, by Lemma C.3, at the end of Stage 1.2, the projection operator in $\dot{\boldsymbol{v}}_1$ will no longer be activated. We also show that $r_2$ remains $\Theta(1)$ throughout Stage 1 in this subsection.

The first layer error is again controlled in a brute-force way. For the second layer spread, we show that since $|v_2|$ is small, $\sigma'(v_2F(\boldsymbol{x}) + r_2) = 1$ for most of $\boldsymbol{x}$ and, as a result, the change of $(v_2, r_2)$ is approximately uniform.

**Lemma C.7.** *Suppose that Induction Hypothesis C.2 is true at time $t$. Then, for any $(v_2, r_2) \in \mu_2$, $\dot{r}_2 > 0$ when $r \le \mathbb{E}f_*/2$ and $\dot{r}_2 < 0$ when $r \ge 2\mathbb{E}f_*$. As a result, $r_2 = \Theta(1)$ throughout Stage 1.*

**Lemma C.8** (Spread of the second layer). *Suppose that Induction Hypothesis C.2 is true at time $t$ and $t \leq T_{1.2}$. Then, for any $(v_2, r_2), (v'_2, r'_2) \in \mu_2$, we have*

$$\frac{\mathrm{d}}{\mathrm{d}t} \|(v_2, r_2) - (v'_2, r'_2)\|^2 \leq O(d^{2.5}) \left(\delta_2^{(1)}\right)^2.$$

Though, by this Lemma, the error $\delta_2^{(1)}$ can grow exponentially fast and the growth rate is quite large, it will not blow up as $\dot{v}_2 \leq -\Theta(\log d/d^{1.5})$, so the time needed for Stage 1.2 is much shorter than $1/d^{2.5}$.

**Lemma C.9** (Main lemma of Stage 1.2). *Stage 1.2 takes at most $O(d^{2.5}\delta_2^{(1)}(T_{1.1}))$ amount of time. At the end of Stage 1.2, we have, for any $(v_2, r_2) \in \mu_2$, $-v_2 \geq \Theta(d)\delta_2^{(1)}(T_{1.1})$.*

*For the errors, the spread of the second layer is $(1 + o(1))\delta_2^{(1)}(T_{1.1})$, and both $\delta_{1,R}^{(1)}(T_{1.2})$ and $\delta_{1,T}^{(1)}(T_{1.2})$ can be bounded by $O(d^4\delta_2^{(1)}(T_{1.1}))$.*

*Proof of Lemma C.7.* We write

$$\dot{r}_2 = \mathbb{E}_{\boldsymbol{x}} \left\{(f_*(\boldsymbol{x}) - f(\boldsymbol{x}))\sigma'(v_2 F(\boldsymbol{x}) + r_2)\right\} = \mathbb{E}_{\boldsymbol{x}} f_*(\boldsymbol{x}) - \mathbb{E}_{\boldsymbol{x}} \left\{f(\boldsymbol{x})\sigma'(v_2 F(\boldsymbol{x}) + r_2)\right\}$$

Since the spread of $b_2$ is $o(1)$, when $r_2 \leq \mathbb{E}_{\boldsymbol{x}} f_*(\boldsymbol{x})/2 = \Theta(1)$, the RHS is a positive constant. In other word, $r_2$ will keep grow. Meanwhile, since the second term can be bounded as $\mathbb{E}_{\boldsymbol{x}} \left\{f(\boldsymbol{x})\sigma'(v_2 F(\boldsymbol{x}) + r_2)\right\} \geq \mathbb{E}_{\|\boldsymbol{x}\| \leq d^2} \left\{f(\boldsymbol{x})\right\} \geq (1 - o(1))\bar{b}_2$, when $r_2 \geq 2\mathbb{E} f_*(\boldsymbol{x})$, $\dot{r}_2$ will become a negative constant and $r_2$ will decrease. Combine this two cases together, and we complete the proof. $\square$

*Proof of Lemma C.8.* Since $|v_2| \leq d\delta_2^{(1)}(T_{1.1})$, $F(\boldsymbol{x}) = \Theta(\alpha)\|\boldsymbol{x}\|$ and $r_2 = \Theta(1)$, $v_2 F(\boldsymbol{x}) + r_2 > 0$ for all $\boldsymbol{x}$ with $\|\boldsymbol{x}\| \leq \Theta(\sqrt{d}/\delta_2^{(1)}(T_{1.1}))$. Hence, we can rewrite $\dot{v}_2$ as

$$\dot{v}_2 = \mathbb{E}_{\|\boldsymbol{x}\| \leq \Theta(\sqrt{d}/\delta_2^{(1)}(T_{1.1}))} \left\{\Pi_{R_{v_2}} \left[(f_*(\boldsymbol{x}) - f(\boldsymbol{x}))F(\boldsymbol{x})\right]\right\}$$
$$- \mathbb{E}_{\|\boldsymbol{x}\| \geq \Theta(\sqrt{d}/\delta_2^{(1)}(T_{1.1}))} \left\{\Pi_{R_{v_2}} \left[f(\boldsymbol{x})\sigma'(v_2 F(\boldsymbol{x}) + r_2)F(\boldsymbol{x})\right]\right\}.$$

The first term does not depend on $v_2$ and, by the tail bound, the second term can be bounded by $O(R_{v_2}\delta_2^{(1)}(T_{1.1})/\sqrt{d})$. Similarly, for $\dot{r}_2$, we have

$$\dot{r}_2 = \mathbb{E}_{\|\boldsymbol{x}\| \leq \Theta(\sqrt{d}/\delta_2^{(1)}(T_{1.1}))} \left\{f_*(\boldsymbol{x}) - f(\boldsymbol{x})\right\} \pm O\left(\delta_2^{(1)}(T_{1.1})/\sqrt{d}\right).$$

Hence, for any $(v_2, r_2), (v'_2, r'_2) \in \mu_2$, we have

$$\frac{\mathrm{d}}{\mathrm{d}t} \|(v_2, r_2) - (v'_2, r'_2)\|^2 \leq (v_2 - v'_2)O\left(\frac{R_{v_2}\delta_2^{(1)}(T_{1.1})}{\sqrt{d}}\right) + (r_2 - r'_2)O\left(\frac{\delta_2^{(1)}(T_{1.1})}{\sqrt{d}}\right) \leq O(d^{2.5})\left(\delta_2^{(1)}\right)^2.$$

$\square$

*Proof of Lemma C.9.* Recall from Lemma C.5 that $\dot{v}_2 = -\Theta(\log d/d^{1.5})$, whence Stage 1.2 takes at most $O(d^{2.5}\delta_2^{(1)}(T_{1.1}))$ amount of time. By Lemma C.8, we have

$$\left(\delta_2^{(1)}(T_{1.2})\right)^2 \leq \left(\delta_2^{(1)}(T_{1.1})\right)^2 \exp\left(O(d^5)\delta_2^{(1)}(T_{1.1})\right) \leq (1 + o(1))\left(\delta_2^{(1)}(T_{1.1})\right)^2.$$

For $\boldsymbol{v}_1$, similar to the proof of Lemma C.6, both $\delta_{1,R}^{(1)}(T_{1.2})$ and $\delta_{1,T}^{(1)}(T_{1.2})$ can be bounded by $O(d^4\delta_2^{(1)}(T_{1.1}))$. $\square$

## C.3 STAGE 1.3

The goal of Stage 1.3 is to make sure $-\bar{w}_2 = \Theta(1/R_{v_2})$ for some large constant, so that, by Lemma C.3, the projection operator in $\dot{v}_2$ can be ignored. That is, we define

$$T_{1.3} := \inf\left\{t \geq T_{1.2} \,:\, -\bar{w}_2(t) = \Theta(1/R_{v_2})\right\}.$$

The time needed for this stage is longer than the time needed for previous stages, so we need less brute-force ways to control the errors. For the first layer, we show that the tangent movement is almost zero and the radial movement is approximately uniform. For the second layer, we show that the spread $\delta_2^{(1)}$ cannot grow too fast.

**Lemma C.10** (Main lemma of Stage 1.3). *Stage 1.3 takes at most $O(1/d^{1.5})$ amount of time. At the end of Stage 1.3, we have $-\bar{w}_2 = \Theta(1/R_{v_2})$ and $\alpha = \Theta(\sqrt{d}/R_{v_1})$.*

*For the errors, the spread of the second layer is $O\left(\delta_2^{(1)}(T_{1.2})\right)$ and the first layer errors are $O\left(\delta_{1,R}^{(1)}(T_{1.2}) + \delta_{1,T}^{(1)}(T_{1.2}) + \delta_{1,I} + \log(d)\delta_2^{(1)}(T_{1.2})\right).$*

*Proof.* Since $\dot{v}_2 = -\Omega(\log d/d^{1.5})$ and $R_{v_2} = \Theta(d^3)$, Stage 1.3 takes at most $O(1/d^{1.5})$ amount of time. Within this amount of time, by Lemma C.18, we have

$$(\delta_2^{(1)}(T_{1.3}))^2 \leq (\delta_2^{(1)}(T_{1.2}))^2 \exp\left(\frac{O(1)}{d^{2.5}}\frac{1}{d^{1.5}}\right) = (1 + o(1))(\delta_2^{(1)}(T_{1.2}))^2.$$

For the first layer, by Lemma C.16, we have

$$\delta_{1,R}^{(1)}(T_{1.3}) + \delta_{1,T}^{(1)}(T_{1.3}) \leq \left(\delta_{1,R}^{(1)}(T_{1.2}) + \delta_{1,T}^{(1)}(T_{1.2}) + \frac{O(1)}{d^3}\delta_{1,I} + O\left(\log(d)\delta_2^{(1)}\right)\right)\exp\left(\frac{O(1)}{d^{2.5}}\frac{1}{d^{1.5}}\right)$$

$$= O\left(\delta_{1,R}^{(1)}(T_{1.2}) + \delta_{1,T}^{(1)}(T_{1.2}) + \frac{\delta_{1,I}}{d^3} + \log(d)\delta_2^{(1)}(T_{1.2})\right).$$

Finally, by Lemma C.17, we have $\alpha(T_{1.3}) = (1 + o(1))\alpha(T_{1.2})$. $\qquad\square$

### C.3.1 ESTIMATIONS RELATED TO $\sigma'(v_2 F(\mathbf{x}) + r_2)$

First, we need some helper results to handle $\sigma'(v_2 F(\boldsymbol{x}) + r_2)$. The conditions for them to hold are mild and are always true throughout the entire training procedure, and we will use these results in later stages, too.

First, we show that when the value of $\sigma'(v_2 F(\boldsymbol{x}) + r_2)$ can change across different $(v_2, r_2)$, the function value must be small. Note that the error here depends on the ratio $\delta_2/|\bar{w}_2|$ and this is why we need $|\bar{w}_2|$ to be $\Theta(d)\delta_2$ instead of merely $\Theta(1)\delta_2$ at the end of Stage 1.2.

**Lemma C.11.** *Suppose that $r_2 = \Theta(1)$, $-v_2 \geq \Omega(\delta_2)$ for any $(v_2, r_2) \in \mu_2$, where $\delta_2$ is the spread of the second layer. If $v_2 F(\boldsymbol{x}) + r_2 = 0$ for some $(v_2, r_2) \in \mu_2$, then $v_2' F(\boldsymbol{x}) + r_2' \leq O((|\bar{w}_2|^{-1} + 1)\delta_2)$ for all $(v_2', r_2') \in \mu_2$.*

**Remark.** It is not necessary that there really exists a $(v_2, r_2) \in \mu_2$ with $v_2 F(\boldsymbol{x}) + r_2 = 0$. As long as $v_2' F(\boldsymbol{x}) + r_2' \leq 0$ and $v_2'' F(\boldsymbol{x}) + r_2'' \geq 0$ for some $(v_2', r_2'), (v_2'', r_2'') \in \mu_2$, by the continuity, there always exists some point $(v_2, r_2)$ between $(v_2', r_2')$ and $(v_2'', r_2'')$ such that $v_2 F(\boldsymbol{x}) + r_2 = 0$. Moreover, this point is within the spread of the second layer, so this lemma still applies. ♣

Then, we show that we can absorb $\sigma'$ into $f_*$ and $f$.

**Lemma C.12.** *Suppose that the hypothesis of Lemma C.11 is true, and all second layer neurons are activated on $\{\|\boldsymbol{x}\| \leq 1\}$. Then, for any $(v_2, r_2) \in \mu_2$ and $\boldsymbol{x} \in \mathbb{R}^d$, we have*

$$f_*(\boldsymbol{x})\sigma'(v_2 F(\boldsymbol{x}) + r_2) = f_*(\boldsymbol{x}) \quad \text{and} \quad f(\boldsymbol{x})\sigma'(v_2 F(\boldsymbol{x}) + r_2) = f(\boldsymbol{x}) \pm O\left((|\bar{w}_2|^{-1} + 1)\delta_2\right).$$

*As a corollary, we have*

$$f(\boldsymbol{x}) = \sigma(v_2 F(\boldsymbol{x}) + r_2) \pm O\left((|\bar{w}_2|^{-1} + 1)\delta_2\right),$$
$$f(\boldsymbol{x}) = \sigma(\bar{w}_2 F(\boldsymbol{x}) + \bar{b}_2) \pm O\left((|\bar{w}_2|^{-1} + 1)\delta_2\right).$$

As a corollary of Lemma C.11, the measure on which $\sigma'(v_2 F(\boldsymbol{x}) + r_2)$ can differ for different $(v_2, r_2)$ is also small. Here we also use the fact that those $\boldsymbol{x}$ are around $\Theta(1/|\bar{w}_2 \alpha|)$ the tail bound $\|\mathcal{D}\|(r) \leq O(1/r^2)$.

**Lemma C.13.** *Suppose that Induction Hypothesis C.2 is true at time t. For any* $(v_2, r_2), (v_2', r_2') \in \mu_2$, *we have*

$$\mathbb{E}_{\boldsymbol{x}} \{|\sigma'(v_2 F(\boldsymbol{x}) + r_2) - \sigma'(v_2' F(\boldsymbol{x}) + r_2')|\} \leq O\left(\alpha \delta_2^{(1)}\right).$$

*Proof of Lemma C.11.* For any $(v_2', r_2') \in \mu_2$, we can write

$$v_2' F(\boldsymbol{x}) + r_2' = \underbrace{v_2 F(\boldsymbol{x}) + r_2}_{=0} + (v_2' - v_2)F(\boldsymbol{x}) + (r_2' - r_2)$$

$$= \frac{v_2' - v_2}{v_2} (\underbrace{v_2 F(\boldsymbol{x}) + r_2}_{=0} - r_2) + (r_2' - r_2) = r_2 \frac{v_2 - v_2'}{v_2} + (r_2' - r_2).$$

The last term can be bounded as $O((|\bar{w}_2|^{-1} + 1)\delta_2)$. $\qquad \square$

*Proof of Lemma C.12.* Since all second layer neurons are activated on $\{\|\boldsymbol{x}\| \leq 1\}$, we always have $f_*(\boldsymbol{x})\sigma'(v_2 F(\boldsymbol{x}) + r_2) = f_*(\boldsymbol{x})$. Now we consider $f(\boldsymbol{x})\sigma'(v_2 F(\boldsymbol{x}) + r_2)$. If $v_2 F(\boldsymbol{x}) + r_2 > 0$, then we are done. If $v_2' F(\boldsymbol{x}) + r_2' < 0$ for all $(v_2', r_2') \in \mu_2$, then both $f(\boldsymbol{x})\sigma'(v_2 F(\boldsymbol{x}) + r_2)$ and $f(\boldsymbol{x})$ are 0. Therefore, it suffices to consider the case where $v_2 F(\boldsymbol{x}) + r_2 \leq 0$ while $f(\boldsymbol{x}) > 0$. By Lemma D.6, in this case, we have $f(\boldsymbol{x}) \leq O\left((|\bar{w}_2|^{-1} + 1)\delta_2\right)$. $\qquad \square$

*Proof of Lemma C.13.* Since the norm and direction of $\boldsymbol{x}$ are independent, it suffices to fix a direction $\bar{\boldsymbol{x}}$ and consider

$$\mathbb{E}_{r \sim \|\mathcal{D}\|} \{|\sigma'(v_2 r F(\bar{\boldsymbol{x}}) + r_2) - \sigma'(v_2' r F(\bar{\boldsymbol{x}}) + r_2')|\}.$$

For notational simplicity, define $h(v_2, r_2, r) = v_2 r F(\bar{\boldsymbol{x}}) + r_2$. The integrand is nonzero iff the signs of $h(v_2, r_2, r)$ and $h(v_2', r_2', r)$ are different. To bound the length of the interval on which the signs can differ, we write

$$h(v_2, r_2, r) = \bar{w}_2 r F(\bar{\boldsymbol{x}}) + \bar{b}_2 + (v_2 - \bar{w}_2)r F(\bar{\boldsymbol{x}}) + (r_2 - \bar{b}_2)$$

$$= \left(\bar{w}_2 \pm O\left(\delta_2^{(1)}\right)\right) r F(\bar{\boldsymbol{x}}) + \bar{b}_2 \pm O\left(\delta_2^{(1)}\right).$$

Therefore, the length of this interval can be bounded by $O(\delta_2^{(1)}/(\bar{w}_2^2 \alpha))$. Moreover, note that this interval is at $\Theta(1/|\bar{w}_2 \alpha|)$, whence the density on it is $O(\bar{w}_2^2 \alpha^2)$. Thus, the measure of this interval is $O(\alpha \delta_2^{(1)})$. $\qquad \square$

### C.3.2 ESTIMATIONS FOR THE FIRST LAYER

Before we control the error growth, we need a lemma that relates the approximation error with the tangent movement and radial spread of the first layer.

**Lemma C.14.** *Suppose that the tangent movement and radial spread of the first layer neurons can be bounded as* $\|\bar{\boldsymbol{v}}_1(t) - \bar{\boldsymbol{v}}_1(0)\| \leq \delta_{1,T}$ *and* $\|\boldsymbol{v}_1\|^2 = (1 \pm \delta_{1,R}) \mathbb{E}_{\boldsymbol{w}_1} \|\boldsymbol{w}_1\|^2$. *Then*

$$F(\boldsymbol{x}; \mu_1) = \left(1 + \delta_{1,I} + \sqrt{d}\delta_{1,R} + \sqrt{d}\delta_{1,T}\right) \alpha \|\boldsymbol{x}\|.$$

As a simple corollary, we have the following.

**Corollary C.15.** *Suppose that Induction Hypothesis C.2 is true at time t. Then, we have*

$$|f(\boldsymbol{x}) - \tilde{f}(\boldsymbol{x})| = \left(\delta_{1,I} + \sqrt{d}\delta_{1,R}^{(1)} + \sqrt{d}\delta_{1,R}^{(1)}\right) \bar{w}_2 \alpha \|\boldsymbol{x}\|.$$

*As a result, we have*

$$\mathbb{E}_{\boldsymbol{x}} \left\{(f(\boldsymbol{x}) - \tilde{f}(\boldsymbol{x})) \|\boldsymbol{x}\|\right\} \leq \left(\delta_{1,I} + \sqrt{d}\delta_{1,R}^{(1)} + \sqrt{d}\delta_{1,R}^{(1)}\right) \bar{w}_2 \alpha \mathbb{E} \|\boldsymbol{x}\|^2$$

$$\leq \delta_{1,I} + \sqrt{d}\delta_{1,R}^{(1)} + \sqrt{d}\delta_{1,R}^{(1)}.$$

Now, we are ready the control the error of the first layer.

**Lemma C.16.** *Suppose that Induction Hypothesis C.2 is true at time $t$ and $t \in [T_{1.2}, T_{1.3}]$. Then we have*

$$\frac{\mathrm{d}}{\mathrm{d}t}\left(\delta_{1,R}^{(1)} + \delta_{1,T}^{(1)}\right) \leq O\left(\left(\delta_{1,I} + \sqrt{d}\delta_{1,R}^{(1)} + \sqrt{d}\delta_{1,R}^{(1)}\right)\bar{w}_2\right) + O\left(\log(d)\delta_2^{(1)}\right)$$

$$\leq \frac{O(1)}{d^{2.5}}\left(\delta_{1,R}^{(1)} + \delta_{1,T}^{(1)}\right) + \frac{O(1)}{d^3}\delta_{1,I} + O\left(\log(d)\delta_2^{(1)}\right).$$

Finally, we estimate the radial speed of $\boldsymbol{v}_1$ to provide an estimation for the magnitude of $\alpha$ at the end of Stage 1.

**Lemma C.17.** *Suppose that Induction Hypothesis C.2 is true at time $t$ and $t \in [T_{1.2}, T_{1.3}]$. Then we have*

$$\frac{\mathrm{d}}{\mathrm{d}t}\|\boldsymbol{v}_1\|^2 = \Theta\left(\frac{\log d}{\sqrt{d}}\right)\bar{w}_2\|\boldsymbol{v}_1\|^2.$$

*Proof of Lemma C.14.* Define $N^2 = \mathbb{E}_{\boldsymbol{w}_1}\|\boldsymbol{w}_1\|^2$. Let $\mu_1'$ be the distribution obtained by setting the norm of neurons in $\mu_1$ to $N$. We have

$$F(\boldsymbol{x}; \mu_1) = \mathop{\mathbb{E}}_{\boldsymbol{w}_1 \sim \mu_1}\left\{(1 \pm \delta_{1,R})N^2\sigma(\bar{\boldsymbol{w}}_1 \cdot \boldsymbol{x})\right\} = F(\boldsymbol{x}; \mu_1') \pm O(\delta_{1,R}N^2\|\boldsymbol{x}\|).$$

Let $\mu_1''$ be the distribution obtained by moving $\bar{\boldsymbol{v}}_1(t)$ to $\bar{\boldsymbol{v}}_1(0)$ in $\mu_1'$. Then, we have

$$F(\boldsymbol{x}; \mu_1') = N^2 \mathop{\mathbb{E}}_{\boldsymbol{w}_1 \sim \mu_1(0)}\left\{\sigma(\bar{\boldsymbol{w}}_1 \cdot \boldsymbol{x})\right\} \pm O\left(\delta_{1,T}N^2\|\boldsymbol{x}\|\right) = F(\boldsymbol{x}; \mu_1'') \pm O\left(\delta_{1,T}N^2\|\boldsymbol{x}\|\right).$$

Finally, note that

$$F(\boldsymbol{x}; \mu_1'') = \frac{N_t^2}{N_0^2}F(\boldsymbol{x}; \mu_1(0)) = \frac{N_t^2}{N_0^2}(1 \pm \delta_{1,I})\alpha_0\|\boldsymbol{x}\| = (1 \pm \delta_{1,I})\alpha_t\|\boldsymbol{x}\|.$$

Combine these together and we complete the proof. $\square$

*Proof of Lemma C.16.* First, we decompose $\dot{\boldsymbol{v}}_1$ along the tangent and radial directions as follows:

$$\mathrm{Rad}(\dot{\boldsymbol{v}}_1) := \langle \dot{\boldsymbol{v}}_1, \bar{\boldsymbol{v}}_1\rangle \bar{\boldsymbol{v}}_1 = 2\mathop{\mathbb{E}}_{\boldsymbol{x}}\left\{S(\boldsymbol{x})\sigma(\boldsymbol{v}_1 \cdot \boldsymbol{x})\right\}\bar{\boldsymbol{v}}_1,$$

$$\mathrm{Tan}(\dot{\boldsymbol{v}}_1) := (\boldsymbol{I} - \bar{\boldsymbol{v}}_1\bar{\boldsymbol{v}}_1^\top)\dot{\boldsymbol{v}}_1 = \|\boldsymbol{v}_1\|\mathop{\mathbb{E}}_{\boldsymbol{x}}\left\{S(\boldsymbol{x})\sigma'(\boldsymbol{v}_1 \cdot \boldsymbol{x})(\boldsymbol{I} - \bar{\boldsymbol{v}}_1\bar{\boldsymbol{v}}_1^\top)\boldsymbol{x}\right\}.$$

Note that $\dot{\boldsymbol{v}}_1 = \mathrm{Rad}(\dot{\boldsymbol{v}}_1) + \mathrm{Tan}(\dot{\boldsymbol{v}}_1)$. By Lemma C.12, we have

$$\mathrm{Rad}(\dot{\boldsymbol{v}}_1) = 2\bar{w}_2\mathop{\mathbb{E}}_{\boldsymbol{x}}\left\{(f_*(\boldsymbol{x}) - f(\boldsymbol{x}))\sigma(\boldsymbol{v}_1 \cdot \boldsymbol{x})\right\}\bar{\boldsymbol{v}}_1 \pm O\left(\log(d)\delta_2^{(1)}\|\boldsymbol{v}_1\|\right),$$

$$\mathrm{Tan}(\dot{\boldsymbol{v}}_1) = \|\boldsymbol{v}_1\|\bar{w}_2\mathop{\mathbb{E}}_{\boldsymbol{x}}\left\{(f_*(\boldsymbol{x}) - f(\boldsymbol{x}))\sigma'(\boldsymbol{v}_1 \cdot \boldsymbol{x})(\boldsymbol{I} - \bar{\boldsymbol{v}}_1\bar{\boldsymbol{v}}_1^\top)\boldsymbol{x}\right\} \pm O\left(\log(d)\delta_2^{(1)}\|\boldsymbol{v}_1\|\right).$$

For the radial term, by Lemma B.3 and Lemma C.15, we have

$$\mathrm{Rad}(\dot{\boldsymbol{v}}_1) = 2\bar{w}_2\mathop{\mathbb{E}}_{\boldsymbol{x}}\left\{(f_*(\boldsymbol{x}) - \tilde{f}(\boldsymbol{x}))\sigma(\boldsymbol{v}_1 \cdot \boldsymbol{x})\right\}\bar{\boldsymbol{v}}_1 + 2\bar{w}_2\mathop{\mathbb{E}}_{\boldsymbol{x}}\left\{(\tilde{f}(\boldsymbol{x}) - f(\boldsymbol{x}))\sigma(\boldsymbol{v}_1 \cdot \boldsymbol{x})\right\}\bar{\boldsymbol{v}}_1 \pm O\left(\log(d)\delta_2^{(1)}\|\boldsymbol{v}_1\|\right)$$

$$= \frac{2C_\Gamma\bar{w}_2}{\sqrt{d}}\mathop{\mathbb{E}}_{\boldsymbol{x}}\left\{(f_*(\boldsymbol{x}) - \tilde{f}(\boldsymbol{x}))\|\boldsymbol{x}\|\right\}\boldsymbol{v}_1$$

$$\pm O\left(\left(\delta_{1,I} + \sqrt{d}\delta_{1,R}^{(1)} + \sqrt{d}\delta_{1,R}^{(1)}\right)\bar{w}_2\|\boldsymbol{v}_1\|\right) \pm O\left(\log(d)\delta_2^{(1)}\|\boldsymbol{v}_1\|\right).$$

Therefore,

$$\frac{\mathrm{d}}{\mathrm{d}t}\|\boldsymbol{v}_1\|^2 = 2\langle \boldsymbol{v}_1, \mathrm{Rad}(\dot{\boldsymbol{v}}_1)\rangle$$

$$= \frac{4C_\Gamma\bar{w}_2}{\sqrt{d}}\mathop{\mathbb{E}}_{\boldsymbol{x}}\left\{(f_*(\boldsymbol{x}) - \tilde{f}(\boldsymbol{x}))\|\boldsymbol{x}\|\right\}\|\boldsymbol{v}_1\|^2$$

$$\pm O\left(\left(\delta_{1,I} + \sqrt{d}\delta_{1,R}^{(1)} + \sqrt{d}\delta_{1,R}^{(1)}\right)\bar{w}_2\|\boldsymbol{v}_1\|^2\right) \pm O\left(\log(d)\delta_2^{(1)}\|\boldsymbol{v}_1\|^2\right).$$

For any $\boldsymbol{v}_1, \boldsymbol{v}_1' \in \mu_1$ with $\|\boldsymbol{v}_1\| \geq \|\boldsymbol{v}_1'\|$, we have

$$
\begin{aligned}
\frac{\mathrm{d}}{\mathrm{d}t} \frac{\|\boldsymbol{v}_1\|^2 - \|\boldsymbol{v}_1'\|^2}{\|\boldsymbol{v}_1'\|^2} &= \frac{\frac{\mathrm{d}}{\mathrm{d}t}\left(\|\boldsymbol{v}_1\|^2 - \|\boldsymbol{v}_1'\|^2\right)}{\|\boldsymbol{v}_1'\|^2} - \frac{\|\boldsymbol{v}_1\|^2 - \|\boldsymbol{v}_1'\|^2}{\|\boldsymbol{v}_1'\|^2} \frac{\frac{\mathrm{d}}{\mathrm{d}t}\|\boldsymbol{v}_1'\|^2}{\|\boldsymbol{v}_1'\|^2} \\
&= \frac{4 C_\Gamma \bar{w}_2}{\sqrt{d}} \mathbb{E}_{\boldsymbol{x}} \left\{ (f_*(\boldsymbol{x}) - \tilde{f}(\boldsymbol{x})) \|\boldsymbol{x}\| \right\} \frac{\|\boldsymbol{v}_1\|^2 - \|\boldsymbol{v}_1'\|^2}{\|\boldsymbol{v}_1'\|^2} \\
&\quad \pm O\left(\left(\delta_{1,I} + \sqrt{d}\delta_{1,R}^{(1)} + \sqrt{d}\delta_{1,R}^{(1)}\right)\bar{w}_2\right) \pm O\left(\log(d)\delta_2^{(1)}\right) \\
&\quad - \frac{\|\boldsymbol{v}_1\|^2 - \|\boldsymbol{v}_1'\|^2}{\|\boldsymbol{v}_1'\|^2} \frac{4 C_\Gamma \bar{w}_2}{\sqrt{d}} \mathbb{E}_{\boldsymbol{x}} \left\{ (f_*(\boldsymbol{x}) - \tilde{f}(\boldsymbol{x})) \|\boldsymbol{x}\| \right\} \\
&\quad \pm \frac{\|\boldsymbol{v}_1\|^2 - \|\boldsymbol{v}_1'\|^2}{\|\boldsymbol{v}_1'\|^2} O\left(\left(\delta_{1,I} + \sqrt{d}\delta_{1,R}^{(1)} + \sqrt{d}\delta_{1,R}^{(1)}\right)\bar{w}_2\right) \pm \frac{\|\boldsymbol{v}_1\|^2 - \|\boldsymbol{v}_1'\|^2}{\|\boldsymbol{v}_1'\|^2} O\left(\log(d)\delta_2^{(1)}\right) \\
&= \pm O\left(\left(\delta_{1,I} + \sqrt{d}\delta_{1,R}^{(1)} + \sqrt{d}\delta_{1,R}^{(1)}\right)\bar{w}_2\right) \pm O\left(\log(d)\delta_2^{(1)}\right).
\end{aligned}
$$

Now we consider the tangent movement. By Lemma B.5 and Lemma C.15, we have

$$
\begin{aligned}
\mathrm{Tan}(\dot{\boldsymbol{v}}_1) &= \|\boldsymbol{v}_1\| \, \bar{w}_2 \, \mathbb{E}_{\boldsymbol{x}} \left\{ (f_*(\boldsymbol{x}) - \tilde{f}(\boldsymbol{x}))\sigma'(\boldsymbol{v}_1 \cdot \boldsymbol{x})(\boldsymbol{I} - \bar{\boldsymbol{v}}_1 \bar{\boldsymbol{v}}_1^\top)\boldsymbol{x} \right\} \\
&\quad + \|\boldsymbol{v}_1\| \, \bar{w}_2 \, \mathbb{E}_{\boldsymbol{x}} \left\{ (\tilde{f}(\boldsymbol{x}) - f(\boldsymbol{x}))\sigma'(\boldsymbol{v}_1 \cdot \boldsymbol{x})(\boldsymbol{I} - \bar{\boldsymbol{v}}_1 \bar{\boldsymbol{v}}_1^\top)\boldsymbol{x} \right\} \pm O\left(\log(d)\delta_2^{(1)} \|\boldsymbol{v}_1\|\right) \\
&= \pm O\left(\left(\delta_{1,I} + \sqrt{d}\delta_{1,R}^{(1)} + \sqrt{d}\delta_{1,R}^{(1)}\right)\bar{w}_2 \|\boldsymbol{v}_1\|\right) \pm O\left(\log(d)\delta_2^{(1)} \|\boldsymbol{v}_1\|\right).
\end{aligned}
$$

As a result,

$$
\frac{\mathrm{d}}{\mathrm{d}t}\bar{\boldsymbol{v}}_1 = \frac{\mathrm{Tan}(\dot{\boldsymbol{v}}_1)}{\|\boldsymbol{v}_1\|} = \pm O\left(\left(\delta_{1,I} + \sqrt{d}\delta_{1,R}^{(1)} + \sqrt{d}\delta_{1,R}^{(1)}\right)\bar{w}_2\right) \pm O\left(\log(d)\delta_2^{(1)}\right).
$$

Combine these two bounds together and we complete the proof. $\qquad \square$

*Proof of Lemma C.17.* By the proof of Lemma C.16, we have

$$
\begin{aligned}
\mathrm{Rad}(\dot{\boldsymbol{v}}_1) &= \frac{2 C_\Gamma \bar{w}_2}{\sqrt{d}} \mathbb{E}_{\boldsymbol{x}} \left\{ (f_*(\boldsymbol{x}) - \tilde{f}(\boldsymbol{x})) \|\boldsymbol{x}\| \right\} \boldsymbol{v}_1 \\
&\quad \pm O\left(\left(\delta_{1,I} + \sqrt{d}\delta_{1,R}^{(1)} + \sqrt{d}\delta_{1,R}^{(1)}\right)\bar{w}_2 \|\boldsymbol{v}_1\|\right) \pm O\left(\log(d)\delta_2^{(1)} \|\boldsymbol{v}_1\|\right) \\
&= \Theta\left(\frac{\log d}{\sqrt{d}}\right) \bar{w}_2 \boldsymbol{v}_1 \pm O\left(\left(\delta_{1,I} + \sqrt{d}\delta_{1,R}^{(1)} + \sqrt{d}\delta_{1,R}^{(1)}\right)\bar{w}_2 \|\boldsymbol{v}_1\|\right) \pm O\left(\log(d)\delta_2^{(1)} \|\boldsymbol{v}_1\|\right).
\end{aligned}
$$

Recall that $\delta_2^{(1)} \leq |\bar{w}_2|/d$. Hence,

$$
\begin{aligned}
\frac{\mathrm{d}}{\mathrm{d}t}\|\boldsymbol{v}_1\|^2 &= \Theta\left(\frac{\log d}{\sqrt{d}}\right) \bar{w}_2 \|\boldsymbol{v}_1\|^2 \pm O\left(\left(\delta_{1,I} + \sqrt{d}\delta_{1,R}^{(1)} + \sqrt{d}\delta_{1,R}^{(1)}\right)\bar{w}_2 \|\boldsymbol{v}_1\|^2\right) \pm O\left(\log(d)\delta_2^{(1)} \|\boldsymbol{v}_1\|^2\right) \\
&= \Theta\left(\frac{\log d}{\sqrt{d}}\right) \bar{w}_2 \|\boldsymbol{v}_1\|^2,
\end{aligned}
$$

$$\qquad \square$$

### C.3.3 ESTIMATIONS FOR THE SECOND LAYER

Now, we bound the growth of the spread of the second layer. Readers may first check the proof of Lemma D.14, which is essentially a simpler case of this result where we do not need to deal with the projections. In Lemma D.14, we show that the spread will never grow. Here, the error comes from the projection.

**Lemma C.18.** *Suppose that Induction Hypothesis C.2 is true at time $t$. Then we have*

$$
\frac{\mathrm{d}}{\mathrm{d}t}(\delta_2^{(1)})^2 \leq \frac{O(1)}{d^{2.5}}(\delta_2^{(1)})^2.
$$

*Proof.* Let $(v_2, r_2), (v_2', r_2') \in \mu_2$ and define $h_2(\boldsymbol{x}) = v_2 F(\boldsymbol{x}) + r_2$ and $h_2'(\boldsymbol{x}) = v_2' F(\boldsymbol{x}) + r_2'$. We write

$$\dot{v}_2 = \underset{\|\boldsymbol{x}\| \leq 1}{\mathbb{E}} \{(f_*(\boldsymbol{x}) - f(\boldsymbol{x}))F(\boldsymbol{x})\} - \underset{\|\boldsymbol{x}\| \geq 1}{\mathbb{E}} \{\Pi_{R_{v_2}}[f(\boldsymbol{x})\sigma'(h_2(\boldsymbol{x}))F(\boldsymbol{x})]\} =: \mathrm{T}_1(\dot{v}_2) + \mathrm{T}_2(\dot{v}_2).$$

$\mathrm{T}_1$ does not depend on $v_2$. For $\mathrm{T}_2$, note that

$$\Pi_{R_{v_2}}[f(\boldsymbol{x})\sigma'(h_2(\boldsymbol{x}))F(\boldsymbol{x})] = \Pi_{R_{v_2}/F(\boldsymbol{x})}[f(\boldsymbol{x})]\,\sigma'(h_2(\boldsymbol{x}))F(\boldsymbol{x}).$$

Similarly, for $\dot{r}_2$, we have

$$\frac{\mathrm{d}}{\mathrm{d}t}(r_2 - r_2')^2 = -2 \underset{\|\boldsymbol{x}\| \geq 1}{\mathbb{E}} \{f(\boldsymbol{x})(\sigma'(h_2(\boldsymbol{x})) - \sigma'(h_2'(\boldsymbol{x})))(r_2 - r_2')\}$$

$$= -2 \underset{\|\boldsymbol{x}\| \geq 1}{\mathbb{E}} \left\{ \Pi_{R_{v_2}/F(\boldsymbol{x})}[f(\boldsymbol{x})](\sigma'(h_2(\boldsymbol{x})) - \sigma'(h_2'(\boldsymbol{x})))(r_2 - r_2') \right\}$$

$$- 2 \underset{\|\boldsymbol{x}\| \geq 1}{\mathbb{E}} \left\{ \left( f(\boldsymbol{x}) - \Pi_{R_{v_2}/F(\boldsymbol{x})}[f(\boldsymbol{x})] \right) (\sigma'(h_2(\boldsymbol{x})) - \sigma'(h_2'(\boldsymbol{x})))(r_2 - r_2') \right\}.$$

Combine these two equations together and we obtain

$$\frac{\mathrm{d}}{\mathrm{d}t}\left((v_2 - v_2')^2 + (r_2 - r_2')^2\right)$$

$$= -2 \underset{\|\boldsymbol{x}\| \geq 1}{\mathbb{E}} \left\{ \Pi_{R_{v_2}/F(\boldsymbol{x})}[f(\boldsymbol{x})] \left(\sigma'(h_2(\boldsymbol{x})) - \sigma'(h_2'(\boldsymbol{x}))\right)(h_2(\boldsymbol{x}) - h_2'(\boldsymbol{x})) \right\}$$

$$- 2 \underset{\|\boldsymbol{x}\| \geq 1}{\mathbb{E}} \left\{ \left( f(\boldsymbol{x}) - \Pi_{R_{v_2}/F(\boldsymbol{x})}[f(\boldsymbol{x})] \right) (\sigma'(h_2(\boldsymbol{x})) - \sigma'(h_2'(\boldsymbol{x})))(r_2 - r_2') \right\}.$$

Since $\sigma'$ is non-decreasing, the first term is nonpositive. For the second term, by Lemma C.11 and Lemma C.13, it can be bounded as

$$\underset{\boldsymbol{x}:\mathrm{sgn}(h_2(\boldsymbol{x})) \neq \mathrm{sgn}(h_2'(\boldsymbol{x}))}{\max} f(\boldsymbol{x}) \times \underset{\|\boldsymbol{x}\|}{\mathbb{E}} \{|\sigma'(h_2(\boldsymbol{x})) - \sigma'(h_2'(\boldsymbol{x}))|\} \times |r_2 - r_2'| \leq O\left(\frac{\alpha(\delta_2^{(1)})^3}{|\bar{w}_2|}\right) \leq \frac{O(1)}{d^{2.5}}(\delta_2^{(1)})^2.$$

$\square$

## D  STAGE 2

The goal of Stage 2 is for gradient flow to converge to a point with loss $\varepsilon$. Similar to Stage 1, we maintain a set of induction hypotheses.

**Induction Hypothesis D.1.** *Define $T_2 := \inf\{t \geq T_1 : \mathcal{L} = \varepsilon\}$. Define $\delta_{1,L^2}^{(2)}, \delta_{1,L^\infty}, \delta_2^{(2)}$ as*

$$\frac{\mathrm{d}}{\mathrm{d}t}\delta_{1,L^2}^{(2)} = \mathrm{ReLU}\left(\frac{\mathrm{d}}{\mathrm{d}t}\|\bar{F} - \|\cdot\|\|_{L^2}\right), \quad \frac{\mathrm{d}}{\mathrm{d}t}\delta_{1,L^\infty}^{(2)} = \mathrm{ReLU}\left(\frac{\mathrm{d}}{\mathrm{d}t}\|\bar{F}|_{\mathbb{S}^{d-1}} - 1\|_{L^\infty}\right), \quad \frac{\mathrm{d}}{\mathrm{d}t}\delta_2^{(2)} = 0,$$

*with initial value satisfying*[11]

$$\Theta\left(\frac{d^{17}}{\varepsilon}(\delta_{1,L^\infty}^{(2)})^2\right) \leq \delta_{1,L^2}^{(2)} \leq \Theta\left(\frac{\varepsilon}{d^6}\delta_{1,L^\infty}^{(2)}\right),$$

$$\delta_{1,L^2}^{(2)} \leq O\left(\frac{\varepsilon^2}{d^7}\right), \quad \delta_{1,L^\infty}^{(2)}(T_1) \leq O\left(\frac{\varepsilon}{d^{14}}\right), \quad \delta_2^{(2)} \leq O\left(\frac{\varepsilon^2}{d^{10}}\right).$$

*For any $t \in [T_1, T_2]$, we say that this Induction Hypothesis is true if the following hold.*

---

[11]As we have mentioned in the footnote in Induction Hypothesis C.2, these $\delta$'s are defined as upper bounds for the corresponding errors. This gives certain degree of freedom in choosing their initial value. By Lemma C.4, we can choose the parameters so that the errors at the beginning of Stage 2 is arbitrarily small and these conditions can indeed be satisfied. The first condition, which requires the $L^2$ error to be left and right controlled by the $L^\infty$ error, may seem strange at the first sight. The only reason we need it is to merge some second order error terms into first order ones.

(a) **Error of the first layer.** $\left\|\bar{F} - \|\cdot\|\right\|_{L^2} \leq \delta_{1,L^2}^{(2)}$ and $\left\|\bar{F}|_{\mathbb{S}^{d-1}} - 1\right\|_{L^\infty} \leq \delta_{1,L^\infty}^{(2)}$.

(b) **Spread of the second layer.** $\|(v_2, r_2) - (v_2', r_2')\| \leq \delta_2^{(2)}$ for all $(v_2, r_2), (v_2', r_2') \in \mu_2$.

(c) **Regularity conditions.** $\bar{b}_2 \leq 1 - \Theta(\sqrt{\varepsilon})$. $\bar{w}_2\alpha \geq -1 + \Theta(\sqrt{\varepsilon})$. $|\bar{w}_2| \leq d$. $|\bar{w}_2| \geq \Theta(1/d^3)$. $\alpha \geq \Theta(1/d^{1.5})$.

(d) **Bounds for the errors.** $\delta_{1,L^\infty}^{(2)} = O(\delta_{1,L^\infty}^{(2)}(T_1))$ and $\delta_{1,L^2}^{(2)} = O(\delta_{1,L^2}^{(2)}(T_1))$.

The main lemma for Stage 2 is as follows.

**Lemma D.2** (Stage 2). *Induction Hypothesis D.1 is true throughout Stage 2 and Stage 2 takes at most $O(d^3/\varepsilon)$ amount of time.*

The rest of this section is organized as follows. In Section D.1, we collect some auxiliary results that will be used later. In Section D.2, we show that Induction Hypothesis D.1 is always true throughout Stage 2. (Also see Section B.1 for discussion on the techniques used and some conventions.) Then, we derive a lower bound on the convergence rate in Section D.3. Finally, we prove Lemma D.2 in Section D.4.

### D.1 AUXILIARY LEMMAS

#### D.1.1 THE DYNAMICS OF $F$, $f$ AND $\mathcal{L}$

Recall that, in Stage 2, we can ignore the projection operators, whence the dynamics of the neurons is given by

$$\dot{\boldsymbol{v}}_1 = \mathbb{E}_{\boldsymbol{x}}\left\{S(\boldsymbol{x})\left(\bar{\boldsymbol{v}}_1\sigma(\boldsymbol{v}_1 \cdot \boldsymbol{x}) + \|\boldsymbol{v}_1\|\,\sigma'(\boldsymbol{v}_1 \cdot \boldsymbol{x})\boldsymbol{x}\right)\right\},$$

$$\dot{v}_2 = \mathbb{E}_{\boldsymbol{x}}\left\{(f_*(\boldsymbol{x}) - f(\boldsymbol{x}))\sigma'(v_2 F(\boldsymbol{x}) + r_2)F(\boldsymbol{x})\right\},$$

$$\dot{r}_2 = \mathbb{E}_{\boldsymbol{x}}\left\{(f_*(\boldsymbol{x}) - f(\boldsymbol{x}))\sigma'(v_2 F(\boldsymbol{x}) + r_2)\right\}.$$

Now, we derive the equations which describes the dynamics of $\alpha$, $F$, and the loss $\mathcal{L}$.

**Lemma D.3** (Dynamics of $\alpha$). *In Stage 2, we have*

$$\dot{\alpha} = \frac{4C_\Gamma}{\sqrt{d}}\,\mathbb{E}_{\boldsymbol{x}'}\left\{S(\boldsymbol{x}')F(\boldsymbol{x}')\right\}.$$

**Lemma D.4** (Dynamics of $F$). *In Stage 2, for each fixed $\boldsymbol{x}$, we have*

$$\frac{\mathrm{d}}{\mathrm{d}t}F(\boldsymbol{x}) = 4\,\mathbb{E}_{\boldsymbol{x}'}\left\{S(\boldsymbol{x}')\,\mathbb{E}_{\boldsymbol{w}_1}\left\{\sigma(\boldsymbol{w}_1 \cdot \boldsymbol{x}')\sigma(\boldsymbol{w}_1 \cdot \boldsymbol{x})\right\}\right\}$$
$$+ \mathbb{E}_{\boldsymbol{x}'}\left\{S(\boldsymbol{x}')\,\mathbb{E}_{\boldsymbol{w}_1}\left\{\|\boldsymbol{w}_1\|^2\,\sigma'(\boldsymbol{v}_1 \cdot \boldsymbol{x}')\sigma'(\boldsymbol{v}_1 \cdot \boldsymbol{x})\left\langle(\boldsymbol{I} - \bar{\boldsymbol{v}}_1\bar{\boldsymbol{v}}_1^\top)\boldsymbol{x}', \boldsymbol{x}\right\rangle\right\}\right\}.$$

Note that in the above lemma, we decompose $\frac{\mathrm{d}}{\mathrm{d}t}F(\boldsymbol{x})$ into two terms where the first term corresponds to the radial movement of $\boldsymbol{v}_1$ and the second term the tangent movement.

**Lemma D.5** (Dynamics of $\mathcal{L}$). *Define $\bar{W}_2(\boldsymbol{x}) = \mathbb{E}_{w_2,b_2}\left\{\sigma'(w_2 F(\boldsymbol{x}) + b_2)w_2\right\}$. In Stage 2, we have*

$$\frac{\mathrm{d}}{\mathrm{d}t}\mathcal{L} = -\mathbb{E}_{w_2,b_2,\boldsymbol{w}_1}\|\nabla_{w_2,b_2,\boldsymbol{w}_1}\|^2,$$

*where*

$$\nabla_{w_2,b_2,\boldsymbol{w}_1} := \mathbb{E}_{\boldsymbol{x}}\left\{(f_*(\boldsymbol{x}) - f(\boldsymbol{x}))\begin{bmatrix}\sigma'(w_2 F(\boldsymbol{x}) + b_2)F(\boldsymbol{x})\\\sigma'(w_2 F(\boldsymbol{x}) + b_2)\\2\bar{W}_2(\boldsymbol{x})\sigma(\boldsymbol{w}_1 \cdot \boldsymbol{x})\\\|\boldsymbol{w}_1\|\,\bar{W}_2(\boldsymbol{x})\sigma'(\boldsymbol{w}_1 \cdot \boldsymbol{x})(\boldsymbol{I} - \bar{\boldsymbol{w}}_1\bar{\boldsymbol{w}}_1^\top)\boldsymbol{x}\end{bmatrix}\right\}.$$

The entries of $\nabla_{w_2,b_2,\boldsymbol{w}_1}$ correspond to the movements of $v_2$, $r_2$, radial movement of $\boldsymbol{v}_1$ and tangent movement of $\boldsymbol{v}_1$, respectively.

The proofs of these three lemmas are as follows.

*Proof of Lemma D.3.* Recall that $\alpha := \frac{C_\Gamma}{\sqrt{d}} \mathbb{E}_{\boldsymbol{w}_1} \|\boldsymbol{w}_1\|^2$. Hence, $\dot\alpha = \frac{2C_\Gamma}{\sqrt{d}} \mathbb{E}_{\boldsymbol{w}_1} \langle \boldsymbol{w}_1, \dot{\boldsymbol{w}}_1 \rangle$. We compute

$$\langle \dot{\boldsymbol{v}}_1, \boldsymbol{v}_1 \rangle = \mathbb{E}_{\boldsymbol{x}} \left\{ S(\boldsymbol{x}) \left( \sigma(\boldsymbol{v}_1 \cdot \boldsymbol{x}) \langle \bar{\boldsymbol{v}}_1, \boldsymbol{v}_1 \rangle + \|\boldsymbol{v}_1\| \sigma'(\boldsymbol{v}_1 \cdot \boldsymbol{x}) \langle \boldsymbol{x}, \boldsymbol{v}_1 \rangle \right) \right\} = 2 \mathbb{E}_{\boldsymbol{x}} \left\{ S(\boldsymbol{x}) \|\boldsymbol{v}_1\| \sigma(\boldsymbol{v}_1 \cdot \boldsymbol{x}) \right\}.$$

Hence,

$$\dot\alpha = \frac{4C_\Gamma}{\sqrt{d}} \mathbb{E}_{\boldsymbol{w}_1} \left\{ \mathbb{E}_{\boldsymbol{x}} \left\{ S(\boldsymbol{x}) \|\boldsymbol{w}_1\| \sigma(\boldsymbol{w}_1 \cdot \boldsymbol{x}) \right\} \right\} = \frac{4C_\Gamma}{\sqrt{d}} \mathbb{E}_{\boldsymbol{x}} \left\{ S(\boldsymbol{x}) \mathbb{E}_{\boldsymbol{w}_1} \left\{ \|\boldsymbol{w}_1\| \sigma(\boldsymbol{w}_1 \cdot \boldsymbol{x}) \right\} \right\} = \frac{4C_\Gamma}{\sqrt{d}} \mathbb{E}_{\boldsymbol{x}} \left\{ S(\boldsymbol{x}) F(\boldsymbol{x}) \right\}.$$

$\square$

*Proof of Lemma D.4.* First, we write

$$\frac{\mathrm{d}}{\mathrm{d}t} F(\boldsymbol{x}) = \frac{\mathrm{d}}{\mathrm{d}t} \mathbb{E}_{\boldsymbol{w}_1} \left\{ \|\boldsymbol{w}_1\|^2 \sigma(\bar{\boldsymbol{w}}_1 \cdot \boldsymbol{x}) \right\} = \mathbb{E}_{\boldsymbol{w}_1} \left\{ \left( \frac{\mathrm{d}}{\mathrm{d}t} \|\boldsymbol{w}_1\|^2 \right) \sigma(\bar{\boldsymbol{w}}_1 \cdot \boldsymbol{x}) \right\} + \mathbb{E}_{\boldsymbol{w}_1} \left\{ \|\boldsymbol{w}_1\|^2 \frac{\mathrm{d}}{\mathrm{d}t} \sigma(\bar{\boldsymbol{w}}_1 \cdot \boldsymbol{x}) \right\}.$$

By the proof of Lemma D.3, the first term is $4 \mathbb{E}_{\boldsymbol{x}'} \left\{ S(\boldsymbol{x}') \mathbb{E}_{\boldsymbol{w}_1} \left\{ \sigma(\boldsymbol{w}_1 \cdot \boldsymbol{x}') \sigma(\boldsymbol{w}_1 \cdot \boldsymbol{x}) \right\} \right\}$. For the second term, we compute

$$\frac{\mathrm{d}}{\mathrm{d}t} \sigma(\bar{\boldsymbol{v}}_1 \cdot \boldsymbol{x}) = \sigma'(\boldsymbol{v}_1 \cdot \boldsymbol{x}) \left\langle (\boldsymbol{I} - \bar{\boldsymbol{v}}_1 \bar{\boldsymbol{v}}_1^\top) \frac{\dot{\boldsymbol{v}}_1}{\|\boldsymbol{v}_1\|}, \boldsymbol{x} \right\rangle$$

$$= \sigma'(\boldsymbol{v}_1 \cdot \boldsymbol{x}) \left\langle \mathbb{E}_{\boldsymbol{x}'} \left\{ S(\boldsymbol{x}') \sigma'(\boldsymbol{v}_1 \cdot \boldsymbol{x}')(\boldsymbol{I} - \bar{\boldsymbol{v}}_1 \bar{\boldsymbol{v}}_1^\top) \boldsymbol{x}' \right\}, \boldsymbol{x} \right\rangle$$

$$= \mathbb{E}_{\boldsymbol{x}'} \left\{ S(\boldsymbol{x}') \sigma'(\boldsymbol{v}_1 \cdot \boldsymbol{x}') \sigma'(\boldsymbol{v}_1 \cdot \boldsymbol{x}) \left\langle (\boldsymbol{I} - \bar{\boldsymbol{v}}_1 \bar{\boldsymbol{v}}_1^\top) \boldsymbol{x}', \boldsymbol{x} \right\rangle \right\}.$$

Hence, the second term is

$$\mathbb{E}_{\boldsymbol{w}_1} \left\{ \|\boldsymbol{w}_1\|^2 \frac{\mathrm{d}}{\mathrm{d}t} \sigma(\bar{\boldsymbol{w}}_1 \cdot \boldsymbol{x}) \right\} = \mathbb{E}_{\boldsymbol{x}'} \left\{ S(\boldsymbol{x}') \mathbb{E}_{\boldsymbol{w}_1} \left\{ \|\boldsymbol{w}_1\|^2 \sigma'(\boldsymbol{v}_1 \cdot \boldsymbol{x}') \sigma'(\boldsymbol{v}_1 \cdot \boldsymbol{x}) \left\langle (\boldsymbol{I} - \bar{\boldsymbol{v}}_1 \bar{\boldsymbol{v}}_1^\top) \boldsymbol{x}', \boldsymbol{x} \right\rangle \right\} \right\}.$$

Combine these together and we complete the proof. $\square$

*Proof of Lemma D.5.* First, we write

$$\frac{\mathrm{d}}{\mathrm{d}t} f(\boldsymbol{x}) = \mathbb{E}_{w_2, b_2} \left\{ \sigma'(w_2 F(\boldsymbol{x}) + b_2) \dot{w}_2 F(\boldsymbol{x}) \right\} + \mathbb{E}_{w_2, b_2} \left\{ \sigma'(w_2 F(\boldsymbol{x}) + b_2) \dot{b}_2 \right\} + \bar{W}_2(\boldsymbol{x}) \frac{\mathrm{d}}{\mathrm{d}t} F(\boldsymbol{x})$$

$$=: \mathrm{T}_1 \left( \frac{\mathrm{d}}{\mathrm{d}t} f(\boldsymbol{x}) \right) + \mathrm{T}_2 \left( \frac{\mathrm{d}}{\mathrm{d}t} f(\boldsymbol{x}) \right) + \mathrm{T}_3 \left( \frac{\mathrm{d}}{\mathrm{d}t} f(\boldsymbol{x}) \right).$$

Note that $\frac{\mathrm{d}}{\mathrm{d}t} \mathcal{L} = -\sum_{i=1}^3 \mathbb{E}_{\boldsymbol{x}} \left\{ (f_*(\boldsymbol{x}) - f(\boldsymbol{x})) \mathrm{T}_i \left( \frac{\mathrm{d}}{\mathrm{d}t} f(\boldsymbol{x}) \right) \right\}$. Now we compute each of these three terms separately. We have

$$\mathbb{E}_{\boldsymbol{x}} \left\{ (f_*(\boldsymbol{x}) - f(\boldsymbol{x})) \mathrm{T}_1 \left( \frac{\mathrm{d}}{\mathrm{d}t} f(\boldsymbol{x}) \right) \right\} = \mathbb{E}_{w_2, b_2} \left\{ \mathbb{E}_{\boldsymbol{x}} \left\{ (f_*(\boldsymbol{x}) - f(\boldsymbol{x})) \sigma'(w_2 F(\boldsymbol{x}) + b_2) F(\boldsymbol{x}) \right\} \dot{w}_2 \right\}$$

$$= \mathbb{E}_{w_2, b_2} \left\{ \left( \mathbb{E}_{\boldsymbol{x}} \left\{ (f_*(\boldsymbol{x}) - f(\boldsymbol{x})) \sigma'(w_2 F(\boldsymbol{x}) + b_2) F(\boldsymbol{x}) \right\} \right)^2 \right\},$$

$$\mathbb{E}_{\boldsymbol{x}} \left\{ (f_*(\boldsymbol{x}) - f(\boldsymbol{x})) \mathrm{T}_2 \left( \frac{\mathrm{d}}{\mathrm{d}t} f(\boldsymbol{x}) \right) \right\} = \mathbb{E}_{w_2, b_2} \left\{ \mathbb{E}_{\boldsymbol{x}} \left\{ (f_*(\boldsymbol{x}) - f(\boldsymbol{x})) \sigma'(w_2 F(\boldsymbol{x}) + b_2) \right\} \dot{b}_2 \right\}$$

$$= \mathbb{E}_{w_2, b_2} \left\{ \left( \mathbb{E}_{\boldsymbol{x}} \left\{ (f_*(\boldsymbol{x}) - f(\boldsymbol{x})) \sigma'(w_2 F(\boldsymbol{x}) + b_2) \right\} \right)^2 \right\}.$$

Meanwhile, for $\mathrm{T}_3$, by Lemma D.4, we have

$$\mathbb{E}_{\boldsymbol{x}} \left\{ (f_*(\boldsymbol{x}) - f(\boldsymbol{x})) \mathrm{T}_3 \left( \frac{\mathrm{d}}{\mathrm{d}t} f(\boldsymbol{x}) \right) \right\}$$

$$= \mathbb{E}_{\boldsymbol{x}} \left\{ (f_*(\boldsymbol{x}) - f(\boldsymbol{x})) \bar{W}_2(\boldsymbol{x}) \frac{\mathrm{d}}{\mathrm{d}t} F(\boldsymbol{x}) \right\}$$

$$= 4 \mathbb{E}_{\boldsymbol{x}} \left\{ (f_*(\boldsymbol{x}) - f(\boldsymbol{x})) \bar{W}_2(\boldsymbol{x}) \mathbb{E}_{\boldsymbol{x}'} \left\{ S(\boldsymbol{x}') \mathbb{E}_{\boldsymbol{w}_1} \left\{ \sigma(\boldsymbol{w}_1 \cdot \boldsymbol{x}') \sigma(\boldsymbol{w}_1 \cdot \boldsymbol{x}) \right\} \right\} \right\}$$

$$+ \mathbb{E}_{\boldsymbol{x}} \left\{ (f_*(\boldsymbol{x}) - f(\boldsymbol{x})) \bar{W}_2(\boldsymbol{x}) \mathbb{E}_{\boldsymbol{x}'} \left\{ S(\boldsymbol{x}') \mathbb{E}_{\boldsymbol{w}_1} \left\{ \|\boldsymbol{w}_1\|^2 \sigma'(\boldsymbol{w}_1 \cdot \boldsymbol{x}') \sigma'(\boldsymbol{w}_1 \cdot \boldsymbol{x}) \left\langle (\boldsymbol{I} - \bar{\boldsymbol{w}}_1 \bar{\boldsymbol{w}}_1^\top) \boldsymbol{x}', \boldsymbol{x} \right\rangle \right\} \right\} \right\}$$

$$= 4 \mathbb{E}_{\boldsymbol{w}_1} \left\{ \left( \mathbb{E}_{\boldsymbol{x}} \left\{ S(\boldsymbol{x}) \sigma(\boldsymbol{w}_1 \cdot \boldsymbol{x}) \right\} \right)^2 \right\} + \mathbb{E}_{\boldsymbol{w}_1} \left\{ \left\| \mathbb{E}_{\boldsymbol{x}} \left\{ S(\boldsymbol{x}) \|\boldsymbol{w}_1\| \sigma'(\boldsymbol{w}_1 \cdot \boldsymbol{x}) (\boldsymbol{I} - \bar{\boldsymbol{w}}_1 \bar{\boldsymbol{w}}_1^\top) \boldsymbol{x} \right\} \right\|^2 \right\}.$$

Combine these together and we complete the proof. □

### D.1.2 ERROR-RELATED ESTIMATIONS

We collect some error-related estimations here. Most of them have been proved in Stage 1 except that here we have used $|\bar{w}_2| \geq \Theta(1/d^3)$ to replace $(|\bar{w}_2|^{-1}+1)$ with $O(d^3)$. We repeat the statement here for easier reference.

**Lemma D.6.** *Suppose that Induction Hypothesis D.1 is true at time $t$. If $v_2 F(\boldsymbol{x}) + r_2 = 0$ for some $(v_2, r_2) \in \mu_2$, then $v_2' F(\boldsymbol{x}) + r_2' \leq O\left(d^3 \delta_2^{(2)}\right)$ for all $(v_2', r_2') \in \mu_2$.*

*Proof.* See Lemma C.11. □

**Lemma D.7.** *Suppose that Induction Hypothesis D.1 is true at time $t$. Then, for any $(v_2, r_2) \in \mu_2$ and $\boldsymbol{x} \in \mathbb{R}^d$, we have*

$$f_*(\boldsymbol{x})\sigma'(v_2 F(\boldsymbol{x}) + r_2) = f_*(\boldsymbol{x}) \quad and \quad f(\boldsymbol{x})\sigma'(v_2 F(\boldsymbol{x}) + r_2) = f(\boldsymbol{x}) \pm O\left(d^3 \delta_2^{(2)}\right).$$

*As a corollary, we have*

$$f(\boldsymbol{x}) = \sigma(v_2 F(\boldsymbol{x}) + r_2) \pm O\left(d^3 \delta_2^{(2)}\right),$$

$$f(\boldsymbol{x}) = \sigma(\bar{w}_2 F(\boldsymbol{x}) + \bar{b}_2) \pm O\left(d^3 \delta_2^{(2)}\right).$$

*Proof.* See Lemma C.12. □

**Lemma D.8.** *Suppose that Induction Hypothesis D.1 is true at time $t$. Then we have $\left\|f - \tilde{f}\right\|_{L^2} \leq O\left(|\bar{w}_2 \alpha| \delta_{1,L^2}^{(2)}\right).$*

*Proof.* Since $\sigma$ is 1-Lipschitz, we have

$$|f(\boldsymbol{x}) - \tilde{f}(\boldsymbol{x})| = \left| \mathbb{E}_{w_2, b_2} \left\{ \sigma(w_2 F(\boldsymbol{x}) + b_2) - \sigma(w_2 \tilde{F}(\boldsymbol{x}) + b_2) \right\} \right| \leq O\left(|\bar{w}_2||F(\boldsymbol{x}) - \tilde{F}(\boldsymbol{x})|\right).$$

Thus,

$$\left\|f - \tilde{f}\right\|_{L^2}^2 \leq O\left(\bar{w}_2^2 \alpha^2 \left\|\bar{F} - \|\cdot\|\right\|_{L^2}^2\right) \leq O\left(\bar{w}_2^2 \alpha^2 (\delta_{1,L^2}^{(2)})^2\right).$$

□

## D.2 MAINTAINING THE INDUCTION HYPOTHESIS

In this section, we show that Induction Hypothesis D.1 is true throughout Stage 2. See Section B.1 for discussion and conventions on the techniques used here.

### D.2.1 ERROR OF THE FIRST LAYER

Recall that we can decompose the loss as

$$\mathcal{L} = \frac{1}{2} \mathbb{E}_{\boldsymbol{x}} \left\{ (f_*(\boldsymbol{x}) - \tilde{f}(\boldsymbol{x}))^2 \right\} + \frac{1}{2} \mathbb{E}_{\boldsymbol{x}} \left\{ (\tilde{f}(\boldsymbol{x}) - f(\boldsymbol{x}))^2 \right\} + \mathbb{E}_{\boldsymbol{x}} \left\{ (f_*(\boldsymbol{x}) - \tilde{f}(\boldsymbol{x}))(\tilde{f}(\boldsymbol{x}) - f(\boldsymbol{x})) \right\}$$
$$=: \mathcal{L}_1 + \mathcal{L}_2 + \mathcal{L}_3.$$

As we have discussed in the main text, the goal is to show that $\mathcal{L}_2 \approx \frac{\bar{w}_2^2}{2} \mathbb{E}\left\{ (\tilde{F}(\boldsymbol{x}) - F(\boldsymbol{x}))^2 \right\}$ and $\mathcal{L}_3 \approx 0$, so that $\mathcal{L}$ can be decomposed into two terms where the first term captures the difference between the target function $f_*$ and the infinite-width network $f$, and the second term measures the approximation error between $F$ and $\tilde{F}$. We will show in Lemma D.11 that, as one may expect, $\mathcal{L}_1$ does not affect $\bar{F}$. Estimating the gradients of $\mathcal{L}_2$ and $\mathcal{L}_3$ is slightly more complicated. First we need to introduce the following partition on the input space.

**Lemma D.9.** *Define*

$$R_1 := \left\{ R > 0 \ : \ \forall (v_2, r_2) \in \mu_2, \ \boldsymbol{x} \in R\mathbb{S}^{d-1}, \ v_2 F(\boldsymbol{x}) + r_2 > 0 \right\},$$
$$R_2 := \left\{ R > 0 \ : \ \exists (v_2, r_2) \in \mu_2, \ \boldsymbol{x} \in R\mathbb{S}^{d-1}, \ v_2 F(\boldsymbol{x}) + r_2 > 0 \right\}.$$

*Then, we partition the input space into*

$$X_1 := \{\|\boldsymbol{x}\| \le R_1\}, \qquad X_2 := \{R_1 \le \|\boldsymbol{x}\| \le R_2\}, \qquad X_3 := \{R_2 \le \|\boldsymbol{x}\|\}.$$

*In words, $X_1$ is the largest spherically symmetric set on which all second layer neurons are activated, and $X_1 \cup X_2$ is the largest spherically symmetric set on which at least one second layer neuron is activated. Suppose that Induction Hypothesis D.1 is true at time $t$. Then the following hold.*

(a) *$f_*$ vanishes on $X_2 \cup X_3$, i.e., $R_1 \ge 1$, $f$ vanishes on $X_3$, and $R_3 \le O(1/|\bar{w}_2|/\alpha)$.*

(b) *$R_2 - R_1 \le O\left(\delta_{1,L^\infty}^{(2)} + d\delta_2^{(2)}\right) \frac{1}{|\bar{w}_2|\alpha} =: \delta_{X_2}^{(2)}$. As a corollary, we have $\mathbb{P}[X_2] \le O\left(\delta_{X_2}^{(2)}\right)$.*

(c) *$f \le O\left(\delta_{X_2}^{(2)}\right)$ on $X_2$.*

The above lemma implies that $\mathcal{L}_2 \approx \frac{1}{2} \mathbb{E}_{X_1} \left\{ (\tilde{f}(\boldsymbol{x}) - f(\boldsymbol{x}))^2 \right\} = \frac{\bar{w}_2^2}{2} \mathbb{E}_{X_1} \left\{ (\tilde{F}(\boldsymbol{x}) - F(\boldsymbol{x}))^2 \right\}$ and $\mathcal{L}_3 \approx \mathbb{E}_{X_1} \left\{ (f_*(\boldsymbol{x}) - \tilde{f}(\boldsymbol{x}))(\tilde{f}(\boldsymbol{x}) - f(\boldsymbol{x})) \right\} = \bar{w}_2 \mathbb{E}_{X_1} \left\{ (f_*(\boldsymbol{x}) - \tilde{f}(\boldsymbol{x}))(\tilde{F}(\boldsymbol{x}) - F(\boldsymbol{x})) \right\} = 0$. We formally establish this approximation in the following lemma.

**Lemma D.10** (Gradient of $\mathcal{L}_2$ and $\mathcal{L}_3$). *Suppose that Induction Hypothesis D.1 is true at time $t$. Then, for each $\boldsymbol{v}_1 \in \mu_1$, we have*

$$\nabla_{\boldsymbol{v}_1} \mathcal{L}_2 = \nabla_{\boldsymbol{v}_1} \left( \frac{\bar{w}_2^2}{2} \mathbb{E}_{X_1} \left\{ (\tilde{F}(\boldsymbol{x}) - F(\boldsymbol{x}))^2 \right\} \right) \pm O\left( \left(\delta_{X_2}^{(2)}\right)^2 \frac{1}{\alpha} \right) \|\boldsymbol{v}_1\|,$$
$$\nabla_{\boldsymbol{v}_1} \mathcal{L}_3 = \pm O\left( \left(\delta_{X_2}^{(2)}\right)^2 \frac{1}{\alpha} \right) \|\boldsymbol{v}_1\|.$$

Now, we are ready to derive the equation that governs the dynamics of $\bar{F}$. Note that this Lemma implies that, at least approximately, the dynamics of $\bar{F}$ depends only on $\mathcal{L}_2$.

**Lemma D.11** (Dynamics of $\bar{F}$). *Suppose that Induction Hypothesis D.1 is true at time $t$. Then, for each fixed $\boldsymbol{x}$, we have*

$$\frac{\mathrm{d}}{\mathrm{d}t} \bar{F}(\boldsymbol{x}) = -\frac{\bar{w}_2^2}{2} \mathbb{E}_{\boldsymbol{w}_1} \left\{ \left\langle \nabla_{\boldsymbol{w}_1} \bar{F}(\boldsymbol{x}), \nabla_{\boldsymbol{w}_1} \mathbb{E}_{\boldsymbol{x}' \in X_1} \left\{ (\tilde{F}(\boldsymbol{x}') - F(\boldsymbol{x}'))^2 \right\} \right\rangle \right\} \pm O\left( \sqrt{d} \left(\delta_{X_2}^{(2)}\right)^2 \frac{1}{\alpha} \right) \|\boldsymbol{x}\|.$$

Then, we show that the signal term in $\frac{\mathrm{d}}{\mathrm{d}t} \bar{F}(\boldsymbol{x})$ can only decrease the $L^2$ error, which is intuitively true as, after all, $\mathcal{L}_2$ is the (rescaled) $L^2$ error. As a result, the $L^2$ error barely grows.

**Lemma D.12** ($L^2$ approximation error). *Suppose that Induction Hypothesis D.1 is true at time $t$. Then we have*

$$\frac{\mathrm{d}}{\mathrm{d}t} \left\| \bar{F} - \|\cdot\| \right\|_{L^2}^2 \le O\left( d^5 \delta_{1,L^2}^{(2)} \left(\delta_{X_2}^{(2)}\right)^2 \right).$$

Finally, we show that the change $\bar{F}|_{\mathbb{S}^{d-1}}$ depends on the $L^2$ error. As a result, as long as the $L^2$ error is small, the $L^\infty$ error cannot grow too fast.

**Lemma D.13** ($L^\infty$ approximation error). *Suppose that Induction Hypothesis D.1 is true at time $t$. Then, for any $\bar{\boldsymbol{x}} \in \mathbb{S}^{d-1}$, we have*

$$\left| \frac{\mathrm{d}}{\mathrm{d}t} \bar{F}(\bar{\boldsymbol{x}}) \right| \le O\left( d^3 \delta_{1,L^2}^{(2)} + d^2 \left(\delta_{X_2}^{(2)}\right)^2 \right).$$

The proofs of these lemmas are as follows.

*Proof of Lemma D.9.*

(a) This one follows directly from the construction of the partition and Induction Hypothesis D.1.

(b) First, we write

$$F(\boldsymbol{x}) = \alpha \|\boldsymbol{x}\| + \alpha \|\boldsymbol{x}\| (\bar{F}(\bar{\boldsymbol{x}}) - 1) = \alpha \|\boldsymbol{x}\| \pm \alpha \|\boldsymbol{x}\| \delta_{1,L^\infty}^{(2)} = \alpha \|\boldsymbol{x}\| \pm O\left(\frac{\delta_{1,L^\infty}^{(2)}}{|\bar{w}_2|}\right),$$

where the last equality comes from the fact $f$ vanishes on $\{\|\boldsymbol{x}\| \geq \Omega(-\bar{b}_2/(\alpha|\bar{w}_2|))\}$. Similarly, for any $(v_2, r_2) \in \mu_2$, we have

$$v_2 F(\boldsymbol{x}) + r_2 = \left\langle \begin{bmatrix} v_2 \\ r_2 \end{bmatrix}, \begin{bmatrix} F(\boldsymbol{x}) \\ 1 \end{bmatrix} \right\rangle = \left\langle \begin{bmatrix} \bar{w}_2 \\ \bar{b}_2 \end{bmatrix}, \begin{bmatrix} F(\boldsymbol{x}) \\ 1 \end{bmatrix} \right\rangle + \left\langle \begin{bmatrix} v_2 \\ r_2 \end{bmatrix} - \begin{bmatrix} \bar{w}_2 \\ \bar{b}_2 \end{bmatrix}, \begin{bmatrix} F(\boldsymbol{x}) \\ 1 \end{bmatrix} \right\rangle$$

$$= \bar{w}_2 F(\boldsymbol{x}) + \bar{b}_2 \pm O\left(\delta_2^{(2)} \sqrt{\alpha^2 \|\boldsymbol{x}\|^2 + 1}\right)$$

$$= \bar{w}_2 F(\boldsymbol{x}) + \bar{b}_2 \pm O\left(d^3 \delta_2^{(2)}\right).$$

Hence, for any $R > 0$ and $\boldsymbol{x} \in R\mathbb{S}^{d-1}$, we have

$$v_2 F(\boldsymbol{x}) + r_2 = \bar{w}_2 \alpha \|\boldsymbol{x}\| + \bar{b}_2 \underbrace{\pm O\left(\delta_{1,L^\infty}^{(2)}\right) \pm O\left(d^3 \delta_2^{(2)}\right)}_{=: \delta_{\text{Tmp}}}.$$

Therefore,

$$v_2 F(\boldsymbol{x}) + r_2 > 0, \quad \text{if } \|\boldsymbol{x}\| < \frac{\bar{b}_2 - \delta_{\text{Tmp}}}{-\bar{w}_2 \alpha} = \tilde{R} - \frac{\delta_{\text{Tmp}}}{-\bar{w}_2 \alpha},$$

$$v_2 F(\boldsymbol{x}) + r_2 < 0, \quad \text{if } \|\boldsymbol{x}\| > \frac{\bar{b}_2 + \delta_{\text{Tmp}}}{-\bar{w}_2 \alpha} = \tilde{R} + \frac{\delta_{\text{Tmp}}}{-\bar{w}_2 \alpha}.$$

In other words, $R_1 \geq \tilde{R} - \frac{\delta_{\text{Tmp}}}{-\bar{w}_2 \alpha}$ and $R_2 \leq \tilde{R} + \frac{\delta_{\text{Tmp}}}{-\bar{w}_2 \alpha}$. Thus,

$$R_2 - R_1 \leq \frac{\delta_{\text{Tmp}}}{-\bar{w}_2 \alpha} \leq O\left(\delta_{1,L^\infty}^{(2)} + O\left(d^3 \delta_2^{(2)}\right)\right) d^{4.5} = \delta_{X_2}.$$

To complete the proof, it suffices to invoke Lemma B.1.

(c) Note that by the definition of $R_2$, for any $\boldsymbol{x}_0 \in R_2 \mathbb{S}^{d-1}$, we have $f(\boldsymbol{x}_0) = 0$. Hence, for any $\boldsymbol{x} \in X_2$, there exists some $\boldsymbol{x}_0$ with $f(\boldsymbol{x}_0) = 0$ and $\|\boldsymbol{x} - \boldsymbol{x}_0\| \leq R_2 - R_1 = \delta_{X_2}^{(2)}$. Since $f$ is $O(1)$-Lipschitz, we have, for any $\boldsymbol{x} \in X_2$, $f(\boldsymbol{x}) = f(\boldsymbol{x}) - f(\boldsymbol{x}_0) \leq O(\delta_{X_2}^{(2)})$.

$\square$

*Proof of Lemma D.10.* Since both $f_*$ and $f$ vanishes on $X_3$, it suffices to consider $X_1$ and $X_2$. Recall that that all second layer neurons are activated on $X_1$. Hence,

$$\mathcal{L}_2\Big|_{X_1} := \frac{1}{2} \mathop{\mathbb{E}}_{X_1} \left\{ (\tilde{f}(\boldsymbol{x}) - f(\boldsymbol{x}))^2 \right\} = \frac{\bar{w}_2^2}{2} \mathop{\mathbb{E}}_{X_1} \left\{ (\tilde{F}(\boldsymbol{x}) - F(\boldsymbol{x}))^2 \right\},$$

$$\mathcal{L}_3\Big|_{X_1} := \mathop{\mathbb{E}}_{X_1} \left\{ (f_*(\boldsymbol{x}) - \tilde{f}(\boldsymbol{x}))(\tilde{f}(\boldsymbol{x}) - f(\boldsymbol{x})) \right\} = \bar{w}_2 \mathop{\mathbb{E}}_{X_1} \left\{ (f_*(\boldsymbol{x}) - \tilde{f}(\boldsymbol{x}))(\tilde{F}(\boldsymbol{x}) - F(\boldsymbol{x})) \right\} = 0,$$

where the last equality comes from Corollary B.4. Now, we bound the influence of $X_2$. Note that both $\nabla_{\boldsymbol{v}_1} f(\boldsymbol{x})$ and $\nabla_{\boldsymbol{v}_1} \tilde{f}(\boldsymbol{x})$ are bounded by $O(|\bar{w}_2| \|\boldsymbol{v}_1\| \|\boldsymbol{x}\|)$. Recall from Lemma D.9 that $f \leq O(\delta_{X_2}^{(2)})$ on $X_2$ and $\mathbb{P}[X_2] \leq O(\delta_{X_2}^{(2)})$. Therefore,

$$\left\| \nabla_{\boldsymbol{v}_1} \mathcal{L}_2 \Big|_{X_2} \right\| \leq O(\delta_{X_2}^{(2)}) \times O\left(\delta_{X_2}^{(2)}\right) \times O\left(|\bar{w}_2| \|\boldsymbol{v}_1\| \frac{1}{|\bar{w}_2| \alpha}\right) \leq O\left(\left(\delta_{X_2}^{(2)}\right)^2 \frac{1}{\alpha}\right) \|\boldsymbol{v}_1\|.$$

The proof for $\nabla_{\boldsymbol{v}_1} \mathcal{L}_3|_{X_2}$ is the same. $\square$

*Proof of Lemma D.11.* For fixed $\boldsymbol{x} \in \mathbb{R}^d$, we write

$$\frac{\mathrm{d}}{\mathrm{d}t}\bar{F}(\boldsymbol{x}) = \frac{\frac{\mathrm{d}}{\mathrm{d}t}F(\boldsymbol{x})}{\alpha} - \bar{F}(\boldsymbol{x})\frac{\dot{\alpha}}{\alpha} = -\frac{1}{\alpha}\mathop{\mathbb{E}}_{\boldsymbol{w}_1}\langle\nabla_{\boldsymbol{w}_1}F(\boldsymbol{x}),\nabla_{\boldsymbol{w}_1}\mathcal{L}\rangle + \bar{F}(\boldsymbol{x})\frac{1}{\alpha}\mathop{\mathbb{E}}_{\boldsymbol{w}_1}\langle\nabla_{\boldsymbol{w}_1}\alpha,\nabla_{\boldsymbol{w}_1}\mathcal{L}\rangle.$$

First, we consider $\mathcal{L}_1$. For each $\boldsymbol{v}_1 \in \mu_1$, we have

$$\nabla_{\boldsymbol{v}_1}\mathcal{L}_1 = -\mathop{\mathbb{E}}_{\boldsymbol{x}}\left\{(f_*(\boldsymbol{x}) - \tilde{f}(\boldsymbol{x}))\nabla_{\boldsymbol{v}_1}\tilde{f}(\boldsymbol{x})\right\}$$

$$= -\frac{2C_\Gamma}{\sqrt{d}}\mathop{\mathbb{E}}_{\boldsymbol{x}}\left\{(f_*(\boldsymbol{x}) - \tilde{f}(\boldsymbol{x}))\mathop{\mathbb{E}}_{w_2,b_2}\{\sigma(w_2\alpha\|\boldsymbol{x}\| + b_2)w_2\}\right\}\boldsymbol{v}_1 =: C_{\mathrm{Tmp},1}\boldsymbol{v}_1.$$

Meanwhile, note that

$$\langle\nabla_{\boldsymbol{v}_1}F(\boldsymbol{x}),\boldsymbol{v}_1\rangle = \left\langle\nabla_{\boldsymbol{v}_1}(\|\boldsymbol{v}_1\|^2\sigma(\bar{\boldsymbol{v}}_1\cdot\boldsymbol{x})),\boldsymbol{v}_1\right\rangle = \left\langle\nabla_{\boldsymbol{v}_1}(\|\boldsymbol{v}_1\|^2)\sigma(\bar{\boldsymbol{v}}_1\cdot\boldsymbol{x}),\boldsymbol{v}_1\right\rangle = 2\|\boldsymbol{v}_1\|^2\sigma(\bar{\boldsymbol{v}}_1\cdot\boldsymbol{x}),$$

$$\langle\nabla_{\boldsymbol{v}_1}\alpha,\boldsymbol{v}_1\rangle = \frac{C_\Gamma}{\sqrt{d}}\left\langle\nabla_{\boldsymbol{v}_1}\|\boldsymbol{v}_1\|^2,\boldsymbol{v}_1\right\rangle = \frac{2C_\Gamma}{\sqrt{d}}\|\boldsymbol{v}_1\|^2.$$

Hence,

$$\left.\frac{\mathrm{d}}{\mathrm{d}t}\bar{F}(\boldsymbol{x})\right|_{\mathcal{L}_1} := -\frac{1}{\alpha}\mathop{\mathbb{E}}_{\boldsymbol{w}_1}\langle\nabla_{\boldsymbol{w}_1}F(\boldsymbol{x}),\nabla_{\boldsymbol{w}_1}\mathcal{L}_1\rangle + \bar{F}(\boldsymbol{x})\frac{1}{\alpha}\mathop{\mathbb{E}}_{\boldsymbol{w}_1}\langle\nabla_{\boldsymbol{w}_1}\alpha,\nabla_{\boldsymbol{w}_1}\mathcal{L}_1\rangle$$

$$= -C_{\mathrm{Tmp},1}\frac{2}{\alpha}\mathop{\mathbb{E}}_{\boldsymbol{w}_1}\left\{\|\boldsymbol{w}_1\|^2\sigma(\bar{\boldsymbol{w}}_1\cdot\boldsymbol{x})\right\} + C_{\mathrm{Tmp},1}\bar{F}(\boldsymbol{x})\frac{1}{\alpha}\frac{2C_\Gamma}{\sqrt{d}}\mathop{\mathbb{E}}_{\boldsymbol{w}_1}\|\boldsymbol{w}_1\|^2$$

$$= -C_{\mathrm{Tmp},1}\frac{2}{\alpha}F(\boldsymbol{x}) + 2C_{\mathrm{Tmp},1}\bar{F}(\boldsymbol{x})$$

$$= 0.$$

Namely, $\mathcal{L}_1$ does not affect $\bar{F}$. Now we consider $\mathcal{L}_2$. By Lemma D.10, we have

$$\left.\frac{\mathrm{d}}{\mathrm{d}t}\bar{F}(\boldsymbol{x})\right|_{\mathcal{L}_2} := -\frac{1}{\alpha}\mathop{\mathbb{E}}_{\boldsymbol{w}_1}\langle\nabla_{\boldsymbol{w}_1}F(\boldsymbol{x}),\nabla_{\boldsymbol{w}_1}\mathcal{L}_2\rangle + \bar{F}(\boldsymbol{x})\frac{1}{\alpha}\mathop{\mathbb{E}}_{\boldsymbol{w}_1}\langle\nabla_{\boldsymbol{w}_1}\alpha,\nabla_{\boldsymbol{w}_1}\mathcal{L}_2\rangle$$

$$= -\frac{1}{\alpha}\frac{\bar{w}_2^2}{2}\mathop{\mathbb{E}}_{\boldsymbol{w}_1}\left\{\left\langle\nabla_{\boldsymbol{w}_1}F(\boldsymbol{x}),\nabla_{\boldsymbol{w}_1}\mathop{\mathbb{E}}_{\boldsymbol{x}'\in X_1}\left\{(\tilde{F}(\boldsymbol{x}') - F(\boldsymbol{x}'))^2\right\}\right\rangle\right\}$$

$$+ \frac{1}{\alpha}\frac{\bar{w}_2^2}{2}\bar{F}(\boldsymbol{x})\mathop{\mathbb{E}}_{\boldsymbol{w}_1}\left\{\left\langle\nabla_{\boldsymbol{w}_1}\alpha,\mathop{\mathbb{E}}_{\boldsymbol{x}'\in X_1}\left\{(\tilde{F}(\boldsymbol{x}') - F(\boldsymbol{x}'))^2\right\}\right\rangle\right\}$$

$$\pm O\left(\sqrt{d}\left(\delta_{X_2}^{(2)}\right)^2\frac{1}{\alpha}\right)\|\boldsymbol{x}\|.$$

Note that we can rewrite the $\nabla_{\boldsymbol{w}_1}F(\boldsymbol{x})$ in the first term as $(\nabla_{\boldsymbol{w}_1}\alpha)\bar{F}(\boldsymbol{x}) + \alpha\nabla_{\boldsymbol{w}_1}\bar{F}(\boldsymbol{x})$ so that part of it cancel with the second term. Then, we get

$$\left.\frac{\mathrm{d}}{\mathrm{d}t}\bar{F}(\boldsymbol{x})\right|_{\mathcal{L}_2} = -\frac{\bar{w}_2^2}{2}\mathop{\mathbb{E}}_{\boldsymbol{w}_1}\left\{\left\langle\nabla_{\boldsymbol{w}_1}\bar{F}(\boldsymbol{x}),\nabla_{\boldsymbol{w}_1}\mathop{\mathbb{E}}_{\boldsymbol{x}'\in X_1}\left\{(\tilde{F}(\boldsymbol{x}') - F(\boldsymbol{x}'))^2\right\}\right\rangle\right\} \pm O\left(\sqrt{d}\left(\delta_{X_2}^{(2)}\right)^2\frac{1}{\alpha}\right)\|\boldsymbol{x}\|.$$

For $\mathcal{L}_3$, we can simply merge it into the error term of $\frac{\mathrm{d}}{\mathrm{d}t}\bar{F}(\boldsymbol{x})|_{\mathcal{L}_2}$. $\qquad\square$

*Proof of Lemma D.12.* By Lemma D.11, we have

$$\frac{\mathrm{d}}{\mathrm{d}t}\left\|\bar{F} - \|\cdot\|\right\|_{L^2}^2 = \mathop{\mathbb{E}}_{\boldsymbol{x}}\left\{(\bar{F}(\boldsymbol{x}) - \|\boldsymbol{x}\|)\frac{\mathrm{d}}{\mathrm{d}t}F(\boldsymbol{x})\right\}$$

$$= -\frac{\bar{w}_2^2}{2}\mathop{\mathbb{E}}_{\boldsymbol{x}}\left\{(\bar{F}(\boldsymbol{x}) - \|\boldsymbol{x}\|)\mathop{\mathbb{E}}_{\boldsymbol{w}_1}\left\{\left\langle\nabla_{\boldsymbol{w}_1}\bar{F}(\boldsymbol{x}),\nabla_{\boldsymbol{w}_1}\mathop{\mathbb{E}}_{\boldsymbol{x}'\in X_1}\left\{(\tilde{F}(\boldsymbol{x}') - F(\boldsymbol{x}'))^2\right\}\right\rangle\right\}\right\}$$

$$\pm \mathop{\mathbb{E}}_{\boldsymbol{x}}\left\{(\bar{F}(\boldsymbol{x}) - \|\boldsymbol{x}\|)O\left(\sqrt{d}\left(\delta_{X_2}^{(2)}\right)^2\frac{1}{\alpha}\right)\|\boldsymbol{x}\|\right\}.$$

The second term can be bounded by $O\left(\delta_{1,L^2}^{(2)}\left(\delta_{X_2}^{(2)}\right)^2 d^5\right)$. The first term is equal to

$$\mathrm{Tmp} := -\frac{\bar{w}_2^2}{4}\mathop{\mathbb{E}}_{\boldsymbol{w}_1}\left\{\left\langle\nabla_{\boldsymbol{w}_1}\mathop{\mathbb{E}}_{\boldsymbol{x}}\{(\bar{F}(\boldsymbol{x}) - \|\boldsymbol{x}\|)^2\},\nabla_{\boldsymbol{w}_1}\mathop{\mathbb{E}}_{\boldsymbol{x}'\in X_1}\{(\alpha\|\boldsymbol{x}'\| - F(\boldsymbol{x}'))^2\}\right\rangle\right\}.$$

To complete the proof, it suffices to show that this is negative. For each $\boldsymbol{w}_1$, we have

$$\nabla_{\boldsymbol{w}_1} \mathop{\mathbb{E}}_{\boldsymbol{x}' \in X_1} \left\{ (\alpha \|\boldsymbol{x}'\| - F(\boldsymbol{x}'))^2 \right\} = \mathop{\mathbb{E}}_{\boldsymbol{x}' \in X_1} \left\{ (\bar{F}(\boldsymbol{x}') - \|\boldsymbol{x}'\|)^2 \right\} \nabla_{\boldsymbol{w}_1} \alpha^2 + \alpha^2 \mathop{\mathbb{E}}_{\boldsymbol{x}' \in X_1} \left\{ \nabla_{\boldsymbol{w}_1} (\bar{F}(\boldsymbol{x}') - \|\boldsymbol{x}'\|)^2 \right\}.$$

Since the distribution of $\boldsymbol{x}$ is spherically symmetric, $\mathbb{E}_{\boldsymbol{x}' \in X_1} \left\{ \nabla_{\boldsymbol{w}_1} (\bar{F}(\boldsymbol{x}') - \|\boldsymbol{x}'\|)^2 \right\}$ and $\mathbb{E}_{\boldsymbol{x}} \left\{ \nabla_{\boldsymbol{w}_1} (\bar{F}(\boldsymbol{x}) - \|\boldsymbol{x}\|)^2 \right\}$ have the same direction. Hence,

$$\begin{aligned}
\texttt{Tmp} &\leq -\frac{\bar{w}_2^2}{4} \mathop{\mathbb{E}}_{\boldsymbol{w}_1} \left\{ \left\langle \nabla_{\boldsymbol{w}_1} \mathop{\mathbb{E}}_{\boldsymbol{x}} \left\{ (\bar{F}(\boldsymbol{x}) - \|\boldsymbol{x}\|)^2 \right\}, \nabla_{\boldsymbol{w}_1} \alpha^2 \right\rangle \right\} \mathop{\mathbb{E}}_{\boldsymbol{x}' \in X_1} \left\{ (\bar{F}(\boldsymbol{x}') - \|\boldsymbol{x}'\|)^2 \right\} \\
&= -\frac{C_\Gamma}{\sqrt{d}} \bar{w}_2^2 \alpha \mathop{\mathbb{E}}_{\boldsymbol{x}' \in X_1} \left\{ (\bar{F}(\boldsymbol{x}') - \|\boldsymbol{x}'\|)^2 \right\} \mathop{\mathbb{E}}_{\boldsymbol{x}} \left\{ \mathop{\mathbb{E}}_{\boldsymbol{w}_1} \left\{ \left\langle \nabla_{\boldsymbol{w}_1} (\bar{F}(\boldsymbol{x}) - \|\boldsymbol{x}\|)^2, \boldsymbol{w}_1 \right\rangle \right\} \right\}.
\end{aligned}$$

Then, we compute

$$\begin{aligned}
\left\langle \nabla_{\boldsymbol{w}_1} (\bar{F}(\boldsymbol{x}) - \|\boldsymbol{x}\|)^2, \boldsymbol{w}_1 \right\rangle &= 2(\bar{F}(\boldsymbol{x}) - \|\boldsymbol{x}\|) \left\langle \frac{\nabla_{\boldsymbol{w}_1} F(\boldsymbol{x})}{\alpha} - \bar{F}(\boldsymbol{x}) \frac{\nabla_{\boldsymbol{w}_1} \alpha}{\alpha}, \boldsymbol{w}_1 \right\rangle \\
&= 2(\bar{F}(\boldsymbol{x}) - \|\boldsymbol{x}\|) \left( \frac{2 \|\boldsymbol{w}_1\|^2 \sigma(\bar{\boldsymbol{w}}_1 \cdot \boldsymbol{x})}{\alpha} - \bar{F}(\boldsymbol{x}) \frac{1}{\alpha} \frac{2 C_\Gamma}{\sqrt{d}} \|\boldsymbol{w}_1\|^2 \right).
\end{aligned}$$

Take expectation over $\boldsymbol{w}_1$ and one can see that this is 0. Thus, $\texttt{Tmp} \leq 0$. $\qquad\square$

*Proof of Lemma D.13.* Recall from Lemma D.11 that

$$\frac{\mathrm{d}}{\mathrm{d}t} \bar{F}(\boldsymbol{x}) = -\frac{\bar{w}_2^2}{2} \mathop{\mathbb{E}}_{\boldsymbol{w}_1} \left\{ \left\langle \nabla_{\boldsymbol{w}_1} \bar{F}(\boldsymbol{x}), \nabla_{\boldsymbol{w}_1} \mathop{\mathbb{E}}_{\boldsymbol{x}' \in X_1} \left\{ (\tilde{F}(\boldsymbol{x}') - F(\boldsymbol{x}'))^2 \right\} \right\rangle \right\} \pm O\left( \sqrt{d} \left( \delta_{X_2}^{(2)} \right)^2 \frac{1}{\alpha} \right) \|\boldsymbol{x}\|.$$

For the first term, we have

$$\left\| \nabla_{\boldsymbol{w}_1} \bar{F}(\boldsymbol{x}) \right\| \leq \left\| \frac{\nabla_{\boldsymbol{w}_1} F(\boldsymbol{x})}{\alpha} \right\| + \left\| \bar{F}(\boldsymbol{x}) \frac{\dot{\alpha}}{\alpha} \right\| \leq O\left( \frac{\|\boldsymbol{w}_1\| \|\boldsymbol{x}\|}{\alpha} \right),$$

$$\begin{aligned}
\left\| \nabla_{\boldsymbol{w}_1} \mathop{\mathbb{E}}_{\boldsymbol{x}' \in X_1} \left\{ \left( \tilde{F}(\boldsymbol{x}') - F(\boldsymbol{x}') \right)^2 \right\} \right\| &\leq \mathop{\mathbb{E}}_{\boldsymbol{x}' \in X_1} \left\{ \left| \tilde{F}(\boldsymbol{x}') - F(\boldsymbol{x}') \right| \left( \left\| \nabla_{\boldsymbol{w}_1} \tilde{F}(\boldsymbol{x}') \right\| + \left\| \nabla_{\boldsymbol{w}_1} F(\boldsymbol{x}') \right\| \right) \right\} \\
&\leq O(1) \mathop{\mathbb{E}}_{\boldsymbol{x}' \in X_1} \left\{ \left| \tilde{F}(\boldsymbol{x}') - F(\boldsymbol{x}') \right| \|\boldsymbol{x}'\| \right\} \|\boldsymbol{w}_1\| \\
&\leq O\left( \delta_{1,L^2}^{(2)} \frac{1}{\sqrt{\alpha |\bar{w}_2|}} \|\boldsymbol{w}_1\| \right).
\end{aligned}$$

Thus,

$$\begin{aligned}
\left| \frac{\mathrm{d}}{\mathrm{d}t} \bar{F}(\boldsymbol{x}) \right| &\leq O\left( \bar{w}_2^2 \mathop{\mathbb{E}}_{\boldsymbol{w}_1} \left\{ \frac{\|\boldsymbol{w}_1\| \|\boldsymbol{x}\|}{\alpha} \delta_{1,L^2}^{(2)} \frac{1}{\sqrt{\alpha |\bar{w}_2|}} \|\boldsymbol{w}_1\| \right\} \right) + O\left( \sqrt{d} \left( \delta_{X_2}^{(2)} \right)^2 \frac{1}{\alpha} \right) \|\boldsymbol{x}\| \\
&\leq O\left( \frac{\sqrt{d} |\bar{w}_2|^{1.5}}{\sqrt{\alpha}} \delta_{1,L^2}^{(2)} \right) \|\boldsymbol{x}\| + O\left( \sqrt{d} \left( \delta_{X_2}^{(2)} \right)^2 \frac{1}{\alpha} \right) \|\boldsymbol{x}\| \\
&\leq O\left( d^3 \delta_{1,L^2}^{(2)} + d^2 \left( \delta_{X_2}^{(2)} \right)^2 \right) \|\boldsymbol{x}\|.
\end{aligned}$$

$\qquad\square$

### D.2.2 SPREAD OF THE SECOND LAYER

**Lemma D.14.** *Suppose that Induction Hypothesis D.1 is true at time $t$. Then for any $(v_2, r_2), (v'_2, r'_2) \in \mu_2$, $\frac{\mathrm{d}}{\mathrm{d}t} \|(v_2, r_2) - (v'_2, r'_2)\|^2 \leq 0$. In words, the spread of the second layer never grows.*

*Proof.* Let $(v_2, r_2), (v_2', r_2') \in \mu_2$ be two second layer neurons. For notational convenience, we define $h_2(\boldsymbol{x}) = v_2 F(\boldsymbol{x}) + r_2$ and $h_2'(\boldsymbol{x}) = v_2' F(\boldsymbol{x}) + r_2'$. We have

$$
\frac{1}{2} \frac{\mathrm{d}}{\mathrm{d}t} \left( (v_2 - v_2')^2 + (r_2 - r_2')^2 \right)
$$
$$
= (v_2 - v_2') \mathop{\mathbb{E}}_{\boldsymbol{x}} \left\{ (f_*(\boldsymbol{x}) - f(\boldsymbol{x})) F(\boldsymbol{x}) \left( \sigma'(h_2(\boldsymbol{x})) - \sigma'(h_2'(\boldsymbol{x})) \right) \right\}
$$
$$
+ (r_2 - r_2') \mathop{\mathbb{E}}_{\boldsymbol{x}} \left\{ (f_*(\boldsymbol{x}) - f(\boldsymbol{x})) \left( \sigma'(h_2(\boldsymbol{x})) - \sigma'(h_2'(\boldsymbol{x})) \right) \right\}
$$
$$
= \mathop{\mathbb{E}}_{\boldsymbol{x}} \left\{ (f_*(\boldsymbol{x}) - f(\boldsymbol{x})) (h_2(\boldsymbol{x}) - h_2'(\boldsymbol{x})) \left( \sigma'(h_2(\boldsymbol{x})) - \sigma'(h_2'(\boldsymbol{x})) \right) \right\}.
$$

By Lemma D.9, $\sigma'(h_2(\boldsymbol{x})) - \sigma'(h_2'(\boldsymbol{x})) = 0$ for all $\boldsymbol{x}$ with $\|\boldsymbol{x}\| \le 1$. Hence,

$$
\frac{1}{2} \frac{\mathrm{d}}{\mathrm{d}t} \left( (v_2 - v_2')^2 + (r_2 - r_2')^2 \right)
$$
$$
= \mathop{\mathbb{E}}_{\boldsymbol{x}: \|\boldsymbol{x}\| > 1} \left\{ (f_*(\boldsymbol{x}) - f(\boldsymbol{x})) (h_2(\boldsymbol{x}) - h_2'(\boldsymbol{x})) \left( \sigma'(h_2(\boldsymbol{x})) - \sigma'(h_2'(\boldsymbol{x})) \right) \right\}
$$
$$
= - \mathop{\mathbb{E}}_{\boldsymbol{x}: \|\boldsymbol{x}\| > 1} \left\{ f(\boldsymbol{x}) (h_2(\boldsymbol{x}) - h_2'(\boldsymbol{x})) \left( \sigma'(h_2(\boldsymbol{x})) - \sigma'(h_2'(\boldsymbol{x})) \right) \right\}.
$$

Note that $f \ge 0$ and, since $\sigma'$ is non-decreasing, $(h_2(\boldsymbol{x}) - h_2'(\boldsymbol{x})) (\sigma'(h_2(\boldsymbol{x})) - \sigma'(h_2'(\boldsymbol{x}))) \ge 0$. Thus, $\frac{1}{2} \frac{\mathrm{d}}{\mathrm{d}t} \left( (v_2 - v_2')^2 + (r_2 - r_2')^2 \right) \le 0$. $\qquad\square$

### D.2.3 REGULARITY CONDITIONS

As we have mentioned earlier, we will mainly use the continuity argument to maintain the regularity conditions, so the problem can be reduced into estimating the derivative on the boundary. As an example, suppose that $\bar{b}_2 = 1 - \delta$ for some small $\delta > 0$. Then by Lemma D.15, which upper bounds the loss using $1 - \bar{b}_2$ and $-1 - \bar{w}_2 \alpha$, we know $|-1 - \bar{w}_2 \alpha|$ must be large, otherwise we would have $\mathcal{L} < \varepsilon$. Then, we can use the fact that $|-1 - \bar{w}_2 \alpha|$ is large to estimate the derivative. The proof for the other regularity conditions is similar except the proof for $|\bar{w}_2|$, which is in the same spirit with the ones for first layer errors.

**Lemma D.15.** *Suppose that Induction Hypothesis D.1 is true at time $t$. Then we have*

$$
\mathcal{L} \le O \left( (1 - \bar{b}_2)^2 + \frac{(-1 - \bar{w}_2 \alpha)^2}{\bar{w}_2^2 \alpha^2} + \left( \delta_{1, L^\infty}^{(2)} + d^3 \delta_2^{(2)} \right)^2 \right).
$$

**Lemma D.16.** *Suppose that Induction Hypothesis D.1 is true at time $t$ and $\bar{b}_2 = 1 - \Theta(\sqrt{\varepsilon})$. Then, $\frac{\mathrm{d}}{\mathrm{d}t} \bar{b}_2 < 0$.*

**Lemma D.17.** *Suppose that Induction Hypothesis D.1 is true at time $t$ and $\bar{w}_2 \alpha = -1 + \Theta(\sqrt{\varepsilon})$. Then we have $\frac{\mathrm{d}}{\mathrm{d}t} (\bar{w}_2 \alpha) > 0$.*

**Lemma D.18.** *Suppose that Induction Hypothesis D.1 is true throughout Stage 2. Then $|\bar{w}_2| \le d$.*

**Lemma D.19.** *Suppose that Induction Hypothesis D.1 is true throughout Stage 2. Then $|\bar{w}_2| \ge \Theta(1/d^3)$ and $\alpha \ge \Theta(1/d^{1.5})$.*

The proofs of this subsubsection are gathered bellow.

*Proof of Lemma D.15.* For any $\boldsymbol{x} \in \mathbb{R}^d$, by Lemma D.7 and the Lipschitzness of $\sigma$, we have, for any $\boldsymbol{x} \in X_1 \cup X_2$,

$$
f(\boldsymbol{x}) = \sigma(\bar{w}_2 \alpha \bar{F}(\boldsymbol{x}) + \bar{b}_2) \pm O \left( d^3 \delta_2^{(2)} \right)
$$
$$
= \sigma(1 - \|\boldsymbol{x}\|) \pm |1 - \bar{b}_2| \pm \left| -\|\boldsymbol{x}\| - \bar{w}_2 \alpha \bar{F}(\boldsymbol{x}) \right| \pm O \left( d^3 \delta_2^{(2)} \right).
$$

By Induction Hypothesis D.1, for any $\boldsymbol{x} \in X_1 \cup X_2$, we have

$$
\left| -\|\boldsymbol{x}\| - \bar{w}_2 \alpha \bar{F}(\boldsymbol{x}) \right| = \left| -1 - \bar{w}_2 \alpha \bar{F}(\bar{\boldsymbol{x}}) \right| \|\boldsymbol{x}\|
$$
$$
\le |-1 - \bar{w}_2 \alpha| \|\boldsymbol{x}\| + \left| 1 - \bar{F}(\bar{\boldsymbol{x}}) \right| |\bar{w}_2| \alpha \|\boldsymbol{x}\| \le O \left( \frac{|-1 - \bar{w}_2 \alpha|}{|\bar{w}_2 \alpha|} \right) + O \left( \delta_{1, L^\infty}^{(2)} \right).
$$

Therefore,

$$f(\boldsymbol{x}) = f_*(\boldsymbol{x}) \pm |1 - \bar{b}_2| \pm O\left(\frac{|-1 - \bar{w}_2\alpha|}{|\bar{w}_2\alpha|}\right) \pm O\left(\delta_{1,L^\infty}^{(2)} + d^3\delta_2^{(2)}\right).$$

Thus,

$$\mathcal{L} = \frac{1}{2}\mathop{\mathbb{E}}_{\boldsymbol{x}}\left\{(f_*(\boldsymbol{x}) - f(\boldsymbol{x}))^2\right\} \le \frac{1}{2}\left(|1 - \bar{b}_2| + O\left(\frac{|-1 - \bar{w}_2\alpha|}{|\bar{w}_2\alpha|}\right) + O\left(\delta_{1,L^\infty}^{(2)} + d^3\delta_2^{(2)}\right)\right)^2$$

$$\le O\left((1 - \bar{b}_2)^2 + \frac{(-1 - \bar{w}_2\alpha)^2}{\bar{w}_2^2\alpha^2} + \left(\delta_{1,L^\infty}^{(2)} + d^3\delta_2^{(2)}\right)^2\right).$$

$\square$

*Proof of Lemma D.16.* By Lemma D.7, for any $(v_2, r_2) \in \mu_2$, we have

$$\dot{r}_2 = \mathop{\mathbb{E}}_{\boldsymbol{x}}\left\{f_*(\boldsymbol{x}) - \sigma(\bar{w}_2 F(\boldsymbol{x}) + \bar{b}_2)\right\} \pm O\left(d^3\delta_2^{(2)}\right).$$

Then, by Induction Hypothesis D.1 and the Lipschitzness of $\sigma$, we have

$$\sigma(\bar{w}_2 F(\boldsymbol{x}) + \bar{b}_2) = \sigma(\bar{w}_2\alpha\|\boldsymbol{x}\|\bar{F}(\bar{\boldsymbol{x}}) + \bar{b}_2) = \sigma(\bar{w}_2\alpha\|\boldsymbol{x}\| + \bar{b}_2) \pm O\left(\delta_{1,L^\infty}^{(2)}\right).$$

Therefore,

$$\dot{\bar{b}}_2 = \mathop{\mathbb{E}}_{\boldsymbol{x}}\left\{f_*(\boldsymbol{x}) - \sigma(\bar{w}_2\alpha\|\boldsymbol{x}\| + \bar{b}_2)\right\} \pm O\left(\delta_{1,L^\infty}^{(2)} + d^3\delta_2^{(2)}\right).$$

Since $\mathcal{L} \ge \varepsilon$, by Lemma D.15, we have

$$\frac{(-1 - \bar{w}_2\alpha)^2}{\bar{w}_2^2\alpha^2} \ge \Omega(\varepsilon) - O(\delta^2) - O\left(\delta_{1,L^\infty}^{(2)} + d^3\delta_2^{(2)}\right)^2 \ge \Omega(\varepsilon).$$

Since $\bar{w}_2\alpha \ge -1$, this implies $\bar{w}_2\alpha \ge -1 + \Omega\left(|\bar{w}_2|\alpha\sqrt{\varepsilon}\right)$. In fact, this implies $\bar{w}_2\alpha \ge -1 + \Omega(\sqrt{\varepsilon})$ even when $|\bar{w}_2|\alpha$ is $o(1)$, as, in that case, $\bar{w}_2\alpha \ge -1 + \Omega(\sqrt{\varepsilon})$ directly holds. Hence,

$$\sigma(\bar{w}_2\alpha\|\boldsymbol{x}\| + \bar{b}_2) \ge \sigma\left((-1 + \Omega\left(\sqrt{\varepsilon}\right))\|\boldsymbol{x}\| + 1 - \delta\right) = \sigma\left(1 - \|\boldsymbol{x}\| + \Omega\left(\sqrt{\varepsilon}\|\boldsymbol{x}\|\right) - \delta\right).$$

Thus,

$$\dot{\bar{b}}_2 = \mathop{\mathbb{E}}_{\boldsymbol{x}}\left\{f_*(\boldsymbol{x}) - \sigma\left(1 - \|\boldsymbol{x}\| + \Omega\left(\sqrt{\varepsilon}\|\boldsymbol{x}\|\right) - \delta\right)\right\} \pm O\left(\delta_{1,L^\infty}^{(2)} + d^3\delta_2^{(2)}\right)$$

$$\le \mathop{\mathbb{E}}_{\|\boldsymbol{x}\|\le}\left\{1 - \|\boldsymbol{x}\| - \left(1 - \|\boldsymbol{x}\| + \Omega\left(\sqrt{\varepsilon}\|\boldsymbol{x}\|\right) - \delta\right)\right\} + O\left(\delta_{1,L^\infty}^{(2)} + d^3\delta_2^{(2)}\right)$$

$$= -\Omega\left(\sqrt{\varepsilon}\right) + \delta + O\left(\delta_{1,L^\infty}^{(2)} + d^3\delta_2^{(2)}\right).$$

As long as the constant in $\delta = \Theta(\sqrt{\varepsilon})$ is sufficiently small, this implies $\dot{\bar{b}}_2 < 0$ when $\bar{b}_2 = 1 - \delta$. $\square$

*Proof of Lemma D.17.* By Lemma D.3 and Lemma D.7, we have

$$\frac{\mathrm{d}}{\mathrm{d}t}(\bar{w}_2\alpha) = \mathop{\mathbb{E}}_{\boldsymbol{x}}\left\{(f_*(\boldsymbol{x}) - \sigma(\bar{w}_2\alpha\|\boldsymbol{x}\|\bar{F}(\bar{\boldsymbol{x}}) + \bar{b}_2))F(\boldsymbol{x})\right\}\left(\alpha + \frac{4C_\Gamma\bar{w}_2^2}{\sqrt{d}}\right)$$

$$\pm O\left(d^3\log(d)\delta_2^{(2)}\right)\alpha\left(\alpha + \frac{4C_\Gamma\bar{w}_2^2}{\sqrt{d}}\right).$$

Now we estimate the coefficient of the first term. Suppose that $\bar{w}_2\alpha = -1 + \delta$ for some $\delta \le \Theta(\sqrt{\varepsilon})$ with a sufficiently small constant. Then, by Lemma D.15, we have $(1 - \bar{b}_2)^2 \ge \Omega(\varepsilon) - O(\delta^2) = \Omega(\varepsilon)$. Hence, $\bar{b}_2 \le 1 - \Theta(\sqrt{\varepsilon})$. Also note that $\bar{w}_2\alpha = \Theta(1)$ implies that it suffices to consider $\boldsymbol{x}$ with $\|\boldsymbol{x}\| = \Theta(1)$. As a result, we have

$$\sigma(\bar{w}_2\alpha\|\boldsymbol{x}\|\bar{F}(\bar{\boldsymbol{x}}) + \bar{b}_2) = \sigma(\bar{w}_2\alpha\|\boldsymbol{x}\| + \bar{b}_2) \pm O\left(\delta_{1,L^\infty}^{(2)}\right)$$

$$\le \sigma\left(1 - \|\boldsymbol{x}\| - \Theta(\sqrt{\varepsilon})\right) + O\left(\delta_{1,L^\infty}^{(2)}\right).$$

Then, we decompose the coefficient as

$$
\mathbb{E}_{\boldsymbol{x}} \left\{ \left( f_*(\boldsymbol{x}) - \sigma(\bar{w}_2 \alpha \|\boldsymbol{x}\| \bar{F}(\bar{\boldsymbol{x}}) + \bar{b}_2) \right) F(\boldsymbol{x}) \right\} = \mathbb{E}_{\boldsymbol{x}} \left\{ \left( f_*(\boldsymbol{x}) - \sigma(\bar{w}_2 \alpha \|\boldsymbol{x}\| \bar{F}(\bar{\boldsymbol{x}}) + \bar{b}_2) \right) F(\boldsymbol{x}) \right\}
$$
$$
\geq \mathbb{E}_{\|\boldsymbol{x}\| \leq 1} \left\{ \left( \Theta(\sqrt{\varepsilon}) - O\left( \delta_{1,L^\infty}^{(2)} \right) \right) F(\boldsymbol{x}) \right\}
$$
$$
\geq \Omega\left( \alpha \sqrt{\varepsilon} \right).
$$

Thus,

$$
\frac{\mathrm{d}}{\mathrm{d}t}(\bar{w}_2 \alpha) \geq \left( \Omega(\sqrt{\varepsilon}) - O\left( d^3 \delta_2^{(2)} \right) \log(d) \right) \alpha \left( \alpha + \frac{4C_\Gamma \bar{w}_2^2}{\sqrt{d}} \right) > 0.
$$

$\square$

*Proof of Lemma D.18.* By Lemma D.3 and Lemma D.7, we have

$$
\dot{w}_2 = \mathbb{E}_{\boldsymbol{x}} \left\{ (f_*(\boldsymbol{x}) - f(\boldsymbol{x})) F(\boldsymbol{x}) \right\} \pm \left( d^3 \log d \delta_2^{(2)} \right)
$$
$$
\dot{\alpha} = \frac{4C_\Gamma}{\sqrt{d}} \mathbb{E}_{\boldsymbol{x}} \left\{ (f_*(\boldsymbol{x}) - f(\boldsymbol{x})) F(\boldsymbol{x}) \right\} \bar{w}_2 \pm O\left( d^{2.5} (\log d) \delta_2^{(2)} \right)
$$

As a result,

$$
\left| \frac{\mathrm{d}}{\mathrm{d}t} \left( \alpha - \frac{2C_\Gamma}{\sqrt{d}} \bar{w}_2^2 \right) \right| \leq O\left( d^4 \delta_2^{(2)} \right).
$$

Also recall that $\bar{w}_2^2 \ll \alpha$ at $T_1$. Thus, throughout Stage 2, we always have $\left| \alpha - \frac{2C_\Gamma}{\sqrt{d}} \bar{w}_2^2 \right| \ll 1/d$. Since $|\bar{w}_2 \alpha| \leq 1$, this implies $|\bar{w}_2| \leq O(d^{1/6}) \leq d$. $\square$

*Proof of Lemma D.19.* Recall from the proof of Lemma D.18 that $|\alpha - \frac{2C_\Gamma}{\sqrt{d}} \bar{w}_2^2| \ll 1/d$. Hence, when $\alpha = \Theta(1/d^{1.5})$, we have $|\bar{w}_2| \leq O(1/d)$. The estimations in Stage 1, *mutatis mutandis*, show that both $\alpha$ and $|\bar{w}_2|$ will grow in this case. $\square$

## D.3 CONVERGENCE RATE

Recall from Lemma D.5 that $\frac{\mathrm{d}}{\mathrm{d}t} \mathcal{L} = -\mathbb{E}_{w_2, b_2, \boldsymbol{w}_1} \|\nabla_{w_2, b_2, \boldsymbol{w}_1}\|^2$, where

$$
\nabla_{w_2, b_2, \boldsymbol{w}_1} := \mathbb{E}_{\boldsymbol{x}} \left\{ (f_*(\boldsymbol{x}) - f(\boldsymbol{x})) \begin{bmatrix} \sigma'(w_2 F(\boldsymbol{x}) + b_2) F(\boldsymbol{x}) \\ \sigma'(w_2 F(\boldsymbol{x}) + b_2) \\ 2\bar{W}_2(\boldsymbol{x})\sigma(\boldsymbol{w}_1 \cdot \boldsymbol{x}) \\ \|\boldsymbol{w}_1\| \bar{W}_2(\boldsymbol{x})\sigma'(\boldsymbol{w}_1 \cdot \boldsymbol{x})(\boldsymbol{I} - \bar{\boldsymbol{w}}_1 \bar{\boldsymbol{w}}_1^\top)\boldsymbol{x} \end{bmatrix} \right\}.
$$

**Lemma D.20.** *Suppose that Induction Hypothesis D.1 is true at time $t$. Then we have*

$$
\frac{\mathrm{d}}{\mathrm{d}t} \mathcal{L} \leq -\left\| \tilde{\nabla} \right\|^2 + O\left( \left( \delta_{1,L^2}^{(2)} + d^3 \delta_2^{(2)} \right) d^4 \right),
$$

*where*

$$
\tilde{\nabla} := \mathbb{E}_{\boldsymbol{x}} \left\{ (f_*(\boldsymbol{x}) - f(\boldsymbol{x})) \begin{bmatrix} \|\boldsymbol{x}\| \sqrt{\alpha^2 + \frac{4C_\Gamma}{\sqrt{d}} \bar{w}_2^2 \alpha} \\ 1 \end{bmatrix} \right\}.
$$

**Lemma D.21.** *Suppose that Induction Hypothesis D.1 is true at time $t$. Then we have*

$$
\left\| \tilde{\nabla} \right\| \geq \Omega(\alpha \mathcal{L}) - O\left( \delta_{1,L^2}^{(2)} + d^3 \delta_2^{(2)} \right).
$$

**Lemma D.22** (Stage 2). *Suppose that Induction Hypothesis D.1 is true throughout Stage 2. Then $T_2 - T_1 \leq O(d^3/\varepsilon)$.*

*Proof of Lemma D.20.* Since it is the norm of $\nabla_{w_2, b_2, \boldsymbol{w}_1}$, we can safely ignore the last entry and only consider the first three entries. By Lemma D.7, we have

$$
[\nabla_{w_2, b_2, \boldsymbol{w}_1}]_{1:3} = \mathbb{E}_{\boldsymbol{x}} \left\{ (f_*(\boldsymbol{x}) - f(\boldsymbol{x})) \begin{bmatrix} F(\boldsymbol{x}) \\ 1 \\ 2\bar{w}_2 \sigma(\boldsymbol{w}_1 \cdot \boldsymbol{x}) \end{bmatrix} \right\} \pm O\left( d^3 \delta_2^{(2)} \begin{bmatrix} \alpha \log(d) \\ 1 \\ \bar{w}_2 \|\boldsymbol{w}_1\| \log(d) \end{bmatrix} \right).
$$

Furthermore, we have

$$
\mathbb{E}_{\boldsymbol{x}}\left\{(f_*(\boldsymbol{x}) - f(\boldsymbol{x}))F(\boldsymbol{x})\right\} = \mathbb{E}_{\boldsymbol{x}}\left\{(f_*(\boldsymbol{x}) - f(\boldsymbol{x}))\alpha\left\|\boldsymbol{x}\right\|\right\} + \mathbb{E}_{\boldsymbol{x}}\left\{(f_*(\boldsymbol{x}) - f(\boldsymbol{x}))\alpha(\bar{F}(\boldsymbol{x}) - \left\|\boldsymbol{x}\right\|)\right\}
$$

$$
= \mathbb{E}_{\boldsymbol{x}}\left\{(f_*(\boldsymbol{x}) - f(\boldsymbol{x}))\alpha\left\|\boldsymbol{x}\right\|\right\} + O\left(\alpha\delta_{1,L^2}^{(2)}\right).
$$

Meanwhile, for $[\nabla_{w_2,b_2,\boldsymbol{w}_1}]_3$, by Lemma B.3 and Lemma D.8, we have

$$
2\bar{w}_2\mathbb{E}_{\boldsymbol{x}}\left\{(f_*(\boldsymbol{x}) - f(\boldsymbol{x}))\sigma(\boldsymbol{w}_1\cdot\boldsymbol{x})\right\}
$$

$$
= 2\bar{w}_2\mathbb{E}_{\boldsymbol{x}}\left\{(f_*(\boldsymbol{x}) - \tilde{f}(\boldsymbol{x}))\sigma(\boldsymbol{w}_1\cdot\boldsymbol{x})\right\} + 2\bar{w}_2\mathbb{E}_{\boldsymbol{x}}\left\{(\tilde{f}(\boldsymbol{x}) - f(\boldsymbol{x}))\sigma(\boldsymbol{w}_1\cdot\boldsymbol{x})\right\}
$$

$$
= \frac{2C_\Gamma\bar{w}_2}{\sqrt{d}}\mathbb{E}_{\boldsymbol{x}}\left\{(f_*(\boldsymbol{x}) - \tilde{f}(\boldsymbol{x}))\left\|\boldsymbol{x}\right\|\right\}\left\|\boldsymbol{w}_1\right\| \pm 2\bar{w}_2\left\|\boldsymbol{w}_1\right\|\left\|f - \tilde{f}\right\|_{L^2}\sqrt{\mathbb{E}_{\boldsymbol{x}\in X_2}\left\|\boldsymbol{x}\right\|^2}
$$

$$
= \frac{2C_\Gamma\bar{w}_2}{\sqrt{d}}\mathbb{E}_{\boldsymbol{x}}\left\{(f_*(\boldsymbol{x}) - \tilde{f}(\boldsymbol{x}))\left\|\boldsymbol{x}\right\|\right\}\left\|\boldsymbol{w}_1\right\| \pm O\left(|\bar{w}_2|^{1.5}\alpha^{0.5}\left\|\boldsymbol{w}_1\right\|\delta_{1,L^2}^{(2)}\right).
$$

Repeat the above procedure and we can replace the $\tilde{f}$ in the first term with $f$. Therefore,

$$
[\nabla_{w_2,b_2,\boldsymbol{w}_1}]_{1:3} = \mathbb{E}_{\boldsymbol{x}}\left\{(f_*(\boldsymbol{x}) - f(\boldsymbol{x}))\begin{bmatrix}\alpha\left\|\boldsymbol{x}\right\| \\ 1 \\ \frac{2C_\Gamma\bar{w}_2}{\sqrt{d}}\left\|\boldsymbol{x}\right\|\left\|\boldsymbol{w}_1\right\|\end{bmatrix}\right\}
$$

$$
\pm O\left(\delta_{1,L^2}^{(2)}\begin{bmatrix}\alpha \\ 0 \\ |\bar{w}_2|^{1.5}\alpha^{0.5}\left\|\boldsymbol{w}_1\right\|\end{bmatrix}\right) \pm O\left(d^3\delta_2^{(2)}\begin{bmatrix}\alpha\log(d) \\ 1 \\ \bar{w}_2\left\|\boldsymbol{w}_1\right\|\log(d)\end{bmatrix}\right)
$$

$$
= \mathbb{E}_{\boldsymbol{x}}\left\{(f_*(\boldsymbol{x}) - f(\boldsymbol{x}))\begin{bmatrix}\alpha\left\|\boldsymbol{x}\right\| \\ 1 \\ \frac{2C_\Gamma\bar{w}_2}{\sqrt{d}}\left\|\boldsymbol{x}\right\|\left\|\boldsymbol{w}_1\right\|\end{bmatrix}\right\}
$$

$$
\pm O\left(\left(\delta_{1,L^2}^{(2)} + d^3\delta_2^{(2)}\right)\begin{bmatrix}\alpha\log(d) \\ 1 \\ |\bar{w}_2|\left\|\boldsymbol{w}_1\right\|\log(d)\end{bmatrix}\right).
$$

Now, we estimate the the expected norm of $[\nabla_{w_2,b_2,\boldsymbol{w}_1}]_{1:3}$. First, we have

$$
[\nabla_{w_2,b_2,\boldsymbol{w}_1}]_1^2 = \left(\mathbb{E}_{\boldsymbol{x}}\left\{(f_*(\boldsymbol{x}) - f(\boldsymbol{x}))\left\|\boldsymbol{x}\right\|\right\}\right)^2\alpha^2 \pm O\left(\left(\delta_{1,L^2}^{(2)} + d^3\delta_2^{(2)}\right)\alpha^2\log(d)\right),
$$

$$
[\nabla_{w_2,b_2,\boldsymbol{w}_1}]_2^2 = \left(\mathbb{E}_{\boldsymbol{x}}\left\{f_*(\boldsymbol{x}) - f(\boldsymbol{x})\right\}\right)^2 \pm O\left(\delta_{1,L^2}^{(2)} + d^3\delta_2^{(2)}\right).
$$

For $[\nabla_{w_2,b_2,\boldsymbol{w}_1}]_3$, we have

$$
\mathbb{E}_{\boldsymbol{w}_1}[\nabla_{w_2,b_2,\boldsymbol{w}_1}]_2^3 = \frac{4C_\Gamma^2\bar{w}_2^2}{d}\left(\mathbb{E}_{\boldsymbol{x}}\left\{(f_*(\boldsymbol{x}) - f(\boldsymbol{x}))\left\|\boldsymbol{x}\right\|\right\}\right)^2\mathbb{E}_{\boldsymbol{w}_1}\left\|\boldsymbol{w}_1\right\|^2
$$

$$
\pm O\left(\left(\delta_{1,L^2}^{(2)} + d^3\delta_2^{(2)}\right)\bar{w}_2^2\frac{\mathbb{E}\left\|\boldsymbol{w}\right\|_1^2}{\sqrt{d}}\log(d)\right)
$$

$$
= \left(\mathbb{E}_{\boldsymbol{x}}\left\{(f_*(\boldsymbol{x}) - f(\boldsymbol{x}))\left\|\boldsymbol{x}\right\|\right\}\right)^2\frac{4C_\Gamma\bar{w}_2^2}{\sqrt{d}}\alpha
$$

$$
\pm O\left(\left(\delta_{1,L^2}^{(2)} + d^3\delta_2^{(2)}\right)\bar{w}_2^2\alpha\log(d)\right).
$$

Thus,

$$
\left\|[\nabla_{w_2,b_2,\boldsymbol{w}_1}]_{1:3}\right\|^2 = \left(\mathbb{E}_{\boldsymbol{x}}\left\{(f_*(\boldsymbol{x}) - f(\boldsymbol{x}))\left\|\boldsymbol{x}\right\|\right\}\right)^2\left(\alpha^2 + \frac{4C_\Gamma\bar{w}_2^2}{\sqrt{d}}\alpha\right) + \left(\mathbb{E}_{\boldsymbol{x}}\left\{f_*(\boldsymbol{x}) - f(\boldsymbol{x})\right\}\right)^2
$$

$$
\pm O\left(\left(\delta_{1,L^2}^{(2)} + d^3\delta_2^{(2)}\right)d^4\right)
$$

$$
= \left\|\tilde{\nabla}\right\|^2 \pm O\left(\left(\delta_{1,L^2}^{(2)} + d^3\delta_2^{(2)}\right)d^4\right).
$$

$\square$

*Proof of Lemma D.21.* For notational simplicity, put $A := \sqrt{\alpha^2 + \frac{4C_\Gamma}{\sqrt{d}}\bar{w}_2^2\alpha}$. Then we can write

$$\tilde{\nabla} = \mathbb{E}_{\boldsymbol{x}}\left\{(f_*(\boldsymbol{x}) - f(\boldsymbol{x}))\begin{bmatrix} A\|\boldsymbol{x}\| \\ 1 \end{bmatrix}\right\}.$$

Define

$$\hat{\nabla} = \begin{bmatrix} -1 - \alpha\bar{w}_2 \\ A(1 - \bar{b}_2) \end{bmatrix}.$$

By Induction Hypothesis D.1, $\left\|\hat{\nabla}\right\| \leq O(1)$. Hence, in order to lower bound $\left\|\tilde{\nabla}\right\|$, it suffices to lower bound $\left\langle\tilde{\nabla}, \hat{\nabla}\right\rangle$. We have

$$\left\langle\tilde{\nabla}, \hat{\nabla}\right\rangle = A\mathbb{E}_{\boldsymbol{x}}\left\{(f_*(\boldsymbol{x}) - f(\boldsymbol{x}))\left(-\|\boldsymbol{x}\| + 1 - (\alpha\bar{w}_2\|\boldsymbol{x}\| + \bar{b}_2)\right)\right\}.$$

First, for those $\boldsymbol{x} \in \{\|\boldsymbol{x}\| \leq 1\}$, we have $f_*(\boldsymbol{x}) = -\|\boldsymbol{x}\| + 1$ and

$$f(\boldsymbol{x}) = \bar{w}_2 F(\boldsymbol{x}) + \bar{b}_2 = \bar{w}_2\alpha\|\boldsymbol{x}\| + \bar{b}_2 + \bar{w}_2\alpha(\bar{F}(\boldsymbol{x}) - \|\boldsymbol{x}\|).$$

Hence, we have

$$\mathbb{E}_{\|\boldsymbol{x}\|\leq 1}\left\{(f_*(\boldsymbol{x}) - f(\boldsymbol{x}))\left(-\|\boldsymbol{x}\| + 1 - (\alpha\bar{w}_2\|\boldsymbol{x}\| + \bar{b}_2)\right)\right\}$$

$$= \mathbb{E}_{\|\boldsymbol{x}\|\leq 1}\left\{(f_*(\boldsymbol{x}) - f(\boldsymbol{x}))^2\right\} + \mathbb{E}_{\|\boldsymbol{x}\|\leq 1}\left\{(f_*(\boldsymbol{x}) - f(\boldsymbol{x}))\bar{w}_2\alpha(\bar{F}(\boldsymbol{x}) - \|\boldsymbol{x}\|)\right\}$$

$$= \mathbb{E}_{\|\boldsymbol{x}\|\leq 1}\left\{(f_*(\boldsymbol{x}) - f(\boldsymbol{x}))^2\right\} \pm O\left(|\bar{w}_2\alpha|\delta_{1,L^2}^{(2)}\right).$$

Then, for $\boldsymbol{x} \in \{\|\boldsymbol{x}\| \geq 1\}$, note that $-\|\boldsymbol{x}\| + 1 \leq 0$ and $f_*(\boldsymbol{x}) = 0$. Therefore, we have

$$\mathbb{E}_{\|\boldsymbol{x}\|\geq 1}\left\{(f_*(\boldsymbol{x}) - f(\boldsymbol{x}))\left(-\|\boldsymbol{x}\| + 1 - (\alpha\bar{w}_2\|\boldsymbol{x}\| + \bar{b}_2)\right)\right\}$$

$$= -\mathbb{E}_{\|\boldsymbol{x}\|\geq 1}\left\{f(\boldsymbol{x})\left(-\|\boldsymbol{x}\| + 1 - (\alpha\bar{w}_2\|\boldsymbol{x}\| + \bar{b}_2)\right)\right\} \geq \mathbb{E}_{\|\boldsymbol{x}\|\geq 1}\left\{f(\boldsymbol{x})(\alpha\bar{w}_2\|\boldsymbol{x}\| + \bar{b}_2)\right\}.$$

Then, we compute

$$\mathbb{E}_{\|\boldsymbol{x}\|\geq 1}\left\{f(\boldsymbol{x})(\alpha\bar{w}_2\|\boldsymbol{x}\| + \bar{b}_2)\right\}$$

$$= \mathbb{E}_{\|\boldsymbol{x}\|\geq 1}\left\{f(\boldsymbol{x})(\bar{w}_2 F(\boldsymbol{x}) + \bar{b}_2)\right\} + \mathbb{E}_{\|\boldsymbol{x}\|\geq 1}\left\{f(\boldsymbol{x})(\alpha\bar{w}_2(\|\boldsymbol{x}\| - \bar{F}(\boldsymbol{x})))\right\}$$

$$= \mathbb{E}_{\|\boldsymbol{x}\|\geq 1}\left\{f^2(\boldsymbol{x})\right\} \pm O\left(d^3\delta_2^{(2)}\right) \pm O\left(\alpha\bar{w}_2\delta_{1,L^2}^{(2)}\right).$$

where the second equality comes from Lemma D.7 and Induction Hypothesis D.1. Combine these two cases together and we obtain

$$\left\langle\tilde{\nabla}, \hat{\nabla}\right\rangle \geq A\mathbb{E}_{\boldsymbol{x}}\left\{(f_*(\boldsymbol{x}) - f(\boldsymbol{x}))^2\right\} - O\left(|\bar{w}_2\alpha|\delta_{1,L^2}^{(2)}\right) - O\left(d^3\delta_2^{(2)}\right).$$

Finally, note that $A \geq \alpha$. Thus,

$$\left\|\tilde{\nabla}\right\| \geq \Omega(\alpha\mathcal{L}) - O\left(\delta_{1,L^2}^{(2)} + d^3\delta_2^{(2)}\right).$$

□

*Proof of Lemma D.22.* By Lemma D.20 and Lemma D.21,

$$\frac{\mathrm{d}}{\mathrm{d}t}\mathcal{L} \leq -\Omega(\alpha^2\mathcal{L}^2) + O\left(\left(\delta_{1,L^2}^{(2)} + d^3\delta_2^{(2)}\right)d^4\right) \leq -\Omega\left(\frac{\mathcal{L}^2}{d^3}\right)$$

Thus, for any $T \in [T_1, T_2]$,

$$\mathcal{L}(T) \leq \left(\Omega\left(d^{-3}\right)(T - T_1) + \frac{1}{\mathcal{L}(T_1)}\right)^{-1} \leq O\left(\frac{d^3}{T - T_1}\right).$$

Thus, it takes at most $O(d^3/\varepsilon)$ amount of time for $\mathcal{L}$ to reach $\varepsilon$.

□

### D.4 Proof of the Main Lemma

*Proof of Lemma D.2.* The Induction Hypothesis is maintained in Section D.2 and by Lemma D.22, we have $T_2 - T_1 \leq O(d^3/\varepsilon)$. Now we consider the first layer errors. Recall that

$$\frac{\mathrm{d}}{\mathrm{d}t}(\delta_{1,L^2}^{(2)})^2 = O\left(d^5\delta_{1,L^2}^{(2)}\left(\delta_{X_2}^{(2)}\right)^2\right),$$

$$\frac{\mathrm{d}}{\mathrm{d}t}\delta_{1,L^\infty}^{(2)} = O\left(d^3\delta_{1,L^2}^{(2)} + d^2\left(\delta_{X_2}^{(2)}\right)^2\right).$$

Recall that $\delta_{X_2} := O(1)d^{4.5}(\delta_{1,L^\infty}^{(2)} + d^3\delta_2^{(2)})$. For simplicity, we choose $\delta_{1,L^\infty}^{(2)} \geq d^3\delta_2^{(2)}$ so that $\delta_{X_2} = O(d^{4.5}\delta_{1,L^\infty}^{(2)})$. Then, we have

$$\frac{\mathrm{d}}{\mathrm{d}t}(\delta_{1,L^2}^{(2)})^2 = O\left(d^{14}\delta_{1,L^2}^{(2)}(\delta_{1,L^\infty}^{(2)})^2\right),$$

$$\frac{\mathrm{d}}{\mathrm{d}t}\delta_{1,L^\infty}^{(2)} = O\left(d^3\delta_{1,L^2}^{(2)} + d^{11}(\delta_{1,L^\infty}^{(2)})^2\right).$$

We choose $\delta_{1,L^2}^{(2)}(T_1)$ and $\delta_{1,L^\infty}^{(2)}(T_1)$ such that

$$\Theta\left(\frac{d^{17}}{\varepsilon}(\delta_{1,L^\infty}^{(2)})^2\right) \leq \delta_{1,L^2}^{(2)} \leq \Theta\left(\frac{\varepsilon}{d^6}\delta_{1,L^\infty}^{(2)}\right) \quad \text{and} \quad \delta_{1,L^\infty}^{(2)}(T_1) \leq \Theta\left(\frac{\varepsilon}{d^{14}}\right). \tag{14}$$

Note that this is possible because $\delta_{1,L^2}^{(2)}(T_1)$ and $\delta_{1,L^\infty}^{(2)}(T_1)$ can be chosen to be arbitrarily polynomially small. When this is true, we have

$$\frac{\mathrm{d}}{\mathrm{d}t}(\delta_{1,L^2}^{(2)})^2 \leq O\left(\frac{\varepsilon}{d^3}(\delta_{1,L^2}^{(2)})^2\right) \quad \text{and} \quad \frac{\mathrm{d}}{\mathrm{d}t}\delta_{1,L^\infty}^{(2)} = O\left(\frac{\varepsilon}{d^3}\delta_{1,L^\infty}^{(2)}\right).$$

Thus, by induction, within $O(d^3/\varepsilon)$ amount of time, these two errors can at most $O(\delta_{1,L^2}^{(2)}(T_1))$ and $O(\delta_{1,L^\infty}^{(2)}(T_1))$, respectively. $\qquad\square$

## E From gradient flow to gradient descent

Converting the above gradient flow argument to a gradient descent one can be done in a standard one, provided that we can generate fresh samples at each iteration. First, by choosing a sufficiently small step size, one can make sure within each step, the difference between gradient descent and gradient flow is inverse polynomially small. Note that our argument is built upon the induction hypotheses. Hence, we do not need to worry about the accumulation of errors. Moreover, our estimations can tolerate an inverse polynomially large error. Then, at each step of gradient descent, we generate sufficiently (but still polynomially) many samples to ensure that with high probability, the difference between the population gradient and the finite-sample gradient is sufficiently small. Since it only takes polynomial iterations to finish the process, the total amount of samples needed is polynomial.

