# OpenReview forum: "Depth Separation with Multilayer Mean-Field Networks"
_ICLR.cc/2023/Conference — ICLR 2023 notable top 25%_

### Official Review · Reviewer_JRVm · 2022-10-22

**Confidence:** 4
**Correctness:** 3
**Technical Novelty And Significance:** 2
**Empirical Novelty And Significance:** Not applicable
**Recommendation:** 6

**Clarity, Quality, Novelty And Reproducibility:**

The result is new, though as said the proposed idea for MF networks is known. The writing of the paper is generally clear.

One thing to note is that the paper has not proved well-posedness of the gradient flow. For completeness, the paper should discuss this issue or at least state it as an assumption.

**Strength And Weaknesses:**


The result of the paper constitutes an interesting example where it shows the algorithmic advantage of a deep network against a shallow network. Global convergence results in the MF literature typically show (multilayer) neural nets can learn a certain generic class of target function or distribution, often without a rate and without an explicit comparison against shallower models. Here the paper focuses on a very specific setting and pulls through the main technical hurdle, which is to show the convergence time $T = poly(d)$ while avoiding the typical Gronwall’s inequality bound on the width whose dependency is exponential in time, hence achieving a $poly(d)$ width guarantee. Though I haven’t read the proof carefully, I quite like the analysis that takes a keener look at the role of the gradient flow in Lemma D.12 and D.14, which in particular helps keep the bound on $\frac{d}{dt}(\delta_{1,L_2}^{(2)})^2$ on the smaller, useful order (i.e. $\frac{d}{dt}(\delta_{1,L_2}^{(2)})^2 \ll \delta_{1,L_2}^{(2)}(\delta_{1,L_\infty}^{(2)})^2$ instead of $\frac{d}{dt}(\delta_{1,L_2}^{(2)})^2 \ll \delta_{1,L_2}^{(2)}\delta_{1,L_\infty}^{(2)}$, which would have been bad).

There are a few items that should warrant attention:

- The type of MF networks proposed by the paper is actually a known idea. It is basically a concatenation of multiple 2-layer MF networks. Similar ideas exist, for example, MF Resnets in a paper by Jianfeng Lu, Lexing Ying and coauthors, which is one of the cited references. As far as I know, the same proposal was discussed at least in one talk by Eric Vanden-Eijnden.

- There are several important parameters that have been chosen so as to “hint” the learning at the desired solution. These choices could be unnatural. In particular, the initialization of $w_2$ is chosen to be very small and basically close to 0. As a result, this hints the network to learn an atomic solution. The initialization of $w_1$ (given by $\sigma_1 = 1/\sqrt{d}$ in Lemma C.1) and the unnatural architecture $F(x) = \vert w \vert \sigma(w\cdot x)$ again hint directly at the solution, as $\bar{F}(x; t=0) = \vert x \vert$. Due to the small initial $w_2$, in the first stage, $w_1$ has almost no movement. Hence in the first stage, the network $F(x)$ stays “at the right place”, which is necessary for it to stay again at the right place in the second stage, as shown on page 43.

  Zooming out, one can see that with the choice of parameters, most things are already at their right places from the beginning (the first layer $\bar{F}$, the spread of $w_2$). Perhaps the only thing that has large movement through the learning is $\bar{w}_2$, which is a scalar and does not quite convey the necessity of MF infinite-width networks. The same can be said about the bias term.

- A more critical point is whether the proposed MF network is really important to establishing the result. The proof actually suggests that if the second width $m_2=1$, the result can be readily proven — perhaps with less technical work. That is, it is not necessary that one has $m_2\to\infty$. However when $m_2=1$, the “MF” part of the network would then be just a 2-layer MF network, not a multilayer one.

  One may say it is more natural to consider large $m_2$ than $m_2=1$. But we should also recall the aforementioned design choices that hint at the desired solution. With these design choices, the consideration of large $m_2$ seems to demand heavier technical work and yet does not reveal insights on whether or why a large width is necessary.

- The paper also advertises that the proposed MF network factors out neuron-permutation invariance due to the use of the distributions $\mu_1$ and $\mu_2$ in the representation, but the analysis does not make use of the representations $\mu_1$ and $\mu_2$. This is quite unlike the 2-layer MF literature, where the movement of the neuron distribution $\mu$ is described by a Wasserstein gradient flow and where the analysis makes use of this fact.

  More generally a typical MF framework goes by passing from the large-width MF network to the MF (infinite-width) limit and then using this MF limit as an analytical object. The advantage of this framework is that it removes the width out of the analysis and thus allows to exploit certain properties that only exist in the infinite-width limit. The paper does not follow this approach: it does not identify the MF limit and it analyzes directly the large-width MF network. Furthermore the analysis looks rather restricted to the particular problem setup under consideration. As such, at the time it is hard to judge whether this type of analysis can be expanded into a proper framework.


**Summary Of The Paper:**

The paper analyzes the gradient flow learning of a particular mean-field (MF) network with more than 2 layers and shows that it can learn a radial target function (and a specific data distribution) with width polynomial in the data dimension $d$. This target function was previously shown to be inapproximable by 2-layer networks whose widths are $poly(d)$.

**Summary Of The Review:**

The paper presents an interesting and encouraging result, though there are several downsides, including a somewhat restricted problem setting, the MF network idea being already known, several unnatural design choices of the neural net that already reveal quite a lot about the desired target function before learning takes place, and the questionable necessity of the proposed multilayer MF network.

---

> ### Author Response · Authors · 2022-11-18
> **Response to Reviewer JRVm**
>
> Thank you for your reviews. We'd like to address your concerns as follows.
>
> * It is true that the idea of a bottleneck structure comes from ResNet (we also mention this on page 2). However, how we use this structure is very different from Lu et al. (2020). In ResNet or mean-field ResNet, this bottleneck structure is used to make sure the dimension of the output of each layer is the same so that it makes sense to treat the output of latter layers as residuals and add it to previous layers. This allows one to "chain" the outputs to obtain a neural ODE. In our case, the role of the bottleneck structure is exactly the opposite. We use them to decouple the outputs of different layers so that we can treat each layer in a relatively independent fashion. Unfortunately, we are not aware of the talk by Eric Vanden-Eijnden.
>
> * We indeed choose the parameters to simplify the proof. However, our technique and most of the argument can be carried to more general settings, architectures, and choices of parameters. (See the newly added official comment for details.) We also want to point out that the norm of the first layer neurons also changes a lot in Stage 2.
>
> * Though choosing $m_2 = 1$ can make parts of the proof slightly simpler, it will not change the main technical challenge, which is to control the discretization error of the first layer. One can see from the new comment that the analysis for the second layer is fairly flexible, and not restricting to $m_2 = 1$ makes it easier to adapt the techniques to more general settings.
>
> * Though it is not very explicit, our argument still follows the infinite-width to finite-width paradigm. We observe that, for the infinite-width network, the dynamics of the first layer can be captured by a single real number $\alpha$ and are easy to analyze. Then, to deal with finite-width networks, we factor out the terms that would appear in the infinite-width dynamics and bound the remaining terms. The most apparent place where this strategy is used is in the derivation of the convergence of Stage 2. There, we reduce everything to $\alpha, \bar{w}_2, \bar{b}_2$.
>
>   The general mean-field-style way to establish global convergence relies on the distribution to be non-degenerate or cover all directions, at least until the very end of training. This generic strategy inherently prevents polynomial-width discretization, as it generally needs exponentially many neurons to maintain these conditions approximately. Hence, it is necessary to take a closer look at the infinite-width dynamics. In our paper, the property we leverage is the spherical symmetry of the first layer, which only (strictly) exists in the infinite-width limit. In some sense, the message here is that infinite-width networks can possess certain nice problem-dependent properties which enable simple analysis, and if one tries hard enough, one can discretize an infinite-width network with polynomially many neurons by leveraging these properties.
>
> * Regarding the well-posedness, we state it as an assumption in the revision. Our proof does not require nor imply the uniqueness. In Stage 1, we replace certain parts of certain terms with the projection threshold, and we don't (need to) know what happens there. After Stage 1, the uniqueness follows the standard argument. For the existence, we believe it can be proven via the Peano existence theorem.

---

> ### Comment · Reviewer_JRVm · 2022-12-08
> **Thanks**
>
> Many thanks for the response and the outline for the potential extension!
>
> Overall I do not think the work is groundbreaking, for the same reason that I outlined earlier: the role of the overparameterization of the second layer is almost trivial and hence it is effectively at best a (lightly-parameterized) function of a shallow overparameterized neural net, not a multilayer overparameterized one. From what I understand about the work and authors' responses, one needs the effective parameterization to be 1D  for the analysis to work (where in the analysis does the entire distribution, rather than a single statistic $\bar{w}_2$, play a meaningful role?). This is not just to make the analysis simpler; rather by having $m_2>1$, one has to deal with the spread of the distribution in the second layer (which is eventually proven to be small), which is a technical complication than anything meaningful. Nevertheless I do think the work has interesting elements, so I keep my score at 6.
>
> Out of respect to past works (such as the Resnet work by Jianfeng Lu, Lexing Ying and coworkers), I still stress that despite the difference in the way in which multiple shallow MF neural nets are combined into a single network, the core idea of a combination of multiple shallow MF neural nets already exists well before this work.

---

### Official Review · Reviewer_5SVU · 2022-10-24

**Confidence:** 3
**Correctness:** 4
**Technical Novelty And Significance:** 4
**Empirical Novelty And Significance:** 3
**Recommendation:** 8

**Clarity, Quality, Novelty And Reproducibility:**

Overall, clear, and novel contribution, with a few omissions (see weaknesses) in the literature related to expressivity and optimization of ReLU nets.
Typos:
-page 3: see Equation Equation 5 --> drop one word


**Strength And Weaknesses:**

Strengths:
+ nice technique that extends previous techniques based on mean field analysis of 1 layer.
+ proving that one can train a network to find the hard-to-represent-by-shallow-nets function is itself a clean and nice result.
+ it's interesting that his paper studies training of both the layers. Training dynamics for one layer while keeping the previous fixed, were studied but this paper deviates.
+ perhaps the idea of truncating the mean field analysis, studying the introduced discretization error could be useful for future problems.

Weaknesses:
- the architecture itself is somewhat simple as the intermediate layer has dimension only 1. This suffices for learning the norm function, but it's not clear how crucial it is for the analysis. Would the analysis be able to be carried out for internal dimension d? What would the dependencies on d be?
-Some omissions in the literature and how the paper compares to them:

-What happens in the case where depth is not 3, but rather a parameter D and we want to learn a "simple" function? Telgarsky's paper on "Benefits of depth in neural networks" provides similar separation in the "deep regime" based on an admittedly very simple function, namely the triangle wave f(x)=2x for x in [0,0.5] and f(x) = 2(1-x) for x in [0.5,1].
Could your result somehow be made algorithmic even in the case of learning compositions of the triangle wave?
More generally, Chatziafratis et al. in "Depth-Width Trade-offs for ReLU Networks via Sharkovsky's Theorem" generalized Telgarsky's construction based on periodic 3 functions. Could this also be made algorithmic?

-The authors should compare or at least mention connections to computational complexity and neural networks based on the work of Vardi, Shamir in "Neural Networks with Small Weights and Depth-Separation Barriers", where showing separations beyond depth 4 is connected to some basic questions in complexity.

- The authors should also mention "Efficient Algorithms for Learning Depth-2 Neural Networks with General ReLU Activations" by Awasthi et al. since they study the very related question on learning a ReLU net. Are the techniques used there similar to the ones in your paper?

**Summary Of The Paper:**

The paper studies the benefits of Depth in neural networks with an emphasis on algorithmic separations.

Typically, previous works have focused on proving depth separations by proving the existence of certain functions that on the one hand, can be easily approximated with a depth 3 neural net, yet cannot be approximated well by any depth-2 network. The paper here asks can we prove the analogous statement but with a learnable deep network.

The main result relies on previous work using a radially symmetric function (i.e., one that depends on the norm $||x||$ of the input) and importantly show how to train and converge to a network that provably approximates the function at hand. This requires several technical extensions of prior works to multilayer mean-field networks. The network studied is overparameterized and has polynomially many neurons. One interesting aspect of the analysis is to decompose the loss into the discretization loss (going from infinite width to fixed width) and the approximation error (w.r.t. to the hypothses classes and the function to be approximated).

**Summary Of The Review:**

The paper is a solid contribution to the theory of approximation/expressivity of neural nets. Many prior works have studied the depth separation question from an existential point of view, and this work shows that for some simple case (namely the radial function $||x||$), the result was indeed algorithmic. But to do so several technical obstacles are overcome in terms of the optimization process.

---

> ### Author Response · Authors · 2022-11-18
> **Response to Reviewer 5SVU**
>
> Thank you for your reviews! We add citations to the non-algorithmic depth-separation papers in the revision. Possible ways to generalize the argument to other architectures are discussed in a newly added comment, and we'd like to answer your other questions as follows.
>
> We believe that our technique can potentially be generalized to handle constant-depth networks as one can Taylor expand all layers around its infinite-width counterpart. However, estimations regarding the convergence rate and the current second layer need to be changed accordingly. When the depth is super-constant, things become trickier, and it's not clear how the errors in different layers affect each other. For the function constructed by Telgarsky, there are actually some negative results regarding its learnability: In Malach et al. (2021), the authors show that that function cannot be learned by standard gradient descent. We feel that establishing algorithmic results for networks with super-constant depth is generally tricky and hard. Even in practice, training a very deep fully-connected network is usually difficult.
>
> The algorithm proposed by Awasthi et al. is very different from any gradient-based algorithm used in practice. The point of that paper is closer to "if we can use specialized algorithms, it is possible to learn depth-2 networks".
>
>
> [1] Eran Malach, Gilad Yehudai, Shai Shalev-Shwartz, Ohad Shamir. The Connection Between Approximation, Depth Separation and Learnability in Neural Networks. (2021)

---

> > ### Comment · Reviewer_5SVU · 2022-12-05
> > **thank you**
> >
> > Thank you for your responses!

---

### Official Review · Reviewer_wTzk · 2022-10-25

**Confidence:** 3
**Correctness:** 3
**Technical Novelty And Significance:** 3
**Empirical Novelty And Significance:** Not applicable
**Recommendation:** 6

**Clarity, Quality, Novelty And Reproducibility:**

It seems technically strong, but I feel it lacks some explanation when describing the dynamics for the infinite width. Some variables with bar on top does not seem to be defined or explained anywhere. I also feel the whole multi-layer mean field network can be described in the main paper just with D=1 and shift the generalized version to the appendix and focus a bit more on the loss partitioning and other novelties on the specific simpler architecture.

**Strength And Weaknesses:**

Strengths:
Prior work by (Safran and Lee 2021), train GD on a depth 3-network, but for different activation functions other than the standard ReLU and their first layer-weights are fixed throughout the training. In this work, the authors use a splitting technique to deal with moving weights and also to ensure permutation invariance of the distribution of neurons.
After initialization of the weights and under the infinite width regime they obtain a distribution over the weights, which is spherically symmetric and then they are able to show under an appropriate discretization scheme, they can transfer the results to a finite polynomial width network.  I think it is one of the few results that show polytime learnability with depth 3 network, albeit on a specific instance. Potentially some of the techniques could be of interest to the community.

Weaknesses:
1) As this a depth separation result, even though they train a deep network, the methods are more or less, specific to the particular function. Can the authors comment on whether there is a larger class of functions and input distributions such that a depth 3-network can learn in polytime?
2) The equations in (4) describe the dynamics, please indicate what is $\bar{v}_1$. One of the concerns is the assumption on the spherical symmetry of the distributions, which is satisfied in the beginning, but could the authors please clarify why it continues to be satisfied as the dynamics changes the weights? Does this always ensure that the assumption is valid and also when you discretize and deal with the finite width regime?
3) Can the authors comment about dealing with polynomially many samples from the input distribution? Does one run standard GD, with gradient clipping in this case?


**Summary Of The Paper:**

In (Safran et al. 2019), the authors describe a function which is $ReLU(1-\lVert x\rVert)$, which can be well approximated by a depth 3-network, while a depth 2-network requires $\Omega(\exp(d\log d))$ width for the same. In this paper the authors show that, this separation is also, algorithmic, in the sense that one can learn the hard function $ReLU(1-\lVert x\rVert)$ using a depth 3-network in time polynomial in $(d, 1/\varepsilon)$ using mean-field dynamics and an appropriate discretization scheme.

**Summary Of The Review:**

Overall, even though it is tied to learning a specific hard function, I think the techniques could be useful to understand optimization with deeper architectures. So I feel it might be of interest to the DL theory community.

---

> ### Author Response · Authors · 2022-11-18
> **Response to Reviewer wTzk**
>
> Thank you for your reviews! We discuss some possible extensions in a newly-added comment. We'd like to address the other concerns as follows.
>
> * $\bar{v}_1 := v_1 / \|v_1\|$ in (4). We clarify this in the notation paragraph in the revised version. We also add a section in the appendix where we show the infinite-width network will remain spherically symmetric (Appendix B.4), though our finite-width proof does not depend on that. In the finite-width regime, $\mu_1$ is never spherically symmetric, but we show that the output of the first layer $F(x)$ is always close to the output of its infinite-width counterpart $\tilde{F}(x)$.
> * To convert the GF argument to the GD one, basically one needs to rewrite the proof and replace all occurrences of Gronwall's lemma and similar lemmas with their discrete counterparts to control the error growth. (Note that the gradient estimations are still valid as they rely on the induction hypotheses locally in time.) This will lead to a population GD result that requires $\mathrm{poly}(d, 1/\epsilon)$ iterations. Then, to go from population GD to finite-sample GD, we sample $\mathrm{poly}(d, 1/\epsilon)$ fresh samples at each step of GD to make sure that with high probability, the difference between the population gradient and the finite-sample gradient is sufficiently small and can be merged into the error terms of the current gradient estimations. Since we only run GD for polynomially many iterations and the number of needed samples at each iteration is also polynomial, the total sample complexity is polynomial. We also add discussions on this GF to GD problem in the appendix.

---

### Official Review · Reviewer_i3qA · 2022-10-25

**Confidence:** 3
**Correctness:** 4
**Technical Novelty And Significance:** 4
**Empirical Novelty And Significance:** 2
**Recommendation:** 8

**Clarity, Quality, Novelty And Reproducibility:**

**Clarity**: This work is clearly written and easy to follow even for a non-expert.

**Quality**: The problem studied is theoretically relevant and the results are interesting.

**Novelty**: This work builds on previous results on depth separation. However, it provides an algorithmic separation result which to my best knowledge is new and relevant.

**Reproducibility**: Proof details are given in the appendix. However, code to reproduce the figures is not provided.

**Strength And Weaknesses:**

**Strengths**:
- [+] Algorithmic separation and convergence rates results for neural-nets are scarce. Therefore, even if the setting is specific this is a very interesting result.
- [+] The paper is well-written and easy to follow, even for a non-expert reader. In particular, the authors make an effort to intuitively explain every step in the proof, which is helpful.
- [+] The extension of the mean-field limit to deeper (although low-rank) architectures can be of independent interest to the community.

**Weaknesses**
- [-] The proof is cumbersome, and despite the effort highlighted above, one sometimes get lost in the big scheme. It would be useful to have a bullet list in Section 2 summarising the key steps and pointers to the Lemmas / Theorems establishing them, e.g. "First, we show that for spherical symmetry of initialisation is preserved by the first layer measure throughout the infinite-width dynamics", etc.
- [-] It is not so clear which of the assumptions needed for the result are crucial for the result to hold and which are not. e.g. what is the role of the cumbersome distribution of inputs?
- [-] The setting is quite specific. However, as highlighted above I think this is a minor point.

**Questions**:

- **[Q1]**: The data distribution seems to play an important role in the theoretical analysis. Aside from the technical proof, would changing the distribution (e.g. $x\sim\mathcal{N}(0,I_{d})$ or spherical) completely change the phenomenology?
- **[Q2]**: The separation of scales $T_{1}, T_{2}, T_{3}$ in the dynamics is an important element in the proof. If I understood correctly, these scales are independent of the hidden-layer widths $m_{1}, m_{2}$,and only scale with input dimension $d$. Can the authors clarify this scaling more explicitly?
- **[Q3]**: Overparametrisation plays an important role in the proof. However, as noted by the authors in the discussion below eq. (1) a single second-layer neuron suffices to learn the target. Aside from the technical details of the theorem, can the authors comment on why overpamatrisation is benign to the optimisation in this setting?
- **[Q4]**: If I understand it correctly, the initialisation of the first layer weights on the sphere is important to establish spherical symmetry of the first layer measure throughout the dynamics. Is this condition stable? For instance, would a small angular fluctuation (e.g. Gaussian initialisation) drives the dynamics away from the radial direction?

**Comments**:

- **[C1]**: I find Fig. 1 cryptic. First, the font size is very small and one needs to zoom a lot the pdf to read them. Second, the specific details of what is being plotted is not given. For instance: what are the red dashed vertical lines in Fig. 2 (left)? What is the input dimension $d$ in this simulation (this is quite relevant, since the authors claim convergence is polynomial in $d$)? Are these finite network simulations (if yes, what is $m_1$ and $m_2$?) or theoretical, infinite-width curves?

- **[C2]**: In page 6, first equation $\bar{v}$ is not defined explicitly.

**Summary Of The Paper:**

This work studies the problem of learning the target function $f(x) = \rm{relu}(1+||x||)$ under the following setting:

- *Data*: Data is sampled from a specific distribution $x\sim\mathcal{D}$ (see Section 2, "*Target Function and Input Distribution*")
- *Model*: A three layer neural-network with low-rank second layer weights.
- *Algorithm*: Clipped gradient flow on the population mean-squared error, with small random initialisation.

Despite specific, the interest in the target (and data distribution) above lies on a recent depth separation result by [Safran et al. '19] showing that it requires *at least* three-layers to be approximated with polynomially many neurons. The key theoretical result in this work is to show that indeed it can be learned algorithmically, i.e. the low-rank architecture converges to the target under clipped gradient flow from random initialisation in polynomial time.

On a technical note, the proof strategy introduces a mean-field approximation for multi-layer neural nets with low-rank middle layers which can be of independent interest.


**Summary Of The Review:**

This work studies the relevant problem of depth separation in neural networks. Despite considering a specific setting, it provides interesting results on the convergence of gradient flow for overparametrised three-layer neural networks in polynomial time. This sort of result is scarce in the theoretical literature. Therefore, I believe it is a submission of interest to the ICLR community.

---

> ### Author Response · Authors · 2022-11-18
> **Response to Reviewer i3qA**
>
> Thank you for your reviews! We add some clarification and update Fig. 1 and its caption in the revised version. How the input distribution interacts with the argument is discussed in a newly-added comment. We'd like to answer the other questions as follows.
>
> * On the scale of $T_1, T_2, T_3$. Yes, they don't scale with the network width. Explicit bounds can be found in the appendix (Lemma C.6, C.9, C.10 and Lemma D.2). The overall time needed is upper bounded by $O(d^3 / \varepsilon)$. (We did not try to optimize this bound.)
> * We mainly use over-parameterization to make sure the first layer can approximate $\alpha \|x\|$. For the second layer, it is true that only one neuron is enough. We keep it over-parameterized to match the framework and allow some possible generalizations (see the new comment for more discussions). In our setting, the over-parameterization of the second layer does not break things because, after Stage 1.1, all second layer weights become negative. As a result, there will be no cancellation, and we can control the spread of the second layer and approximate it using a single neuron.
> * We use the sphere distribution to initialize the first layer weights to make the proof cleaner. Technically, any reasonable spherically symmetric distribution, including Gaussian, is OK. This is because the first layer is $2$-homogeneous in weights, and one can just project the neurons onto the unit sphere and reweight them according to the original $\|w_1\|^2$. This strategy is used in [1] and also in one of the early mean-field papers [2].
>
> [1] Colin Wei, Jason D. Lee, Qiang Liu, Tengyu Ma. Regularization Matters: Generalization and Optimization of Neural Nets v.s. their Induced Kernel. 2019
>
> [2] Lénaïc Chizat, Francis Bach. On the Global Convergence of Gradient Descent for Over-parameterized Models using Optimal Transport. 2018

---

### Official Review · Reviewer_Lcb7 · 2022-11-02

**Confidence:** 4
**Correctness:** 4
**Technical Novelty And Significance:** 4
**Empirical Novelty And Significance:** 4
**Recommendation:** 8

**Clarity, Quality, Novelty And Reproducibility:**

The paper is clear on the whole, neat and the result is definitely new (at least to the best of my knowledge). Some of the sections could be improved though as the reader sometimes has to go back several pages in order to retrieve the details of a notation.


**Strength And Weaknesses:**

The paper is well written and is definitely interesting. My main concern is with (1) the relatively restrictive nature of the architecture and training machinery considered. The authors essentially focus on an example from Safran, Eldan, and Shamir (2019)  (which shows that shallow networks are limited in their expressive power and will fail to learn sufficiently complex functions such as Relu(1-||x||^2)) and consider a network that is designed specifically for that example which reduces the applicability of the result. (2) the fact that the readability/impact of the paper would benefit from a longer and more detailed exposition (see my comments below). This is a strong paper, why not submitting it to a journal?



**Summary Of The Paper:**


The paper uses a mean field type analysis on non shallow networks. The work is motivated by a result of Safran, Eldan, and Shamir (2019) which shows that shallow networks cannot capture sufficiently complex functions. The main result guarantees the learning of the function Relu(1-||x||^2), by a neural network with several multiple hidden layers with a polynomial number of units in each layer (which is a novelty)


**Summary Of The Review:**



- On page 2, last paragraph, the sentence “Restricting the intermediate layer to have only one dimension ” is not very clear. What you do is your restrict the connections between the first and second layer
- In Theorem 1, I would recall the definition of the main parameters. This would clearly improve readability (Let d denote the input dimension, …). Also, what do you mean exactly by poly(d, 1/eps)?
- It would also be good to recall that gradient flow is an intractable algorithm in practice as it optimizes in the space of measure (which is infinite dimensional)
- On page 4, when you introduce the distribution related to phi(x), it is not clear whether the impossibility to approximate f_* is related to the distribution or not.
- Which loss are you using between (2) and (3) ? I would use distinct notations
- In your main theorem, when you say gradient flow, you mean a flow algorithm on a polynomial number of particles, right? I think it would improve readability to say it that way or to add that somewhere.
- page 6 first paragraph. what do you mean by “the change in norm is also uniform ”
- your defintions of mu_1 and mu_2 are sometimes ambiguous. From what I understand, by mu_1 and mu_2 (at least when you derive the polynomial width), you mean multi-atomic distributions, don’t you? if so, why not writing the corresponding decomposition mu_1 = sum_{i} alpha_i delta(w - w_i) somewhere?
- On page 6, when you say that the dynamics of the first layer reduces to alpha, from what I understand, it is because you can initialize v1 to a given (let us say random) vector and then derive the update from the update on alpha ? I think you should expand a little more on this. E.g. provide one iteration, v_1 = v_1 + eta* d\alpha/dt * (1/alpha)* v_1

- On page 6, it took me some time, to get the flow on alpha and I think (although everything is sound) it would be good for the (general) reader  to recall the meaning of \cdot{alpha} here. I.e. the fact that \cdot{\alpha} denotes the variation of alpha through the gradient iterations. Perphaps you could add something like alpha(t+eta) = \alpha(t)+ eta d alpha/dw_1  * dw_1/dt and recall that dw_1/dt is the evolution of w_1 throughout the gradient iterations. I.e. indicate that dw_1/dt is computed as w_1(t+eta) = w_1(t) + eta* dL/dw_1
- It would be good to clarify the notation E_x (Given that you have other averages with respect to the distribubtions mu_i and you sometimes use \mathbb{E} without subscript). Perhaps add a line somewhere such as E_x \left\{ f(x)\right\} = \int_{-\infty}^{\infty} f(x)\; dx
- On page 7, when you discuss the vanishing of the gradients, it would be easier for the reader to recall the definitions of f and F
- On page 7, the sentence “As a result, when f decreases sufficiently fast, f will become 0 before ∥x∥ becomes large” is unclear.
- On page 7, when you introduce the times T_{1,1}, T_{1,2} and T_{1,3}, those times are critical to the analysis, I’m wondering if they should not be introduced as a list. Moreover, just before the statement of lemma 4.2, when you introduce the T_{1,i}, I would emphasize that the projectors are not needed anymore after T_{1,1}. At the end of the day this is the point of the section.

My general comments for lemma 4.2. are the following: The statement of the lemma is perfect for a journal but there are too many details for conference proceedings. I would make it even more informal and simply say something like the first phase of the optimisation is characterized by small changes in alpha and (v_2, r_2) (including small changes in the mean \overline{w}_2 and spread \|v_2 - v_2’\| across the layer). Then if you really want you can give the estimates on \overline{w}_2, r_2 and alpha but you should explain why those are important.

- In lemma 4.2., is \overline{v}_1 the same as \mathbb{E} w_1 ?
- In lemma 4.2., how do you control the variance \mathbb{E}\|w_1\|^2 ?
- In lemma 4.2. “spread of the second layer” and “regularity condition”, when you say for all (v_2, r_2) \in \mu_2, I guess you mean for all (v_2, r_2) \in \supp(\mu_2)

- On page 7, if I understand well, T_{1,1} is the stage where you still need the projectors ? if so I would indicate it.
- In the statement of Lemma 4.1 I would recall the meaning of R_{v_1}, R_{v_2} and R_{r_2}.
- On page 7, you use ‘f(x)sigma’(v_2 F(x) + r_2) \approx f(x) ‘ is that because d/dx relu(x) =1 on the positive x ? if so then what does the approximation mean?
- On page 7, “Since f is much flatter than f∗ , the RHS is always negative” ? What does that even mean? I don’t understand why the flatness of f_* vs f implies that f_* - f will be negative?
- In fact at the end of p7 you say “In fact, we show that it is −Θ(α log d) = −Θ(logd/d1.5) “ Where do you show this ?
- “One also needs to show that δ2 cannot change much during Stages 1.1 and 1.2.” —> Why ?
- I would recommend changing the sentence “the dynamics of v2 is approximately uniform in Stage 1.1 and Stage 1.2, and δ2 does not change much” to the distance between successive iterates (v_2’, r_2’) and (v_2, r_2) does not change much
- At the beginning of page 8, the sentence “Recall that stages 1.1. and 1.2. only requires” —> “recall that T_13” corresponds to the time at which |w_2| = Theta(d)\delta_2
- “We can make σ2 small by selecting a small enough σ2” the sentence does not make sense. I would replace by something like “Recall that sigma_2 refers to the variance of the Gaussian distribution used to initialize w_1, by taking this parameter small enough, we can guarantee that the initial value of w_1 will be small enough which can in turn be used to control the deviation between the iterates” or even better “the deviation between the iterates in v_1 and r_2 depend on the value of sigma_2 (which is used to initialize the weights w_1 and which can be taken sufficiently small to control this distance)”
- I don’t see why the fact that “v_2” has a uniform dynamics (i.e. the successive gradient steps are approximately constant) implies that delta_2 will be small. This implies delta_2 will be approximately constant but not necessarily small. I might be missing something here

- I think the paragraph at the end of page 7 and beginning of page 8 should be rewritten or removed. You are trying to explain a proof which you can’t expand and that makes the paper unclear. Honestly I think the best would be to remove this and replace your statement of lemma 4.2 with something very informal


Page 8

- on page 8, it is not clear to me why \overline{w}_2 does not depend on \delta_2
- on page 8, I guess when you say “the length of Stage 1.1 and Stage 1.2 is proportional to \delta_2” what you mean is the total variation in the weights? if so it might be helpful for the reader to specify which weights
- Generally speaking the part where you explain lemma 4.2. is not clear
- When you explain how you control the
- In the last paragraph of page 8, perhaps you could recall the definition of \tilde{f} ?

===================================

Additional typos:

===================================


- page 2, end of paragraph 2: “has also been in used” —> “has not been used” or “”has not been in use” (although the second is less correct)
- page 7, “Stages 1.1 and 1.2 only requires” —> “only require”
- page 8, beginning of the page : “if δ2 remain relatively constant” —> “if delta_2 remainS constant”

---

> ### Author Response · Authors · 2022-11-18
> **Response to Reviewer Lcb7**
>
> Thank you for your reviews and suggestions! We have incorporated most of them into the revised version and added some other clarifications. Some of them are replicated below. ("Page x" refers to the page in the original version.)
> * By saying some quantity is $\mathrm{poly}(d, 1/\varepsilon)$, we mean it is bounded by $C (d / \varepsilon)^C$ for some universal constant $C > 0$ that may change across different $\mathrm{poly}(d, 1/\varepsilon)$.
> * We use the standard MSE (3) as the loss. (2) is an approximation we established in the proof.
> * On page 6, by "the change in norm is also uniform", we mean $\frac{d}{d t} \|v_1\|^2$ does not depend on its direction $\bar{v}_1$. Since $\mu_1$ is the uniform distribution over some sphere, this implies that $\frac{d}{d t} \|v_1\|^2$ are the same for all $v_1$.
> * On page 6, we say the dynamics of the first layer can be reduced to the dynamics of $\alpha$ because, in this infinite-width case, the output of the first layer can be characterized by $\alpha$ (cf. (5)), and the dynamics of $\alpha$ only depend on $\alpha$ itself.
> * It's true that $\mu_1, \mu_2$ are empirical distributions of finitely many neurons in most parts of the proof. We use this notation to unify the finite- and infinite-width discussions and also because it's usually more succinct.
> * In Lemma 4.2, $\bar{v}_1$ means $v_1 / \|v_1\|$.
> * We control the relative spread of $|w_1|^2$ in Lemma C.16. The basic idea is that since the first layer is 2-homogeneous in $w_1$, the growth rate is proportional to $|w_1|^2$. As a result, the relative spread barely grows.
> * On page 7, when we say $f(x) \sigma'(v_2 F(x) + r_2) \approx f(x)$, actually we mean for most $x$, $\sigma'(v_2 F(x) + r_2) = 1$. We rewrote that sentence to make it clearer.
> * On page 7, "$f$ is much flatter than $f_*$" means the slope is smaller. Therefore, $f$ will remain $\Theta(1)$ after $f_*$ becomes $0$ because $\|x\| \ge 1$.
> * On page 7, what we show is $\mathrm{RHS} = - \Theta(\alpha \log d)$. The second part $- \Theta(\alpha \log d) = - \Theta(\log d / d^{1.5})$ comes from the fact $\alpha = \Theta(1 / d^{1.5})$ (cf. Lemma 4.2(c)). We removed this second part of the sentence in the revision as it indeed causes some confusion.
> * For $\delta_2$, what we intended to say is that it does not grow too much.
> * We use a small initialization for $w_2$. Hence, $\delta_2$ is small at initialization.
> * On page 8, we mean the target value of $\bar{w}\_2$ no longer depends on $\delta\_2$. It is $\Theta(1 / R_{v_2})$ in Stage 1.3.

---

### Author Response · Authors · 2022-11-18
**On the setting and possible extensions**

We thank all reviewers for their reviews and suggestions. Here, we are glad to share some thoughts on the setting and possible extensions, as multiple reviewers think the setting to be quite restricted.

**Input distribution**: The use of this specific distribution is mainly for the negative result, which requires the product of the target function and input distribution to have a large enough high-frequency part. Our positive result works for all reasonable spherically symmetric distributions that spread out around $\|x\| = 1$. We need the spread condition to ensure the learner has enough data to know the shape of $f_*$ around those critical points. Using a distribution with a concentrated $\|x\|$ does not make much sense for this problem because, in that case, the target function is more or less equivalent to a constant function. However, most of our argument is still valid
in that case, though one may need to maintain a different set of regularity conditions.

**Target function**: Though we use $f_*(x) = \mathrm{ReLU}(1 - \|x\|)$ as our target function, we believe, with some effort, our argument can potentially be extended to functions of form $f_*(x) = g(\|x\|)$, where $g$ is some nice function. In our proof, we use the fact that $f$ is mostly linear to back-propagate the symmetry-related info from the second layer to the first layer. For a more general second layer, this can be done by Taylor expanding the second layer around the infinite-width network.

Suppose that the learner network is $f(x) = h(F(x))$ for some function $h$ and write $\delta(x) = F(x) - \tilde{F}(\|x\|)$. Then, we have $f(x) = h(\tilde{F}(\|x\|) + \delta(x)) \approx h(\tilde{F}(\|x\|)) + h'(\tilde{F}(\|x\|)) \delta(x)$. (For the current setting, $h'(\tilde{F}(\|x\|))$ is approximately $\bar{w}_2$.) Then, the second term of the loss decomposition (Eq. (2)) will become
$$
  \mathbb{E}_x [ (h'(\tilde{F}(\|x\|)) )^2 ( \tilde{F}(x) - F(x) )^2 ]
  =  \mathbb{E}\_{\|x\|} [ (h'(\tilde{F}(\|x\|)) )^2 \|x\|^2 ]
    \mathbb{E}\_{\bar{x} \in \mathbb{S}^{d-1}} [ (\tilde{F}(\bar{x}) - F(\bar{x}) )^2 ].
$$
It still pushes the first layer towards its infinite-width counterpart. The point here is that since the finite-width network is close to the infinite-width network, we can use Taylor expansion to factor out the first-order terms. Then, by determining the signs of those terms as we have done in the current proof, one can determine whether the approximation error will explode.

For a general second-layer function $g$, the convergence and second-layer analysis do need to change a lot. However, provided that the approximation error of the first layer remains small, the problem is 1D and can potentially be solved by discretizing the mean-field network in a brute-force way. We believe that proving similar results for more general function classes is definitely an interesting future direction.

**Network architecture**: Though the network architecture used in the current version might be somewhat specific to this target function, our technique can be adapted to other architectures to learn a broader class of functions.

The dynamics of the first-layer approximation error are relatively independent of the second-layer dynamics, so the choice of the second-layer architecture is quite flexible (See the previous discussion on target functions). The intermediate layer, i.e., the connection between the first and second layers, can also be made to have a dimension larger than $1$. When the weights of this intermediate layer are all non-negative, this is quite straightforward. When negative weights are allowed, one will have to deal with cancellations between positive and negative parts, and things become tricker.

In one of the earlier versions of the paper, we did have both positive and negative parts in the first layer. That is, the first layer was given by $F(x; \mu_{1, +}) - F(x; \mu_{1, -})$, and we used a small initialization for the first layer too. We used an escaping-saddle-points style argument to show that within polynomially long time, we could reach a state similar to the beginning state of Stage 2, and one of $F(x; \mu_{1, \pm})$ will be dominated by the other. Unfortunately, that argument did not synergize well with the projection, which is needed to deal with heavy-tail distributions, and the proof was more cumbersome. Hence, eventually we decided to remove the negative part for a cleaner proof and to highlight the discretizing-mean-field-networks technique. However, if one is only interested in the positive results, which do not need the distribution to be heavy-tailed, one can indeed add a negative part and observe some other interesting phenomena. In that situation, our main technique is still applicable.

---

### Decision · Program_Chairs · 2023-01-20

**Decision:**

Accept: notable-top-25%

**Justification For Why Not Higher Score:**

Nice paper to highlight as a spotlight. Could also be oral.

**Justification For Why Not Lower Score:**

A nice solid results about dept separation. Certainly worth a highlight.

**Metareview: Summary, Strengths And Weaknesses:**

All reviewers agree this paper should be accepted. I think the reviews summarize very well the strengths and weaknesses of the paper as well as points that the authors should include in the revised version. I think this will be a great addition to the conference.

**Note From Pc:**

if the above contains the word "oral" or "spotlight" please see: "oral" presentation means -> notable-top-5% and "spotlight" means -> notable-top-25%. As stated in our emails, we are disassociating presentation type from AC recommendations